# Diabetic sensory neuropathy and insulin resistance are induced by loss of *UCHL1* in *Drosophila*

Daewon Lee [1,2,5], Eunju Yoon [1,2,5], Su Jin Ham[1,2,5], Kunwoo Lee [3], Hansaem Jang [4], Daihn Woo [1], Da Hyun Lee [1,2], Sehyeon Kim [1,2], Sekyu Choi [3,4] ✉ & Jongkyeong Chung [1,2] ✉

Diabetic sensory neuropathy (DSN) is one of the most common complications of type 2 diabetes (T2D), however the molecular mechanistic association between T2D and DSN remains elusive. Here we identify ubiquitin C-terminal hydrolase L1 (UCHL1), a deubiquitinase highly expressed in neurons, as a key molecule underlying T2D and DSN. Genetic ablation of *UCHL1* leads to neuronal insulin resistance and T2D-related symptoms in *Drosophila*. Furthermore, loss of *UCHL1* induces DSN-like phenotypes, including numbness to external noxious stimuli and axonal degeneration of sensory neurons in flies' legs. Conversely, *UCHL1* overexpression improves DSN-like defects of T2D model flies. UCHL1 governs insulin signaling by deubiquitinating insulin receptor substrate 1 (IRS1) and antagonizes an E3 ligase of IRS1, Cullin 1 (CUL1). Consistent with these results, genetic and pharmacological suppression of CUL1 activity rescues T2D- and DSN-associated phenotypes. Therefore, our findings suggest a complete set of genetic factors explaining T2D and DSN, together with potential remedies for the diseases.

Diabetes mellitus is a metabolic disease that causes abnormally high blood glucose levels. Increased glycemia in diabetic patients is mainly attributed to two reasons: decreased insulin secretion or increased insulin resistance. When the pancreas secretes insulin into the blood, insulin binds to insulin receptor (IR) in target cells and activates insulin receptor substrate 1 (IRS1), a signaling adapter protein[1]. IRS1 plays a crucial role in passing signals from IR to intracellular signaling molecules, such as phosphoinositide 3-kinase (PI3K) and Akt (also called protein kinase B)[2,3]. Activated Akt regulates diverse cellular events, including protein translation, glycogen synthesis, and glucose uptake. Therefore, decreased insulin secretion or increased insulin resistance hampers intracellular insulin signaling and contributes to the development of diabetes, either type 1 diabetes (T1D) or type 2 diabetes (T2D), respectively.

Most patients with T1D or T2D suffer from diabetic complications. Among them, diabetic neuropathy (DN), including diabetic sensory neuropathy (DSN), is the most common complication that lowers the quality of life of diabetic patients. The main symptom of DN is numbness, which begins from the plantar regions and progresses proximally to the upper legs, caused by malfunction of leg sensory neurons[4]. So far, several treatment methods and therapeutic agents for diabetes and DN exist, yet limitations are still prevalent. Insulin injection corrects both hyperglycemia and DN in T1D patients[5]. Metformin, one of the most effective drugs for treating T2D, also improves hyperglycemia in T2D patients; however, it barely mitigates DN in T2D patients[6]. In line with this result, glycemic control in T2D patients is insufficient to alleviate DN in T2D patients[7,8]. Consequently, studies on the underlying mechanisms of insulin resistance specified in neurons,

[1]Institute of Molecular Biology and Genetics, Seoul National University, Seoul 08826, Republic of Korea. [2]School of Biological Sciences, Seoul National University, Seoul 08826, Republic of Korea. [3]School of Interdisciplinary Bioscience and Bioengineering, Pohang University of Science and Technology, Pohang 37673, Republic of Korea. [4]Department of Life Sciences, Pohang University of Science and Technology, Pohang 37673, Republic of Korea. [5]These authors contributed equally: Daewon Lee, Eunju Yoon, Su Jin Ham. ✉e-mail: sekyuchoi@postech.ac.kr; jkc@snu.ac.kr

especially sensory neurons, are necessary to comprehend the pathogenesis of DSN in T2D patients.

Researchers have identified many genetic and environmental factors that contribute to the development of T2D. Notably, numerous studies have proposed the downregulation of insulin signaling caused by impairments of IRS1 as a common and comprehensive explanation for insulin resistance and T2D pathologies[3]. As IRS1 is a large protein of about 180 kDa with various amino acid residues for post-translational modifications (PTMs), PTM-mediated regulation of IRS1 is crucial for understanding insulin resistance[9]. In particular, the regulation of IRS1 protein stability by site-specific ubiquitination has been extensively studied. Several E3 ligases, including Mitsugumin-53 (MG53, also called TRIM72), Cullin 3 (CUL3)-, and Cullin 7 (CUL7)-RING E3 ubiquitin ligases, are known to ubiquitinate and destabilize IRS1. Indeed, mice with the functional loss of each of these genes recover from insulin resistance developed by high-fat diet (HFD) or high-sucrose diet (HSD)[10–12]. However, deubiquitinases (DUBs) for IRS1 that can function as IRS1 stabilizers by antagonizing the E3 ligases for IRS1 have rarely been studied. For these reasons, identifying DUBs for IRS1 and observing the in vivo phenotypes of their knockout or transgenic animal models are critical.

UCHL1 is a DUB belonging to the UCH family and is expressed abundantly in the nervous systems accounting for about 1% of the total proteins of human brains[13]. Therefore, *UCHL1* has been suggested as a gene closely associated with neurodegeneration. For example, an I93M substitution of *UCHL1* was identified in one German family suffering from Parkinson's disease (PD)[14]. Furthermore, *UCHL1* knockout mice display axonal degeneration in the peripheral neurons and gracile axonal dystrophy (GAD)[15,16]. However, the molecular mechanisms of how UCHL1 malfunctions induce neurodegenerative symptoms have scarcely been studied. Notably, previous genome-wide association studies (GWAS) suggested that *UCHL1* has functional associations with T2D development[17]. Additionally, the expression of *UCHL1* has been reported to decrease in human pancreatic islets from obese patients with T2D[18]. Since fine-tuned transmission of insulin signaling is crucial for the maintenance of neuronal physiology, we attempted to investigate the associations between neurodegeneration and insulin signaling using *Drosophila UCHL1* models.

In this study, we discovered that knockout of *UCHL1* induces neuron-specific insulin resistance, T2D-associated symptoms, and DSN-like defects in *Drosophila*. Both *Drosophila* and mammalian studies revealed that UCHL1 regulates insulin signaling by deubiquitinating IRS1. We also identified that the E3 ligase CUL1 can be induced by HSD and specifically antagonizes the DUB activity of UCHL1 on IRS1. Therefore, we suggest that the antagonistic control of IRS1 protein stability by UCHL1 deubiquitinase and CUL1-RING E3 ligase is the main regulatory mechanism involved in the pathogenesis of T2D and DSN.

## Results
### Loss of *UCH* induces T2D-like phenotypes in *Drosophila*
Since the *Drosophila UCH* gene (*cg4265*) is the homologous gene for human *UCHL1*, we employed *UCH* knockout (*UCH^KO^*) and *UCH* C93S knockin (*UCH^C93S^*), encoding a catalytic dead form of UCH, mutant fly lines to observe diabetes-related phenotypes. The two mutants were generated using the clustered regularly interspaced short palindromic repeats (CRISPR)-Cas9 system[19]. Surprisingly, the hemolymph glucose levels of *UCH^KO^* and *UCH^C93S^* flies were increased by approximately 50% compared to those of control flies at 3rd-instar larval, 3-day-old adult, and 30-day-old adult stages (Fig. 1a and Supplementary Fig. 1a). Elevated levels of triglyceride (TAG) (Fig. 1b and Supplementary Fig. 1b), glycogen (Fig. 1c and Supplementary Fig. 1c), and trehalose (Supplementary Fig. 1d) were also observed in the larval and adult stages of the two *UCH* mutants. Subsequently, we measured gene expression levels of *Drosophila insulin-like peptides* (*DILP*s), which encode the homologs of human insulin, to determine whether the increased hemolymph metabolites

observed above were caused by DILPs deficiency. Interestingly, the gene expression of all *DILP*s (*DILP1-7*) was slightly increased in the two different *UCH* loss-of-function mutants (Fig. 1d). To firmly prove this, we stained DILP2 neurons using anti-DILP2 antibody[20]. This approach was chosen due to the robust functional connections between DILP2 and human insulin. We observed a statistically significant increase in DILP2 staining intensity in *UCH^KO^* and *UCH^C93S^* flies compared to control flies (Supplementary Fig. 1e). Additionally, we generated flies expressing GFP under control of the *DILP2*-GAL4 driver (*DILP2 > GFP*) and found that the GFP intensity was also significantly elevated by the *UCH* mutations (Supplementary Fig. 1f). Furthermore, we performed ELISA against Flag antigen using the extracted hemolymph of *DILP2 > DILP2-Flag* flies[20]. Our results showed that the levels of Flag-tagged DILP2 were increased in *UCH^KO^* and *UCH^C93S^* flies (Supplementary Fig. 1g). Since T2D patients and model animals are known for elevated blood metabolites with the increased secretion of insulin[21], we postulated that T2D-related defects were induced by *UCH* mutations. Furthermore, to monitor feeding behaviors, the mutant flies were fed a diet composed of green-dyed food for 1 h after 1 day of fasting, providing visualization of food uptake of these flies. We found that 30-day-old *UCH^KO^* and *UCH^C93S^* flies showed increased food intake compared to *w1118* control flies (Fig. 1e). We further measured the exact volumes of food ingestion by *Drosophila* using spectrophotometric assay and CApillary FEeder (CAFE) assay. Here, we identified that the food intake was increased by *UCH* mutations only with increasing age (Fig. 1f and Supplementary Fig. 1h), which is one of the main phenotypes in T2D patients and model animals[22].

To mimic the glucose tolerance test (GTT) and insulin tolerance test (ITT) performed in humans and mice, we adapted and made these tests suitable for *Drosophila*. For GTT, flies were fasted for 1 day before being fed a diet containing 10% sucrose for 1 h. The flies were then again fasted and their hemolymph glucose levels during this second fasting were measured at 30-min intervals. In *w1118* flies, the increased glucose levels observed after the 1-h sucrose feeding returned to the levels measured prior to the 1-day fasting within 2-3 h. In contrast, *UCH^KO^* and *UCH^C93S^* flies took a significantly longer time for their blood glucose levels to decrease after the 1-h sucrose feeding (Fig. 1g and Supplementary Fig. 1i). For ITT, we transiently expressed DILP2, which has strong functional associations with human insulin, in *DILP2* neurons. Here, human insulin was not directly injected into the flies, as the anesthesia used during insulin injection could hinder the measurement of the real-time glycemia of fruit flies. We generated fly lines overexpressing *DILP2* conditionally in *DILP2* neurons (*DILP2*-GAL4 > *UAS-DILP2*) under high temperatures (29 °C), but not under low temperatures (19 °C) using *tub*-GAL80^TS^. Flies were fasted for 6 h at low temperature conditions before being transferred to high temperature conditions for 30 min, during which *DILP2* was overexpressed. Hemolymph glucose levels were then measured at 15-min intervals during which fasting conditions was maintained. These levels were then normalized to the baseline levels of each genotype, which were measured before *DILP2* expression. Surprisingly, the hemolymph glucose levels of control flies dramatically decreased upon induction of exogenous DILP2 expression. However, the glucose levels of *UCH^KO^* or *UCH^C93S^* flies did not decline as much as those of control flies after conditional induction of DILP2 expression (Fig. 1h and Supplementary Fig. 1j). Therefore, both our GTT and ITT experiments demonstrated that diabetes-like phenotypes are observed in fruit flies with *UCH* mutations.

As *UCHL1* is abundantly expressed in the nervous systems[13], we tested the tissue-specific effects of *UCH* deficiency in developing T2D-related symptoms using several GAL4 drivers, including *collagen* (*cg*)-GAL4, *myocyte enhancer factor 2* (*mef2*)-GAL4, and *embryonic lethal abnormal vision* (*elav*)-GAL4. Intriguingly, hyperglycemia was not induced when *UCH* was knocked down in the fat body or muscle with *cg*-GAL4 or *mef2*-GAL4 drivers, respectively. However, nervous system-specific knockdown of *UCH* using the *elav*-GAL4 driver elevated hemolymph glucose levels as much as those of *UCH^KO^* mutants (Fig. 1i).

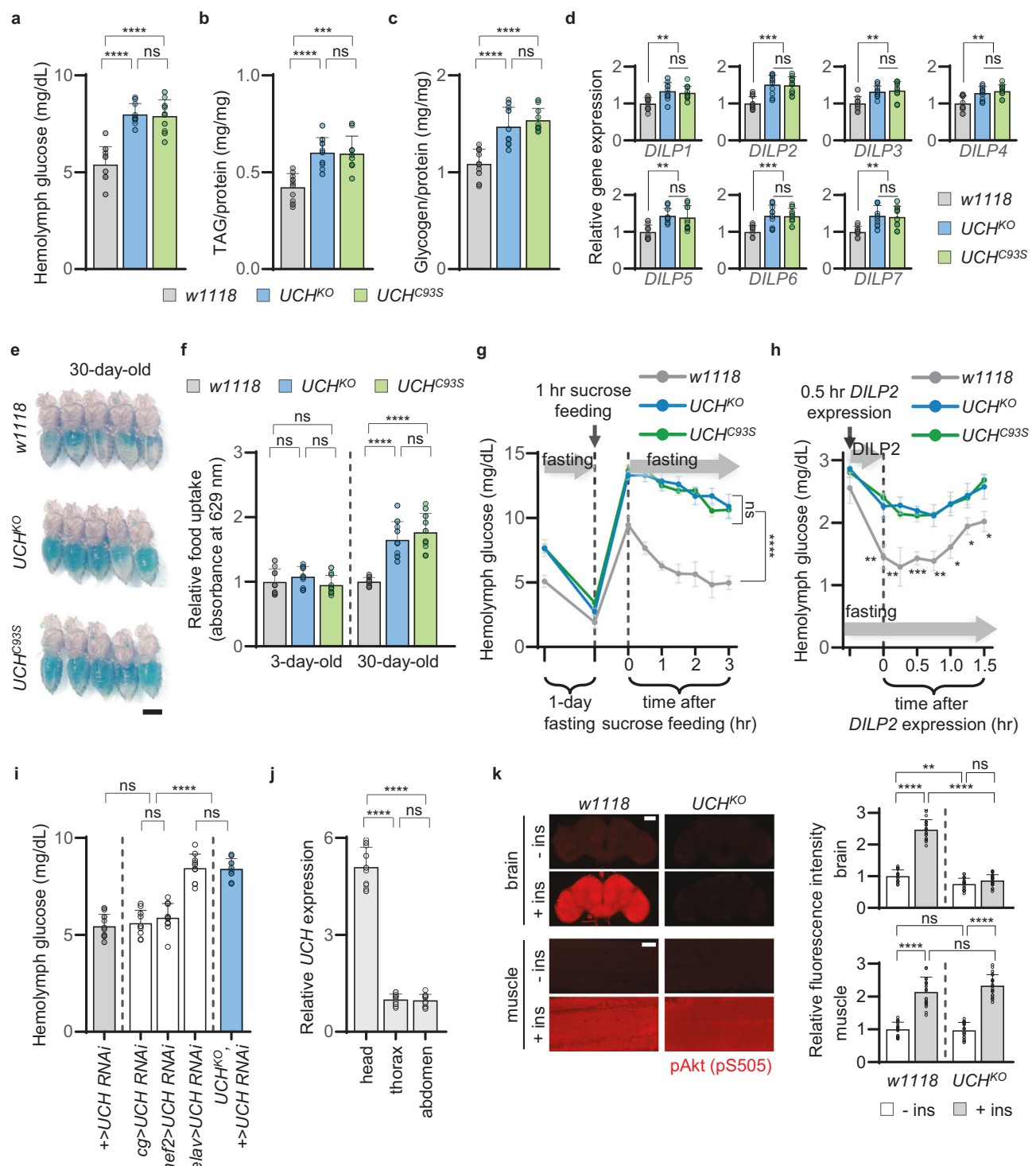

We subsequently measured the expression levels of *UCH* in the heads, thoraces, and abdomens of wild-type flies. The transcript levels of *UCH* in thoraces were similar to those in abdomens, but *UCH* expression was dramatically high in fly heads (Fig. 1j). Then, to identify whether there were differences in insulin sensitivity in nervous systems or muscles between control flies and *UCH* mutant flies, we measured phosphorylation of Akt in the fly brains and indirect flight muscles under human insulin treatment ex vivo. Our findings revealed that Akt phosphorylation in the brains of *UCH^KO* flies was significantly lower than that of control flies even in the absence of insulin treatment; however, there were no significant differences in muscles in this condition. Following insulin treatment, Akt phosphorylation increased in both the brains

and muscles of *w1118* flies. In *UCH^KO* flies, Akt phosphorylation increased in response to insulin treatment in the muscles, while there were no significant increases in the brains (Fig. 1k). Taken together, we concluded that the loss-of-function mutations of *UCH* induce insulin resistance primarily in the nervous systems due to the nervous system-specific expression of *UCH*.

**T2D-like phenotypes of high-sucrose diet (HSD) flies are rescued by the expression of exogenous *UCH***

Having shown that *UCH* deletion induces insulin resistance in the nervous systems, we sought to test whether insulin sensitivity could be enhanced by the expression of exogenous *UCH*. We first generated

**Fig. 1 | Loss of *UCHL1* induces T2D-related pathologies and neuronal insulin resistance in *Drosophila*. a** Glucose concentrations in the hemolymph of 3-day-old flies. $n = 10$. **b** TAG concentrations per total body protein of 3-day-old flies. $n = 10$. **c** Glycogen concentrations per total body protein of 3-day-old flies. $n = 10$. **d** Relative expressions of *DILP1-7* normalized to the *ribosomal protein 49* (*rp49*) expression of 3-day-old flies. $n = 10$. **e** Representative images of 30-day-old flies feeding 10% sucrose solution containing green dye for 1 h after 1-day fasting. Scale bar, 0.5 mm. **f** Measuring food uptake of 3-day-old and 30-day-old flies using spectrophotometric assay. $n = 10$. **g** Glucose concentrations in the hemolymph of the flies. Each 3-day-old fly was on 10% sucrose solution for 1 h after 1-day fasting. After the 1-h sucrose feeding, the hemolymph glucose levels were measured every 30 min while fasting the flies again. The indicated statistical significance was calculated from the data after the 1-h feeding. $n = 3$. Absolute value of the slope from right after the sucrose feeding to 3 h after the sucrose feeding was depicted in Supplementary Fig. 1i. **h** Glucose concentrations in the hemolymph of 3-day-old flies. The flies fasted for 6 h before *DILP2* expression. After the *DILP2* expression for

0.5 h, the hemolymph glucose levels were measured every 15 min. The bar graph at the upper side represents the area under curve (AUC) of the line graph in the same panel. The relative concentrations of hemolymph glucose normalized to the glucose levels of each genotype before DILP2 expression (from baseline) were depicted in Supplementary Fig. 1j. $n = 3$. **i** Glucose concentrations in the hemolymph of 3-day-old flies. $n = 10$. **j** Relative expressions of *UCH* from the head, thorax, and abdomen of *w1118*. Normalized to the *rp49* expression in the same body part. $n = 10$. **k** Left, confocal immunofluorescence images of the brain or muscle with or without treating 1 µM human insulin for 20 min, stained with anti-pAkt (pS505) antibody. Right, the relative fluorescence intensity of pAkt staining normalized to the intensity of *w1118* without insulin incubation. $n = 20$. Scale bar, 50 µm (brain images). Scale bar, 20 µm (muscle images). Data are presented as mean ± SD. One-way ANOVA with Tukey's multiple comparison test was used (**a–d**, **f–j**). Two-way ANOVA with Sidak's multiple comparison test was used (**k**). *$p < 0.05$. **$p < 0.01$. ***$p < 0.001$. ****$p < 0.0001$. ns no significant.

T2D model flies by feeding a high-sucrose diet (HSD), which induced hyperglycemia in the flies. Here, neuronal overexpression of *UCH* under the *elav*-GAL4 driver slightly decreased the hyperglycemia of HSD flies by about 30%. Unexpectedly, the expression of exogenous *UCH* in the muscle or fat body of HSD flies also reduced the increased glycemia by about 30% (Supplementary Fig. 1k). We then drove the whole-body expression of *UCH* using heat-shock (*hs*)-GAL4, which fully rescued the T2D-related phenotypes of HSD flies, including hyperglycemia (Supplementary Fig. 1k, l), increased TAG levels (Supplementary Fig. 1m), and impaired glucose tolerance (Supplementary Fig. 1n). These results indicated that *UCH* played a role in preventing T2D-like symptoms developed by HSD. However, as endogenous *UCH* is poorly expressed in tissues other than nervous systems, we suggested that the physiological role of *UCH* is to maintain insulin sensitivity against HSD specifically in the nervous system.

**Degeneration of sensory neurons in the legs progressively develops with aging in *UCH* mutants**
For further investigation, we tried to elucidate the specific type of neuron that expresses *UCH* most abundantly. Using all RNA-sequencing and chromatin immunoprecipitation (ChIP)-seq sample and signature search (ARCHS4), an integrative gene expression database with various human and mouse RNA-sequencing, we found that sensory neurons express *UCH* more than any other tissues[23]. Likewise, *UCHL1* has been regarded as a marker gene of sensory neurons[24,25]. Moreover, since *UCH* mutations induced T2D-like phenotypes in *Drosophila* as shown above, we postulated that DSN, a diabetic complication caused by impairments of sensory neurons in the legs of T2D patients, would appear in *UCH* mutant flies. We, therefore, attempted to observe DSN-related phenotypes using the front (prothoracic) legs of flies, whose primary role is perceiving external stimuli[26].

First, we measured the transcript levels of *UCH* in sensory neurons of the front legs of flies. Since fly legs are covered with hard cuticle layers, intact sensory neurons could not be easily dissociated from the legs. Thus, we extracted the nuclei of sensory neurons from the legs through the expression of *lamin-GFP* under the *OK371*-GAL4, a driver that specifically targets sensory neurons[27]. To compare the levels of *UCH* expression from other tissues, especially neurons, we further expressed *lamin-GFP* using *hs*-GAL4, *elav*-GAL4, and *OK6*-GAL4 for the extraction of nuclei from whole tissues, neurons, and motor neurons, respectively[28]. From these four genotypes (*hs* > *lamin-GFP*, *elav*>*lamin-GFP*, *OK6* > *lamin-GFP*, and *OK371* > *lamin-GFP*), we extracted GFP-positive (GFP⁺) nuclei from either the entire body or dissected front legs using the fluorescence-activated nuclei sorting (FANS) method and measured the expression of *UCH* (Supplementary Fig. 2a). According to the expression results from entire bodies, the relative transcription levels of *UCH* from whole neurons (*elav* > *lamin-GFP*) were higher than those from whole tissues (*hs* > *lamin-GFP*).

Intriguingly, compared to whole neurons, the relative expression of *UCH* was lower in motor neurons (*OK6* > *lamin-GFP*) and higher in sensory neurons (*OK371* > *lamin-GFP*) (Supplementary Fig. 2b). Furthermore, according to expression results from the front legs, the transcription of *UCH* in front-leg sensory neurons was significantly more abundant than that in sensory neurons of the whole body. Since the nuclei of the motor neurons are located in the ventral nerve cord (VNC) in fruit flies, RNA was poorly acquired from the front-leg samples of *OK6* > *lamin-GFP* (Supplementary Fig. 2b). These findings, along with previously published data, confirmed that *UCH* is highly expressed in the sensory neurons of *Drosophila*, especially in the legs.

Next, we observed sensory neurons in the fly's front legs. The front legs of *Drosophila* are anatomically separated into three parts: femur, tibia, and tarsus. In particular, the tarsus is further subdivided into five tarsal segments, where sensory neurons are mainly located (Fig. 2a). Hence, using GFP-tagged with nuclear localization signals (nlsGFP) and the *OK371*-GAL4 driver, we investigated the sensory somas at tarsal segments 3, 4, and 5, which can perceive external stimuli directly from the surface similar to human feet. The number of sensory somas in control flies (*w1118*), *UCH^KO^*, and *UCH^C93S^* mutants was similar when the flies were young (3-day-old). Although the numbers of sensory somas in 30-day-old *w1118* was similar to that in 3-day-old *w1118*, the numbers of sensory somas in *UCH^KO^* or *UCH^C93S^* flies dramatically decreased when the flies grew old (30-day-old) and there were considerable differences in the numbers of sensory somas between control flies and *UCH* mutants (Fig. 2b). To identify whether the loss of sensory neurons was caused by apoptosis, we performed a terminal deoxynucleotidyl transferase dUTP nick end labeling (TUNEL) assay. Apoptotic signals were highly elevated in the sensory somas of *UCH^KO^* and *UCH^C93S^* fly legs at 20-day-old (Fig. 2c). Since degeneration of the leg sensory neurons was observed in the aged *UCH* mutants, we investigated the sensitivity of the mutants in response to external stimuli. Flies were placed onto a hot plate at 42 °C or an acidic plate-filled with 12% sulfuric acid, either of which causes detrimental effects to *Drosophila* (Fig. 2d)[29]. We subsequently measured the time when flies exhibited the first escape (jumping) response under the two different stimuli. All *w1118*, *UCH^KO^*, and *UCH^C93S^* flies displayed escape responses below 6 s to heat and acid stimuli at 3 days of age; however, as they grew older, the two mutants exhibited significantly delayed responses to the same stimuli, even reaching response times of over 10 s, in comparison with *w1118* control flies (Fig. 2e, f). Furthermore, to explore the possible link between *UCH* mutations and the dysfunction of other peripheral neurons, such as motor neurons, in the legs, we observed locomotor activity of *UCH* mutant flies. No differences in locomotor activity, including moving patterns and walking distance, were observed between *UCH* mutants and control flies (Supplementary Fig. 2c, d). Taken together, we concluded that *UCH*

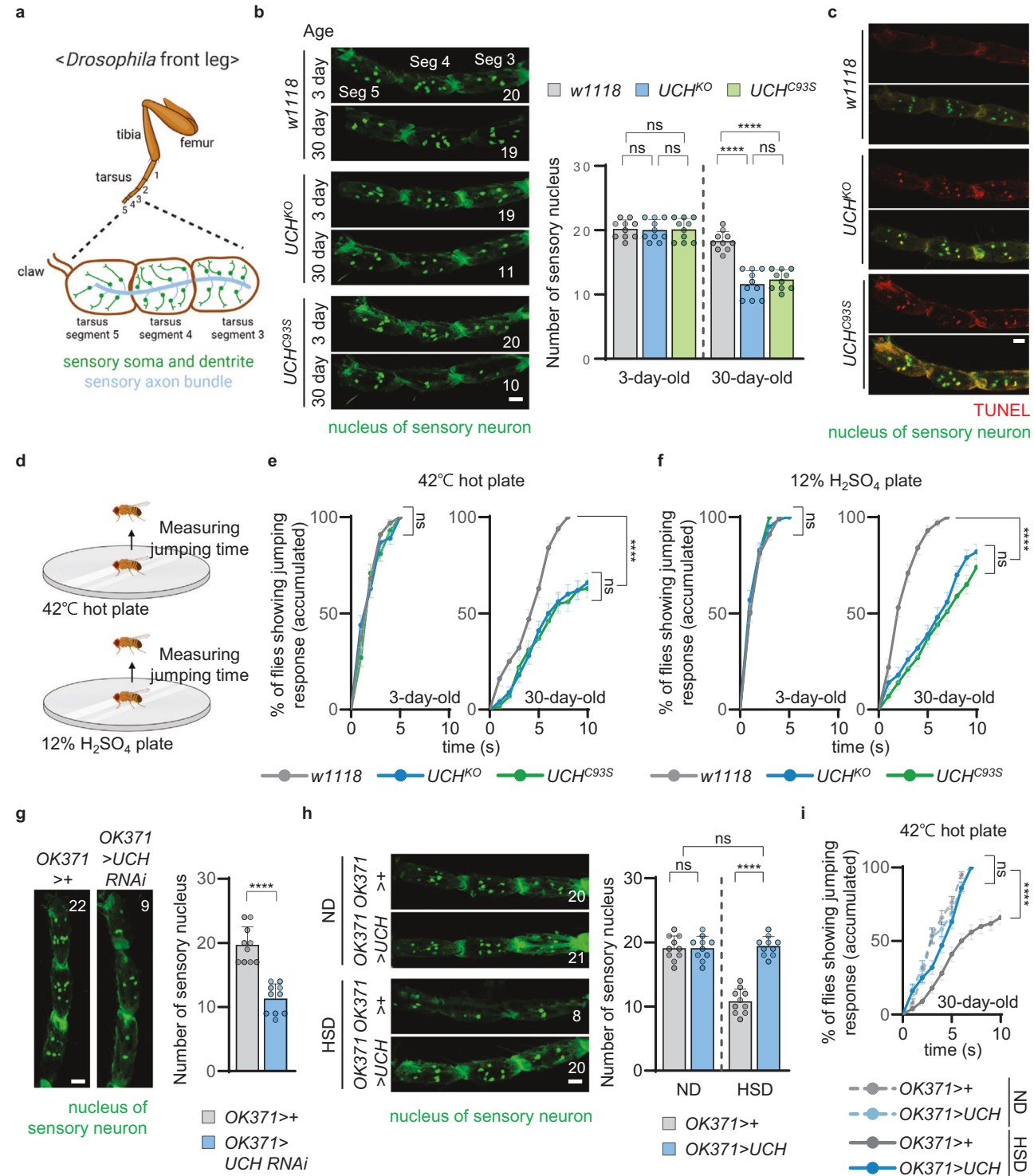

mutations induce specific loss of the leg sensory neurons as well as consequent numbness to external stimuli.

## Autonomous degeneration of sensory neurons in *UCH* mutant flies

We found that sensory neuron-specific knockdown of *UCH* caused degeneration of the leg sensory neurons and decreased sensory responses (Fig. 2g and Supplementary Fig. 2e). Nevertheless, the loss of *UCH* in sensory neurons was not sufficient to increase glycemia in flies (Supplementary Fig. 2f). Additionally, as the expression of exogenous *UCH* rescued the diabetic phenotypes developed by HSD, we sought to

examine whether *UCH* expression could rescue sensory neuronal defects in HSD flies. First, we tested if HSD could induce the defects in sensory responses observed in *UCH^KO* flies. Notably, the number of the front-leg sensory neurons decreased in HSD flies when aged to 30 days (Fig. 2h), and nociception to heat stimuli was also reduced in HSD flies at older ages (Fig. 2i). Surprisingly, the sensory neuron-specific expression of *UCH* using *OK371*-GAL4 driver mitigated the impairments of the leg sensory neurons developed by HSD (Fig. 2h) and improved the decreased escape responses from heat stimuli (Fig. 2i). However, *UCH* expression in sensory neurons was not enough to improve the elevated glycemia caused by HSD (Supplementary Fig. 2g). Likewise, the sensory

**Fig. 2 | Loss of *UCHL1* progressively develops degeneration of leg sensory neurons and leg numbness to external stimuli. a** Schematic diagram of *Drosophila* front leg showing sensory neurons. The illustration was created using Biorender.com. **b** Left, confocal fluorescence images of tarsal segments 3, 4, and 5 at the front legs of 3-day-old or 30-day-old flies expressing *OK371 > nlsGFP*. The number in panels indicates the number of green signals in each image. Respective images were obtained from one of the left or right front legs. Green, the nucleus of sensory neuron. Scale bar, 20 μm. Right, the numbers of green signals at the tarsal segments 3, 4, and 5 of the front legs of 3-day-old or 30-day-old flies expressing *OK371 > nlsGFP*. *n* = 10. **c** Confocal fluorescence images for TUNEL assays of tarsal segments 3, 4, and 5 at the front legs of flies expressing *OK371 > nlsGFP*. Respective images were obtained from one of the left or right front legs. Red, TUNEL signal. Green, the nucleus of sensory neuron. Scale bar, 20 μm. **d** Illustration of the experiments for measuring *Drosophila* escape responses. The illustration was created using Biorender.com. **e** Cumulative percentage of 3-day-old (left) or 30-day-old (right) flies showing escape responses on the 42 °C hot plates within 10 s. *n* = 100. **f** Cumulative percentage of 3-day-old (left) or 30-day-old (right) flies showing escape responses on the 12% H$_2$SO$_4$ plates within 10 s. *n* = 100. **g** Left, confocal fluorescence images of tarsal segments 3, 4, and 5 at the front legs of 30-day-old flies expressing *OK371 > nlsGFP*. The number in panels indicates the number of green signals in each image. Respective images were obtained from one of the left or right front legs. Green, the nucleus of sensory neuron. Scale bar, 20 μm. Right, the numbers of green signals at the tarsal segments 3, 4, and 5 of the front legs of 30-day-old flies expressing *OK371 > nlsGFP*. *n* = 10. **h** Left, confocal fluorescence images of tarsal segments 3, 4, and 5 at the front legs of 30-day-old flies expressing *OK371 > nlsGFP* upon ND or HSD. The number in panels indicates the number of green signals in each image. Respective images were obtained from one of the left or right front legs. Green, the nucleus of sensory neuron. Scale bar, 20 μm. Right, the numbers of green signals at the tarsal segments 3, 4, and 5 of the front legs of 30-day-old flies expressing *OK371 > nlsGFP* upon ND or HSD. *n* = 10. **i** Cumulative percentage of 30-day-old flies upon ND or HSD showing escape responses on the 42 °C hot plates within 10 s. *n* = 100. ND, normal diet. HSD, high-sucrose diet. Data are presented as mean ± SD. Two-way ANOVA with Sidak's multiple comparison test was used (**b** and **h**). Mantel–Cox test was used (**e**, **f**, and **i**). Two-tailed paired Student's *t*-test was used (**g**). ****$p < 0.0001$. ns no significant.

---

neuron-specific expression of *UCH* in *UCH$^{KO}$* flies rescued the sensory neuronal defects (Supplementary Fig. 2h, i), but did not improve their hyperglycemia (Supplementary Fig. 2j). Collectively, these results suggested that increased insulin resistance of the sensory neurons induced by the loss of *UCH* or high-sucrose feeding leads to autonomous degeneration of sensory neurons. Moreover, considering that sensory neurons represent only a small fraction of the entire neuronal population, a decrease in glucose uptake specifically in sensory neurons (*OK371 > UCH RNAi*) may not alone lead to a statistically significant increase in hemolymph glucose levels. However, when *UCH* was completely knocked out in the entire body (*UCH$^{KO}$*) or selectively downregulated in all neuronal types (*elav > UCH RNAi*), including sensory neurons, motor neurons, and interneurons, a reduction in glucose uptake can be induced across multiple neuronal populations. Consequently, our findings suggested that as the number of neurons with impaired glucose uptake increases, the elevation of hemolymph glucose levels become more pronounced in fruit flies.

### Axonal degeneration precedes apoptotic loss of sensory neurons in *UCH* mutant flies

Since our findings had suggested that decreased escape responses to noxious stimuli were a result of the apoptotic loss of the leg sensory neurons, we conducted further investigations to determine whether blocking apoptosis could alleviate sensory neuron loss and impaired escape responses. To achieve this, we generated flies that could block apoptosis in sensory neurons by knocking down *death-related ICE-like caspase* (*Drice*; the *caspase 3* homolog in *Drosophila*[30]), one of the primary triggers of apoptosis, using *OK371*-GAL4. *Drice* knockdown in *UCH$^{KO}$* flies eliminated the apoptotic signals in their sensory neurons at tarsal segments 3, 4, and 5 (Fig. 3a) and alleviated loss of the leg sensory neurons (Fig. 3b). However, the decreased escape responses to heat stimulation were only partially rescued by blocking apoptosis (Fig. 3c), suggesting that the apoptotic loss of sensory neurons alone may not fully account for the decreased escape responses of *UCH* knockout flies.

We, therefore, sought to assess axonal atrophy, known as the primary cause of DSN in human patients[6,31], in the leg sensory neurons of *UCH$^{KO}$* flies. To visualize the axons and somas of sensory neurons located just beneath the surface cuticle layer of tarsal segment 4 in the front legs, we employed the sensory neuron-specific *OK371*-GAL4 driver to express GFP. We observed four sensory neurons in this region, with each axon bundled at the center of the leg and extending to the ventral nerve cord (VNC), which is analogous to the vertebrate spinal cord (Fig. 3d). Due to the hard cuticle layers of flies, it was challenging to visualize the axons of sensory neurons located deep within the legs of fruit flies. As a result, in the investigation using *OK371 > nlsGFP*, eight sensory somas were observed (Fig. 2a, b), while in the observation with

*OK371 > GFP*, only four sensory neurons were visible (Fig. 3d, e). Control flies displayed four sensory neurons with well-connected axons (highlighted by white triangles) that remained intact from 5 to 25 days of age. In contrast, while *UCH$^{KO}$* flies showed four sensory neurons with initially well-connected axons at 5 days of age, the axons began to sever and disconnect (highlighted by red triangles) from the central bundle by 10 days of age. By 15 days of age, all observed axons in tarsal segment 4 had degenerated, and the loss of sensory neurons was observed from 20 days of age. Surprisingly, *UCH$^{KO}$* flies with *Drice* knockdown still showed the sensory axon degeneration after 10 days of age as found in *UCH$^{KO}$* flies. However, sensory neurons were not lost in these flies at 20 and 25 days of age, despite all axons being disconnected (Fig. 3e–g). Additionally, we measured escape responses to heat stimulation in the flies with these genotypes at 5, 10, 15, 20, and 25 days of age. We found that the escape responses to heat stimulation in *UCH$^{KO}$* and *UCH$^{KO}$* with *Drice* knockdown were significantly decreased after 10 days of age, when sensory axons began to degenerate. Surprisingly, at 20 days of age, when only *UCH$^{KO}$* flies showed loss of sensory neurons, *UCH$^{KO}$* flies exhibited steeper decline in heat escape responses compared to *UCH$^{KO}$* flies with *Drice* knockdown (Fig. 3h, i). Thus, our findings suggested that apoptosis of sensory neurons occurs after axonal degeneration in the leg sensory neurons, and that the decreased escape responses in *UCH$^{KO}$* flies result from the combined effects of axonal degeneration and apoptotic cell death in sensory neurons.

For further investigation, we aimed to determine whether axonal degeneration of sensory neurons led to the apoptotic loss of sensory neurons. As genetic strategies to induce or prevent only axonal atrophy in fruit flies has yet to be well established, we utilized conditional *UCH* expression using *tub*-GAL80$^{TS}$ and overexpressed *UCH* using *OK371*-GAL4 at different time points in both control and *UCH$^{KO}$* flies. When UCH was continuously expressed in *UCH$^{KO}$* from 10 days of age, approximately half of the sensory axons were severed by 15 days. Interestingly, while these disconnected neurons were lost with age, the other half of the sensory neurons with intact axons persisted with age (Supplementary Fig. 3e). Additionally, when *UCH$^{KO}$* flies expressed UCH continuously from 15 days of age, all sensory neuron axons were disconnected, followed by the progressive death of these neurons with age (Supplementary Fig. 3f). Although our results did not provide direct evidence that sensory neurons with impaired axons undergo apoptosis, these findings suggested that the death of sensory neurons occurs only in those with axonal atrophy in *UCH* knockout flies.

### T2D- or DSN-like phenotypes resulting from *UCH* mutations do not occur through mechanisms other than insulin resistance

As *UCH* is predominantly expressed in the nervous system, mutations in *UCH* could potentially impact development or behavior of *UCH*

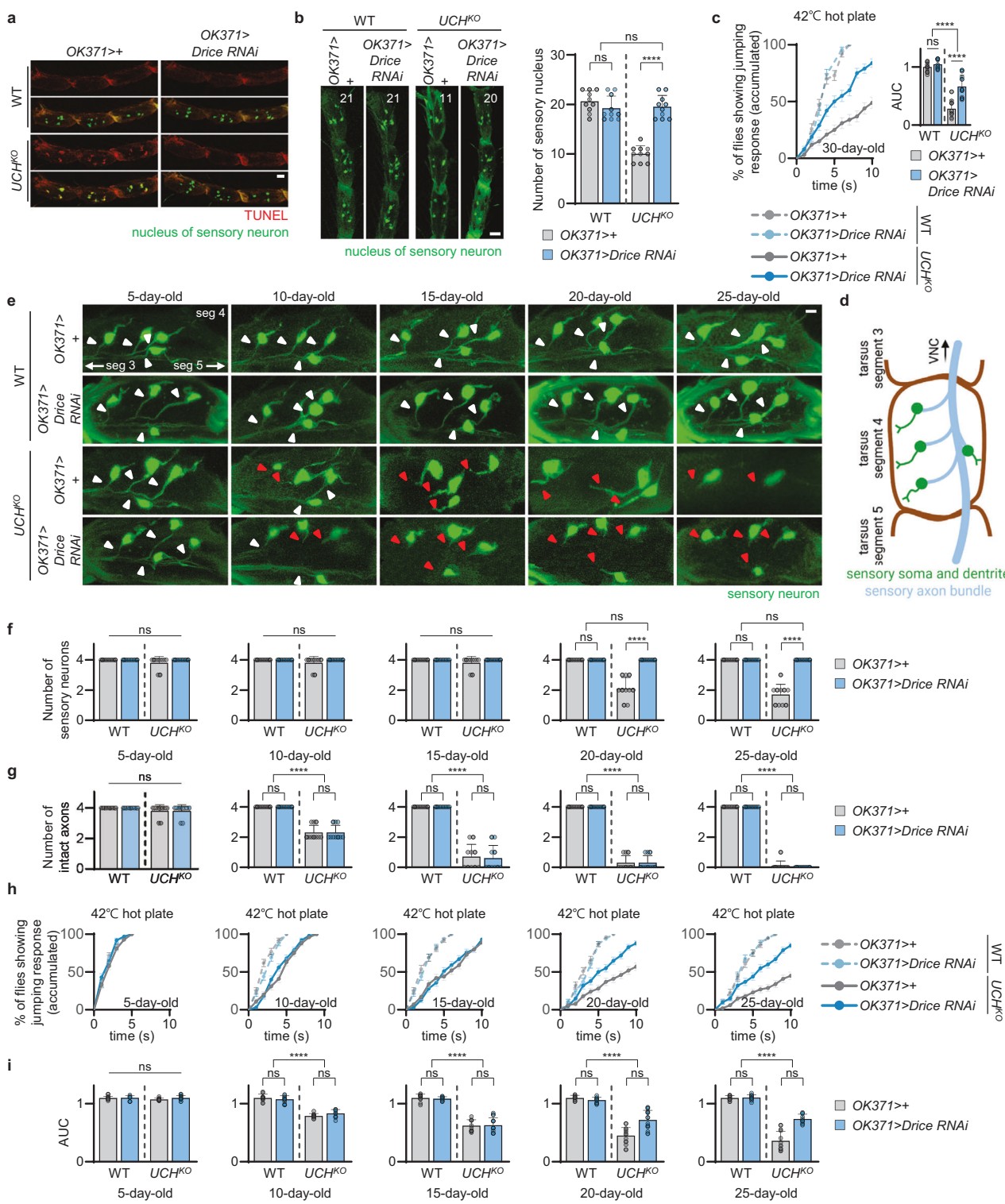

mutant flies, which could in turn lead to indirect changes in metabolism and other metabolic parameters, such as hemolymph glucose levels. To rule out the possibility that any developmental defects or abnormal behaviors were affecting the T2D- or DSN-associated phenotypes in *UCH* mutant flies, we conducted additional experiments. First, we generated flies with inducible *UCH* knockdown using the *UCH* RNAi, *tub*-GAL4, and *tub*-GAL80[TS]. The flies were raised at 18 °C during larval stages to prevent the expression of *UCH* RNAi. After eclosion, they were kept at 29 °C for 3 days to induce *UCH* knockdown, after which hemolymph glucose levels were measured. Similar to *UCH*

mutant flies, the flies expressing *UCH* RNAi only in adult stages had higher levels of hemolymph glucose compared to control flies (Supplementary Fig. 4a). Additionally, we used sensory neuron-specific *OK371*-GAL4, *tub*-GAL80[TS], and *GFP* or *nlsGFP* to express *UCH* RNAi in adult stages for 15 days or 30 days to observe sensory neuron-related phenotypes. Our observations revealed that after a 15-day induction of *UCH* RNAi, the axons of sensory neurons in the tarsal segment 4 of the front legs exhibited degeneration (Supplementary Fig. 4b). Furthermore, following a 30-day induction of *UCH* RNAi, there was a notable reduction in the number of sensory neuron nuclei within tarsal

**Fig. 3 | Axonal degeneration of sensory neurons occurs prior to loss of sensory neurons in *UCHL1* knockout flies. a** Confocal fluorescence images for TUNEL assays of tarsal segments 3, 4, and 5 at the front legs of flies expressing *OK371 > nlsGFP*. Respective images were obtained from one of the left or right front legs. Red, TUNEL signal. Green, the nucleus of sensory neuron. Scale bar, 20 μm. **b** Left, confocal fluorescence images of tarsal segments 3, 4, and 5 at the front legs of 30-day-old flies expressing *OK371 > nlsGFP*. The number in panels indicates the number of green signals in each image. Respective images were obtained from one of the left or right front legs. Green, the nucleus of sensory neuron. Scale bar, 20 μm. Right, the numbers of green signals at the tarsal segments 3, 4, and 5 of the front legs of 30-day-old flies expressing *OK371 > nlsGFP*. *n* = 10. **c** Left, cumulative percentage of 30-day-old flies showing escape responses on the 42 °C hot plates within 10 s. *n* = 100. Right, the AUC of the left graph in the same panel. *n* = 10. **d** Schematic diagram of *Drosophila* tarsal segment 4 at the front leg showing sensory somas and

axons. The illustration was created using Biorender.com. **e** Confocal fluorescence images for axons and somas of sensory neurons of tarsal segments 4 at the front legs of flies expressing *OK371 > GFP* at 5, 10, 15, 20, and 25 days of age. Respective images were obtained from one of the left or right front legs. Green, sensory neuron. White or red triangles indicate intact or impaired axons, respectively. Scale bar, 10 μm. **f** The number of sensory neurons at tarsal segments 4 at the front legs of flies expressing *OK371 > GFP* at 5, 10, 15, 20, and 25 days of age. *n* = 10. **g** The number of intact axons at tarsal segments 4 at the front legs of flies expressing *OK371 > GFP* at 5, 10, 15, 20, and 25 days of age. *n* = 10. **h** Cumulative percentage of 5-, 10-, 15-, 20-, and 25-day-old flies showing escape responses on the 42 °C hot plates within 10 s. *n* = 100. **i** The AUC of the graph in (h). *n* = 10. Data are presented as mean ± SD. Two-way ANOVA with Sidak's multiple comparison test was used (**b**, right graph of **c**, **f**, **g**, and **i**). ****$p$ < 0.0001. ns no significant.

segments 3, 4, and 5 of the front legs (Supplementary Fig. 4c). Since the expression of GFP or nlsGFP was not induced at 18 °C, the data for 18 °C control flies could not be obtained. Moreover, we observed delayed escape responses to heat stimuli upon conditional knockdown of *UCH* (Supplementary Fig. 4d). To further prove the absence of developmental defects, we conducted observations on the morphology of *UCH^KO^* and *UCH^C93S^* larvae at 24, 48, 96, 132, and 180 h after egg laying (AEL). Our findings revealed that *UCH* knockout or *UCH* C93S knockin larvae did not show any developmental defects at these time points (Supplementary Fig. 4e). Furthermore, we measured the percentage of pupariation and found no differences between control flies and *UCH* mutants (Supplementary Fig. 4f). Based on these findings, we concluded that the diabetes-like phenotypes of *UCH* mutants are not due to developmental impairments.

In addition, we attempted to measure behavioral changes, including general activity or sleep patterns, which can indirectly affect metabolism and hemolymph glucose levels. During the light-dark (LD) cycle, *w1118*, *UCH^KO^* and *UCH^C93S^* flies showed a typical bimodal pattern of locomotor activity, indicating anticipation of the light-on and light-off periods (Supplementary Fig. 4g). Additionally, we measured sleep time of flies with *UCH* mutations and found that the pattern and duration of sleep in *UCH^KO^* or *UCH^C93S^* flies were similar to those in *w1118* flies (Supplementary Fig. 4h). Therefore, we concluded that the diabetes-like defects observed in *UCH* mutants are not caused by changes in activity or sleep.

To confirm the effect of *UCH* mutations, we generated trans-heterozygous flies which carried one copy of *UCH^KO^* and one copy of *UCH^C93S^*, and observed phenotypes resembling those of T2D and DSN. Our findings showed that the *UCH^KO^*/*UCH^C93S^* flies had higher levels of hemolymph glucose compared to those of +/+, *UCH^KO^*/+, and *UCH^C93S^*/+ flies (Supplementary Fig. 4i). Also, the trans-heterozygous mutations of *UCH* led to a reduction in the number of intact sensory axons at tarsal segments 4 of the front legs (Supplementary Fig. 4j) and the number of sensory nuclei at tarsal segments 3, 4, and 5 of the front legs (Supplementary Fig. 4k). Moreover, we observed decreased escape responses to heat stimuli in the flies carrying one copy of *UCH^KO^* and one copy of *UCH^C93S^* (Supplementary Fig. 4l). In conclusion, we have identified that possible developmental or behavioral defects which can potentially affect the diabetes-like phenotypes of *UCH* mutant flies are not developed by the disruption of the *UCH* gene.

Moreover, we investigated the potential impact of apoptosis in sensory neurons on feeding behaviors and hemolymph glucose levels. Using *OK371*-GAL4, we knocked down *UCH* specifically in sensory neurons, which resulted in apoptotic signals in the sensory neurons of tarsal segments 3, 4, and 5 in the front legs (Supplementary Fig. 3a), but caused no changes in feeding behavior (Supplementary Fig. 3b). Additionally, we explored whether blocking sensory neuronal apoptosis could alter feeding behaviors. Our findings had revealed that sensory neuron-specific knockdown of *Drice* completely prevented sensory neuronal apoptosis in *UCH* knockout flies (Fig. 3a). However,

the increased food intake observed in *UCH^KO^* flies was not affected by the sensory neuron-specific knockdown of *Drice* (Supplementary Fig. 3c). Similarly, *Drice* knockdown in sensory neurons had no effect on the increased hemolymph glucose levels observed in *UCH^KO^* flies (Supplementary Fig. 3d). Therefore, it appeared that apoptosis in sensory neurons had no significant effect on feeding behaviors or hemolymph glucose levels in *Drosophila*.

## UCHL1 regulates insulin signaling by deubiquitinating IRS1

Our previous research elucidated the glycolytic function of *UCH* by suggesting pyruvate kinase (PKM) as a direct target of UCH[19]. Therefore, we have examined whether PKM can regulate the T2D-related phenotypes of *UCH^KO^* flies. We found that overexpressing *PKM* in neurons using *elav*-GAL4 did not reduce the elevated hemolymph glucose levels of *UCH* mutant flies (Supplementary Fig. 5a). Similarly, overexpression of *PKM* in sensory neurons using *OK371*-GAL4 did not rescue the axonal degeneration of leg sensory neurons (Supplementary Fig. 5b), the loss of leg sensory neuron nuclei (Supplementary Fig. 5c), and the decreased responses to heat stimulation (Supplementary Fig. 5d) in *UCH^KO^* flies. Therefore, we concluded that the glycolytic role of *UCH* is not related to its role in maintaining insulin sensitivity. As a result, we have attempted to identify another target of *UCH* responsible for diabetic phenotypes.

As T2D and its complications are closely associated with insulin signaling, we examined whether UCH interacts with the components of insulin signaling. Hence, we quantified the leg sensory neurons of *UCH^KO^* flies while overexpressing a constitutively active (CA) form of insulin receptor (*IR^CA^*), insulin receptor substrate 1 (*dIRS1; Drosophila IRS1* or *chico* in *Drosophila* name)[32], phosphoinositide 3-kinase with a farnesylation signal (*PI3K^CAAX^*), and Akt with myristoylation-palmitoylation motif (*myrAkt*) using *OK371*-GAL4. The numbers of the leg sensory neurons in *UCH^KO^* flies expressing *IR^CA^* were similar to those in *UCH^KO^* flies. Intriguingly, loss of the leg sensory neurons in *UCH^KO^* was rescued by expressing *dIRS1*, *PI3K^CAAX^*, or *myrAkt* (Fig. 4a). Since insulin signals are transmitted into cells sequentially via IR, IRS, PI3K, and Akt, we hypothesized that UCH interacts with dIRS1. Furthermore, the degeneration of leg sensory axons (Fig. 4b), decreased escape responses against heat stimulation (Fig. 4c), elevated glycemia (Supplementary Fig. 5e), and glucose intolerance (Supplementary Fig. 5f) in *UCH* mutant flies were improved by *dIRS1* expression.

To further investigate the correlations between UCHL1 and insulin signaling, we used human embryonic kidney 293E (HEK293E) cell line and generated a *UCHL1* KO HEK293E cell line using CRISPR-Cas9. We found that the phosphorylation of Akt on serine 473 was reduced in *UCHL1* KO cell lines compared to *UCHL1* WT cell lines with or without insulin treatment (Fig. 4d). Surprisingly, the decreased Akt phosphorylation in *UCHL1* KO cells was rescued by expressing exogenous IRS1, either with or without insulin treatment (Fig. 4d), which led us to hypothesize that IRS1 might be a substrate of UCHL1. To address this possibility, we measured ubiquitination of IRS1 upon expressing

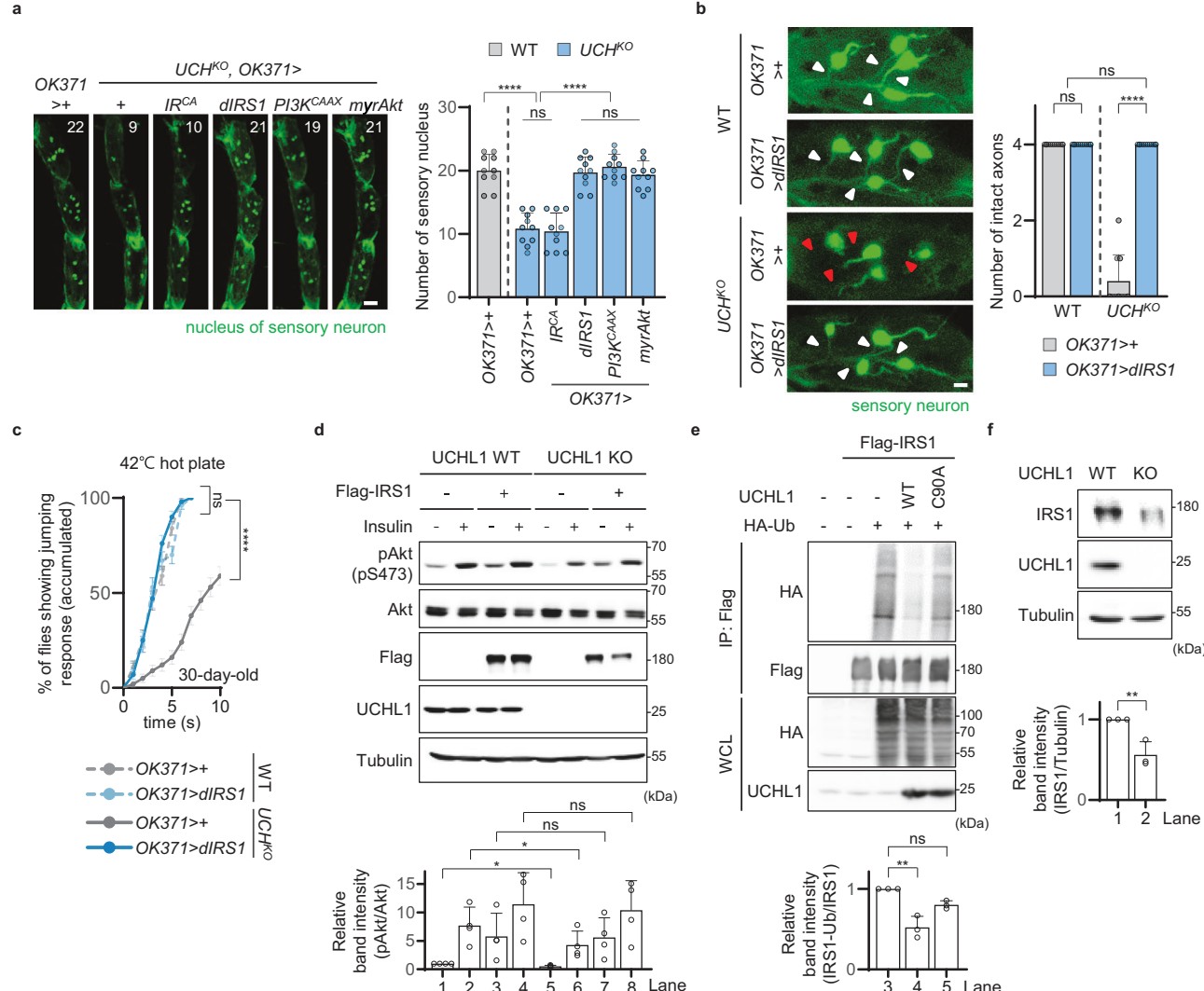

**Fig. 4 | UCHL1 regulates insulin signaling by deubiquitinating IRS1. a** Left, confocal fluorescence images of tarsal segments 3, 4, and 5 at the front legs of 30-day-old flies expressing *OK371 > nlsGFP*. The number in panels indicates the number of green signals in each image. Respective images were obtained from one of the left or right front legs. Green, the nucleus of sensory neuron. Scale bar, 20 μm. Right, the numbers of green signals at the tarsal segments 3, 4, and 5 of the front legs of 30-day-old flies expressing *OK371 > nlsGFP*. *n* = 10. **b** Left, confocal fluorescence images for axons and somas of sensory neurons of tarsal segments 4 at the front legs of 15-day-old flies expressing *OK371 > GFP*. Respective images were obtained from one of the left or right front legs. Green, sensory neuron. White or red triangles indicate intact or impaired axons, respectively. Scale bar, 10 μm. Right, the number of intact axons at tarsal segments 4 at the front legs of 15-day-old flies expressing *OK371 > GFP*. *n* = 10. **c** Cumulative percentage of 30-day-old flies showing escape responses on the 42 °C hot plates within 10 s. *n* = 100. **d** Top, immunoblot analysis of Akt phosphorylation (S473) in HEK293E *UCHL1* WT and KO cells upon insulin treatment. The cells transfected with the empty plasmids or the plasmids carrying Flag-tagged *IRS1* were treated with 50 nM insulin for 1 h. Bottom,

relative quantification of immunoblot band intensity of anti-pAkt (S473) normalized to that of anti-Akt. *n* = 4. **e** Top, immunoblot analysis of IRS1 ubiquitination in HEK293E cells co-expressing *IRS1* and *UCHL1* WT or C90A. The cells were co-transfected with the empty plasmids or the plasmids carrying Flag-tagged *IRS1*, HA-tagged *Ubiquitin*, and *UCHL1* WT or C90A upon 40 μM MG132 treatment for 4 h to all samples. Bottom, relative quantification of anti-HA immunoblot band intensity from anti-Flag immunoprecipitation normalized to anti-Flag immunoblot band intensity from anti-Flag immunoprecipitation. *n* = 3. **f** Top, immunoblot analysis of endogenous IRS1 in *UCHL1* WT and KO HEK293E cells. Bottom, relative quantification of immunoblot band intensity of anti-IRS1 normalized to that of anti-Tubulin. *n* = 3. IP immunoprecipitation. WCL whole cell lysate. Data are presented as mean ± SD. One-way ANOVA with Tukey's multiple comparison test was used (**a** and **e**). Two-way ANOVA with Sidak's multiple comparison test was used (**b**). Mantel–Cox test was used (**c**). Repeated measures (RM) one-way ANOVA with Holm–Sidak's multiple comparisons test was used (**d**). Two-tailed paired Student's *t*-test was used (**f**). *$p < 0.05$, **$p < 0.01$, ****$p < 0.0001$. ns no significant.

*UCHL1*. We discovered that ubiquitination of IRS1 and dIRS1 was diminished by expressing *UCHL1* WT; however, exogenous expression of *UCHL1* C90A, a catalytic dead form of UCHL1[33], did not reduce ubiquitination of IRS1 and dIRS1 (Fig. 4e and Supplementary Fig. 5j). In addition, we tested if UCHL1 could regulate the stability of IRS1 and found that the protein levels of IRS1 were significantly decreased in *UCHL1* KO cell lines (Fig. 4f).

In addition, we conducted further experiments to measure the phosphorylation of Akt on serine 473 by manipulating UCHL1 in

various cell lines, including neuroblastoma (SH-SY5Y), mouse embryonic fibroblasts (MEF), and hepatocellular carcinoma (SNU398). Similar to the results obtained in HEK293E cells, we observed a decrease in Akt phosphorylation in UCHL1 KO SH-SY5Y cells[19] compared to UCHL1 WT SH-SY5Y cells, regardless of insulin treatment (Supplementary Fig. 5g). Moreover, when we knocked down UCHL1 in MEF cell line and examined Akt phosphorylation, we found that MEF cells expressing *UCHL1* siRNA showed reduced levels of Akt phosphorylation compared to MEF cells expressing control siRNA with or

without insulin treatment (Supplementary Fig. 5h). Both control siRNA and *UCHL1* siRNA expressing SNU398 cells showed negligible levels of Akt phosphorylation in the absence of insulin treatment. However, when treated with insulin, SNU398 cells expressing control siRNA displayed higher levels of Akt phosphorylation compared to SNU398 cells expressing *UCHL1* siRNA (Supplementary Fig. 5i).

We also observed that UCHL1 directly interacted with IRS1 via co-immunoprecipitation (IP) assay (Supplementary Fig. 5k). As IRS1 is a large protein comprised of 1,242 amino acids, including a pleckstrin homology (PH) domain and a phosphotyrosine-binding (PTB) domain at its N-terminus, we tried to identify the exact region in which IRS1 interacted with UCHL1. Truncated forms of IRS1 with only the front region of IRS1 (1-866), the middle region (294-866), or PH and PTB domains (PH-PTB) (1-300) were used to perform the experiments (Supplementary Fig. 5l)[34]. We co-expressed UCHL1 and each truncated form of IRS1, including full length (FL), front, middle, and PH-PTB, and performed co-IP. According to the results of the co-IP assay, UCHL1 interacted with the PH-PTB domains of IRS1 (Supplementary Fig. 5m). We then used ColabFold to predict the interactions of PH-PTB domains of IRS1 and UCHL1[35]. Expectedly, ColabFold indicated that UCHL1 interacted with the PH-PTB domains of IRS1 with reliable score (Supplementary Fig. 5n). Therefore, we concluded that UCHL1 interacts with IRS1 to deubiquitinate and stabilize IRS1, and this interaction positively regulates insulin signaling in human cell lines and fruit flies.

Since IRS1 is also a well-known component of insulin-like growth factors (IGF) signaling which is responsible for cell growth or development, we attempted to measure the brain size of the flies with *UCH* mutations. We found that the brain size of *UCH^KO* and *UCH^C93S* flies was comparable to that of control flies (Supplementary Fig. 6a). To further support this result, we generated the flies with *dIRS1* knockdown using *elav-* and *tub-*GAL4, respectively. Consistent with previous studies[36], we found that the flies with *dIRS1* knockdown throughout the body had smaller brains than control flies. However, to our surprise, *dIRS1* knockdown using *elav-*GAL4 did not affect brain size (Supplementary Fig. 6b). Taken together, these results suggested that the brain size of fruit flies is not solely determined by IGF signaling in neurons, but rather regulated by the systemic effects of various DILPs and IGF signaling. Therefore, *UCH* deficiency, as with *dIRS1* knockdown in neurons, might have no effect on the regulation of brain size in *Drosophila*.

Furthermore, we investigated whether inducing insulin resistance solely in neurons can lead to an increase in hemolymph glucose levels, as it is well known that the main organs responsible for glucose uptake are the muscle, fat, and liver. We, thus, generated flies with tissue-specific knockdown of *IRS1* to induce insulin resistance in neurons, muscles, and fat bodies using *elav-* or *nSyb-*GAL4, *mef2-*GAL4, and *cg-*GAL4, respectively. As expected, we observed increased hemolymph glucose levels in *IRS1* knockdown flies in muscles and fat bodies (Supplementary Fig. 6c). Surprisingly, we also found elevated glucose levels in the hemolymph of flies with neuron-specific knockdown of *IRS1* using *elav-*GAL4 or *nSyb-*GAL4 (Supplementary Fig. 6c). These results led us to conclude that neuronal insulin resistance can contribute to hyperglycemia in *Drosophila*. Furthermore, a previous study has shown that mice with neuron-specific disruption of insulin receptor display diabetic phenotypes, such as obesity and hypertriglyceridemia[37]. Nonetheless, additional investigations are required to verify whether neuronal insulin resistance can induce hyperglycemia in vertebrates and clinical settings.

## E3 ligase CUL1 antagonizes the role of UCHL1

Next, we attempted to identify the E3 ligase of IRS1 which might counteract UCHL1. So far, it has been known that mitsugumin-53 (MG53), mouse double minute 2 (MDM2), casitas B-lineage lymphoma (Cbl), Cullin 1 (CUL1), Cullin 3 (CUL3), Cullin 5 (CUL5), Cullin 7 (CUL7), and tumor necrosis factor (TNF) receptor-associated factor 4 (TRAF4) ubiquitinate IRS1[10–12,38–40]. Among them, Cbl, CUL1, CUL3, CUL5, and

TRAF4 are conserved in fruit flies. Therefore, we performed a small-scale screen using RNAi lines against these 5 genes. We expressed each RNAi line in *UCH^KO* flies under the *OK371-*GAL4 driver and measured the number of leg sensory neurons. Although the number of leg sensory neurons in *UCH* knockout flies was similar to that in *UCH* knockout flies with sensory neuron-specific *Cbl*, *CUL3*, *CUL5*, or *TRAF4* knockdown, knocking down *CUL1* in sensory neurons rescued the loss of leg sensory neurons observed in *UCH* knockout flies (Fig. 5a). Furthermore, the axonal defects in sensory neurons and decreased escape responses to heat stimulation of *UCH^KO* were improved by *CUL1* knockdown (Fig. 5b, c). The increased hemolymph glucose levels of *UCH* mutant flies were also rescued by neuron-specific *CUL1* knockdown (Supplementary Fig. 6d). Therefore, we hypothesized that UCHL1 and CUL1 have antagonistic roles on IRS1 ubiquitination.

To prove this hypothesis, we explored the interaction between UCHL1 and CUL1 in HEK293E cells. Since CUL1 is a crucial component of the S-phase kinase-associated protein (Skp)-Cullin-F-box-containing (SCF) E3 ligase complex and is fully activated by neddylation, we overexpressed CUL1, RING-box protein 1 (RBX1), and NEDD8 to induce CUL1 activation and observed the ubiquitination of IRS1. We found that expressing CUL1 WT increased the ubiquitination of human IRS1 and dIRS1, but expressing CUL1 dominant negative (DN), which has no ability to form the SCF complex due to a C-terminal truncation[41], did not increase the ubiquitination of human IRS1 or dIRS1 (Supplementary Fig. 7a, b). Next, we examined if the increased ubiquitination of IRS1 mediated by CUL1 could be reduced by co-expression of UCHL1. The elevated ubiquitination of human IRS1 and dIRS1 induced by CUL1, RBX1, and NEDD8 expression was subsequently decreased by expressing exogenous UCHL1 WT, but not UCHL1 C90A (Fig. 5d and Supplementary Fig. 7c). Furthermore, the phosphorylation of Akt was increased by *CUL1* knockdown and the decreased Akt phosphorylation in *UCHL1* KO cells was rescued by the expression of *CUL1* siRNA (Fig. 5e). Taken together, we concluded that UCHL1 and CUL1 play antagonistic roles in IRS1 ubiquitination by sharing the same sites of action. These results led us to identify the exact sites at which CUL1 ubiquitinates IRS1.

To determine the lysine residues of IRS1 subjected to ubiquitination by CUL1, we aligned protein sequences between human IRS1 and dIRS1. We observed that the lysine residues at positions 21, 23, 27, 51, 52, 61, 79, 171, 190, 523, 538, 759, 867, and 943 in human IRS1 exhibit conservation in dIRS1 using Clustal Omega. Accordingly, we tried to narrow down which of these lysine residues are CUL1-mediated ubiquitination sites. We generated various mutated forms of human IRS1 and dIRS1 of which the lysine sites were changed into arginine (KR). We co-expressed CUL1, RBX1, NEDD8, and IRS1 with various combinations of KR mutation and observed IRS1 ubiquitination. We found that human IRS1 with 523, 538, 759, 867, and 943 lysine residues substituted into arginine (5KR) was not significantly ubiquitinated by CUL1 (Supplementary Figs. 5f and 7e). Likewise, dIRS1 with lysines 538, 551/552, 754, 868, and 950, corresponding to the 523, 538, 759, 867, and 943 lysine residues of human IRS1 (Supplementary Fig. 7d), mutated into arginine (6KR) was not significantly ubiquitinated by CUL1 (Supplementary Fig. 7f, g). These results suggested that these five lysine residues in human IRS1 are majorly ubiquitinated by CUL1.

To investigate further the interaction between UCHL1 and CUL1 in *Drosophila*, we first examined if *CUL1* knockdown could rescue the T2D- or DSN-like phenotypes induced by HSD. Similar to the results of *UCH* overexpression, whole-body knockdown of *CUL1* using *hs-*GAL4 driver rescued the hyperglycemia induced by HSD (Supplementary Fig. 6d). In addition, the axonal degeneration of sensory neurons, decreased number of leg sensory neurons, and the reduced escape responses against heat stimuli developed by HSD were improved by sensory neuron-specific *CUL1* knockdown (Supplementary Fig. 6f–h). Next, we observed whether *CUL1* overexpression could induce T2D- or DSN-related symptoms. Here, whole-body expression of *CUL1* resulted

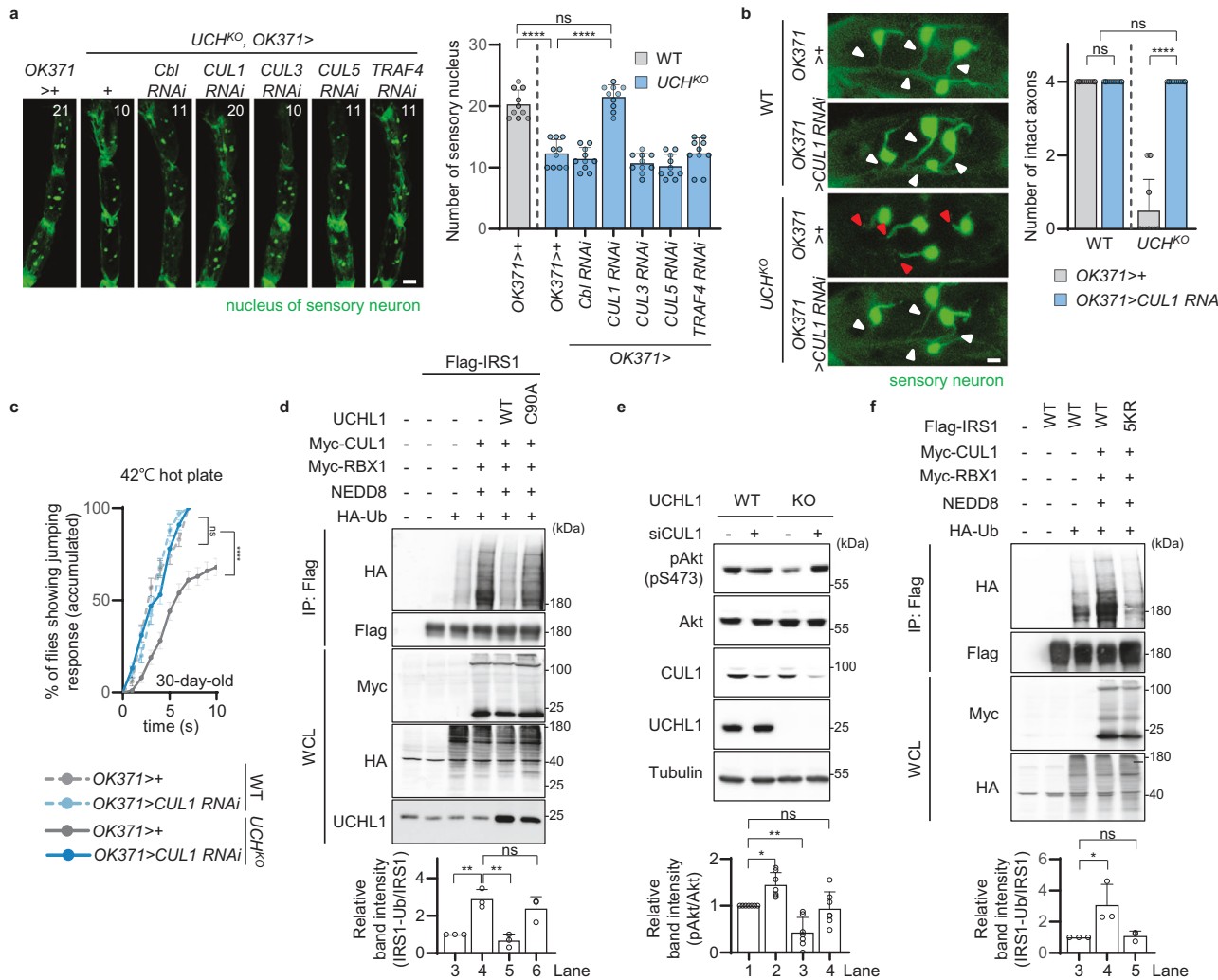

**Fig. 5 | Ubiquitination of IRS1 mediated by CUL1 is antagonized by UCHL1.**
**a** Left, confocal fluorescence images of tarsal segments 3, 4, and 5 at the front legs of 30-day-old flies expressing *OK371 > nlsGFP*. The number in panels indicates the number of green signals in each image. Respective images were obtained from one of the left or right front legs. Green, the nucleus of sensory neuron. Scale bar, 20 μm. Right, the numbers of green signals at the tarsal segments 3, 4, and 5 of the front legs of 30-day-old flies expressing *OK371 > nlsGFP*. *n* = 10. **b** Left, confocal fluorescence images for axons and somas of sensory neurons of tarsal segments 4 at the front legs of 15-day-old flies expressing *OK371 > GFP*. Respective images were obtained from one of the left or right front legs. Green, sensory neuron. White or red triangles indicate intact or impaired axons, respectively. Scale bar, 10 μm. Right, the number of intact axons at tarsal segments 4 at the front legs of 15-day-old flies expressing *OK371 > GFP*. *n* = 10. **c** Cumulative percentage of 30-day-old flies showing escape responses on the 42 °C hot plates within 10 s. *n* = 100. **d** Top, immunoblot analysis of IRS1 ubiquitination in HEK293E cells co-expressing *IRS1*, *CUL1*, *RBX1*, *NEDD8*, and *UCHL1* WT or C90A. The cells were co-transfected with the empty plasmids or the plasmids carrying Flag-tagged *IRS1*, Myc-tagged *CUL1*, Myc-tagged *RBX1*, *NEDD8*, HA-tagged *Ubiquitin*, and *UCHL1* WT or C90A upon 40 μM MG132

treatment for 4 h to all samples. Bottom, relative quantification of anti-HA immunoblot band intensity from anti-Flag immunoprecipitation normalized to anti-Flag immunoblot band intensity from anti-Flag immunoprecipitation. *n* = 3. **e** Top, immunoblot analysis of Akt phosphorylation (pS473) in HEK293E *UCHL1* WT and KO cells upon control siRNA or *CUL1* siRNA (siCUL1) expression. Bottom, relative quantification of immunoblot band intensity of anti-pAkt (pS473) divided by that of anti-Akt. *n* = 6. **f** Top, immunoblot analysis of IRS1 ubiquitination in HEK293E cells co-expressing *IRS1* WT or 5KR (K523, 538, 759, 867, and 943R), *CUL1*, *RBX1*, and *NEDD8*. The cells were co-transfected with the empty plasmids or the plasmids carrying Flag-tagged *IRS1* WT or 5KR, Myc-tagged *CUL1*, Myc-tagged *RBX1*, *NEDD8*, and HA-tagged *Ubiquitin* upon 40 μM MG132 treatment for 4 h to all samples. Bottom, relative quantification of anti-HA immunoblot band intensity from anti-Flag immunoprecipitation normalized to anti-Flag immunoblot band intensity from anti-Flag immunoprecipitation. *n* = 3. IP, immunoprecipitation. WCL, whole cell lysate. Data are presented as mean ± SD. One-way ANOVA with Tukey's multiple comparison test was used (**a**, **d**, **e**, and **f**). Two-way ANOVA with Sidak's multiple comparison test was used (**b**). Mantel−Cox test was used (**c**). *$p < 0.05$. **$p < 0.01$. ****$p < 0.0001$. ns no significant.

in increased hemolymph glucose levels in fruit flies (Supplementary Fig. 6i). Additionally, loss of sensory axons and neurons and defective escape responses to noxious heat were induced by overexpressing *CUL1* under *OK371*-GAL4 driver (Fig. 6a and Supplementary Fig. 6j, k). These T2D- and DSN-associated pathologies caused by *CUL1* overexpression were completely rescued by *UCH* overexpression (Fig. 6a and Supplementary Fig. 6i–k). In conclusion, our findings confirmed that UCHL1 antagonistically interacts with CUL1 in both mammalian cells and *Drosophila*.

Moreover, we conducted a study on the genetic interaction between *CUL1* and *IRS1*. Our findings indicated that increased hemolymph glucose levels in the flies overexpressing *CUL1* using a whole-body driver, *hs-GAL4*, were rescued with *IRS1* co-overexpression (Supplementary Fig. 6l). The axonal defects in sensory neurons in the flies with overexpressing *CUL1* in sensory neurons were also alleviated by co-expressing *IRS1* (Supplementary Fig. 6m). Furthermore, the loss of sensory neuron nuclei at tarsal segments 3, 4, and 5 of the front legs (Supplementary Fig. 6n) as well as the decrease in escape

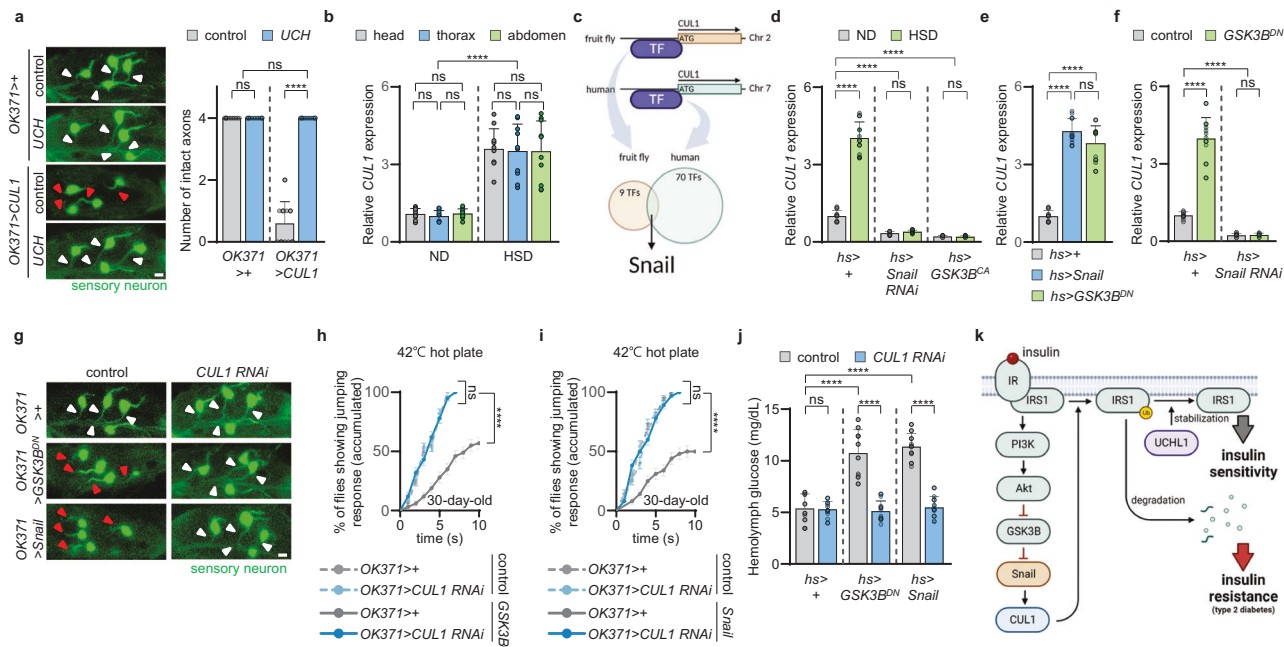

**Fig. 6 | CUL1 expression increases in response to HSD through GSK3B-Snail axis. a** Left, confocal fluorescence images for axons and somas of sensory neurons of tarsal segments 4 at the front legs of 15-day-old flies expressing *OK371 > GFP*. Respective images were obtained from one of the left or right front legs. Green, sensory neuron. White or red triangles indicate intact or impaired axons, respectively. Scale bar, 10–μm. Right, the number of intact axons at tarsal segments 4 at the front legs of 15-day-old flies expressing *OK371 > GFP*. *n* = 10. **b** Relative expressions of *CUL1* from the head, thorax, and abdomen of *w1118* upon ND or HSD. Normalized to *rp49* expressions in the same body part upon ND or HSD. *n* = 10. **c** Schematic diagram to identify a transcription factor for *CUL1* using *JASPAR*. The illustration was created using Biorender.com. **d** Relative expressions of *CUL1* normalized to *rp49* expressions from the flies upon ND or HSD. *n* = 10. **e** Relative expressions of *CUL1* normalized to *rp49* expressions from the flies. *n* = 10. **f** Relative expressions of *CUL1* normalized to *rp49* expressions from the flies. *n* = 10. **g** Confocal fluorescence images for axons and somas of sensory neurons of tarsal

segments 4 at the front legs of 15-day-old flies expressing *OK371 > GFP*. Respective images were obtained from one of the left or right front legs. Green, sensory neuron. White or red triangles indicate intact or impaired axons, respectively. Scale bar, 10 μm. The number of intact axons at tarsal segments 4 at the front legs of 15-day-old flies expressing *OK371 > GFP* was measured and the related statistical analysis was performed in Supplementary Fig. 8k. **h** Cumulative percentage of 30-day-old flies showing escape responses on the 42 °C hot plates within 10 s. *n* = 100. **i** Cumulative percentage of 30-day-old flies showing escape responses on the 42 °C hot plates within 10 s. *n* = 100. **j** Glucose concentrations in the hemolymph of 3-day-old flies. *n* = 10. **k** Illustration summarizing the main findings of this study. The illustration was created using Biorender.com. ND normal diet. HSD high-sucrose diet. Data are presented as mean ± SD. Two-way ANOVA with Sidak's multiple comparison test was used (**a**, **b**, **d**, **f**, and **j**). One-way ANOVA with Tukey's multiple comparison test was used (**e**). Mantel–Cox test was used (**h** and **i**). ****p < 0.0001. ns no significant.

responses to heat stimulation (Supplementary Fig. 6o) observed in the flies with sensory neuron-specific *CUL1* overexpression were both alleviated by co-expression of *IRS1*. For further investigation, we tested whether knocking down other known substrates of CUL1, such as *cyclin D* (*CCND*), *cyclin E* (*CCNE*), *cyclin-dependent kinase 4* (*CDK4*), and *wee1 G2 checkpoint kinase* (*WEE1*) could increase hemolymph glucose levels. However, knockdown of *CCND*, *CCNE*, *CDK4*, and *WEE1* using *hs*-GAL4 did not increase hemolymph glucose levels. Only flies with *IRS1* knockdown exhibited elevated levels of hemolymph glucose (Supplementary Fig. 6p). Based on these findings, we concluded that the diabetic effects of CUL1 are primarily mediated by IRS1.

## GSK3B-Snail axis is responsible for the increase of *CUL1* gene expression in response to HSD

Given our demonstration that exogenous expression of *UCH* ameliorated the T2D- and DSN-like phenotypes induced by HSD or *CUL1* overexpression in *Drosophila*, we sought to explore the correlations between *CUL1* and HSD. Thus, we measured the gene expression of *CUL1* from flies fed either normal diet (ND) or HSD. Although *UCH* was highly expressed in fly heads and neurons, *CUL1* expression did not show any tissue-specific patterns (Fig. 6b). Surprisingly, *CUL1* gene expression in fruit flies was elevated by about 4-fold in response to HSD (Fig. 6b). In contrast, neuron-specific expression of *UCH* was not altered by HSD in *Drosophila* (Supplementary Fig. 8a). Additionally, we measured the levels of *CUL1* transcripts in GFP+ nuclei extracted

through FANS method from the entire bodies or the front legs of *hs > lamin-GFP* or *OK371 > lamin-GFP* flies. Unlike the expression patterns of *UCH*, the expression of *CUL1* was similar between the nuclei from *hs > lamin-GFP* and *OK371 > lamin-GFP* flies for both from the whole body and front legs. As expected, *CUL1* gene expressions from various nuclei samples did not display tissue specificity and were equally elevated by HSD (Supplementary Fig. 8b).

Next, we tried to determine transcription factor (TF) that could promote *CUL1* transcription in response to HSD. We used the *JASPAR* database to specify TF candidates of *CUL1*[42]. Human and *Drosophila* loci around the translation start site [−350 to +150 base pair (bp)] of the *CUL1* gene were analyzed through *JASPAR*. We narrowed down 9 TFs, including Snail, and 70 TFs, including Snail1, 2, and 3, that could regulate the transcription of *CUL1* in *Drosophila* and human, respectively (Fig. 6c). Since human *Snail1, 2,* and *3* are conserved in *Drosophila* as *Snail* (*cg3956*), we hypothesized that Snail might regulate *CUL1* transcription in flies. As expected, whole-body knockdown of *Snail* in fruit flies reduced *CUL1* gene expression (Fig. 6d). Similarly, *CUL1* gene expression was significantly decreased by knocking down *Snail1, 2,* and *3* in HEK293E cells (Supplementary Fig. 8c). Surprisingly, HSD-mediated increases of *CUL1* gene expression were not observed when *Snail* was knocked down using a whole-body GAL4 driver (*hs*-GAL4) in fruit flies (Fig. 6d). Furthermore, we found that overexpression of *Snail* using *hs*-GAL4 upregulated *CUL1* expression (Fig. 6e). Since Snail functions as a transcriptional repressor of

*E-cadherin* (*E-cad*) and a transcriptional activator of *ras homolog family member B* (*RhoB*) and *matrix metallopeptidase 1* (*MMP1*)[43–45], we also measured the transcription levels of *shotgun* (*cg3722*), *Rho1* (*cg8416*), and *MMP1* (*cg4859*), which are homologous to human *E-cad*, *RhoB*, and *MMP1*, respectively. The expression of *shotgun* was downregulated, and the expressions of *Rho1* and *MMP1* were upregulated by HSD (Supplementary Fig. 8d). In addition, we found that the knockdown of *Snail* did not change the expression of *UCH* (Supplementary Fig. 8e). Altogether, we concluded that Snail regulates *CUL1* expression in response to HSD.

It has been well established that Snail is phosphorylated by glycogen synthase kinase-3 beta (GSK3B), leading to the degradation of Snail, in mammalian cells[46]. Therefore, we examined whether the GSK3B-Snail axis is also conserved in fruit flies and could regulate *CUL1* gene expression in response to HSD. First, we tested if the genetic manipulations of *Drosophila GSK3B* (*shaggy*; *cg2621*) could regulate *CUL1* expression. Overexpression of a dominant negative form of GSK3B (GSK3B^DN; shaggy A81T)[47] upregulated *CUL1* expression (Fig. 6e) and a constitutively active form of GSK3B (GSK3B^CA; shaggy S9A)[48] downregulated *CUL1* expression in *Drosophila* (Fig. 6d). Similarly, we found that overexpression of a constitutively active form of human GSK3B (GSK3B^CA; S9A)[49] downregulated *CUL1* transcription, and the dominant negative form of human GSK3B (GSK3B^DN; K85A)[50] upregulated *CUL1* gene expression in HEK293E cells (Supplementary Fig. 8f). In addition, the decreased expression of *CUL1* induced by *GSK3B^CA* overexpression was not changed by HSD (Fig. 6d), suggesting that HSD controls *CUL1* expression via GSK3B. Second, we attempted to observe whether GSK3B functions as an upstream regulator of Snail by investigating genetic interaction between *GSK3B* and *Snail* in *Drosophila*. The increased *CUL1* gene expression induced by expressing GSK3B^DN was almost completely blocked in the flies with subsequent *Snail* knockdown (Fig. 6f). Therefore, we concluded that a conserved GSK3B-Snail pathway exists in *Drosophila* and controls *CUL1* gene expression in response to HSD.

As GSK3B is phosphorylated by activated Akt[51], we sought to measure *CUL1* gene expression from flies with genetically activated Akt by expressing *IR^CA*, *dIRS1*, *PI3K^CAAX*, and *myrAkt*. However, flies showed lethality upon overexpression of *IR^CA*, *PI3K^CAAX*, or *myrAkt* using *hs*-GAL4 driver. We, thus, generated flies that conditionally express *IR^CA*, *dIRS1*, *PI3K^CAAX*, and *myrAkt* in the entire body using *hs*-GAL4 and *tub*-GAL80^TS. Here, we observed that *CUL1* transcription was upregulated by transiently expressing *IR^CA*, *dIRS1*, *PI3K^CAAX*, and *myrAkt* for 12 h (Supplementary Fig. 8g).

For further investigation, we attempted to observe the genetic interaction between one of the *DILP*s, *DILP2*, and *Akt* in terms of *CUL1* expression under HSD. However, as genetic intervention of *DILP2* or *Akt* was lethal to fruit flies, we regulated these genes using *tub*-GAL80^TS. Firstly, flies with *DILP2* knockdown for 24 h under ND exhibited reduced levels of *CUL1* expression compared to control flies under ND. Interestingly, the expression of *CUL1* in the flies with 24-h *DILP2* knockdown combined with simultaneous 24-h HSD was similar to that of the flies with 24-h *DILP2* knockdown under ND (Supplementary Fig. 8h). On the contrary, flies with myrAkt overexpression for 24 h under ND had elevated *CUL1* expression compared to control flies under ND. Furthermore, 24-h *myrAkt* expressing flies under simultaneous 24-h HSD exhibited similar levels of *CUL1* expression compared to 24-h *myrAkt* overexpressing flies under ND (Supplementary Fig. 8h). To our surprise, when both *DILP2* RNAi and *myrAkt* were expressed for 24 h under ND, the expression of *CUL1* remained increased, similar to the levels observed in the flies with *myrAkt* expression for 24 h under ND (Supplementary Fig. 8h). Also, *CUL1* expression in the flies with *DILP2* knockdown and *myrAkt* expression for 24 h under simultaneous 24-h HSD was comparable to the levels observed in the flies with *myrAkt* expression for 24 h under ND (Supplementary Fig. 8h). These findings led us to conclude that HSD elevated *CUL1* expression via

DILP2-Akt axis. Similarly, we tested the genetic interaction between *Akt* and *GSK3B* with respect to *CUL1* expression under HSD. We observed that *CUL1* expression decreased when *Akt* was knocked down for 24 h under ND. Also, flies with 24-h *Akt* knockdown under simultaneous 24-h HSD displayed similar levels of *CUL1* expression compared to the flies with 24-h *Akt* knockdown under ND. Conversely, *CUL1* expression increased when *GSK3B^DN* was overexpressed for 24 h even under ND compared to control flies under ND. Furthermore, *CUL1* expression in the flies with 24-h *GSK3B^DN* overexpression under simultaneous 24-h HSD was similar to that in the flies with 24-h *GSK3B^DN* overexpression under ND (Supplementary Fig. 8i). When *Akt* RNAi and *GSK3B^DN* were expressed for 24 h under ND, *CUL1* expression remained increased, similar to the levels observed in the flies with *GSK3B^DN* expressed for 24 h under ND (Supplementary Fig. 8i). Also, *CUL1* expression in the flies with *Akt* knockdown and *GSK3B^DN* expression for 24 h under simultaneous 24-h HSD was comparable to the levels observed in the flies expressing *GSK3B^DN* for 24 h under ND (Supplementary Fig. 8i). Based on these findings, we concluded that HSD elevates *CUL1* expression through a sequential mechanism involving DILP2, Akt, and GSK3B. These results suggested that elevated expression of *CUL1* in response to HSD was a result of the Akt-GSK3B axis.

When fruit flies are exposed to excessive amounts of nutrients through HSD, it stimulates the secretion of DILPs. As a result, insulin signaling is activated in cells, which leads to the activation of Akt and the inhibition of GSK3B through sequential phosphorylation. This, in turn, stabilizes Snail, which upregulates the expression of *CUL1*. Subsequently, increased CUL1 proteins ubiquitinate and destabilize IRS1 to turn off insulin signaling. However, when overnutrition persists owing to continuous exposure to HSD, CUL1 proteins accumulate in cells and eliminate IRS1 proteins, leading to insulin resistance. Based on our findings, we suggest that increased insulin signaling promotes the expression of *CUL1*, resulting in the breakdown of IRS1 to turn the signaling off temporarily; however, prolonged overnutrition disrupts IRS1 consistently, leading to the development of insulin resistance.

Furthermore, we investigated the expression levels of *CUL1* in the brains of *UCH^KO* flies under ND and HSD conditions. We observed a decrease in *CUL1* expression in the brains of *UCH^KO* flies compared to control flies under ND. Additionally, while the expression of *CUL1* increased in the brains of *w1118* flies under HSD, it did not have the same effect on *CUL1* expression in the brains of *UCH^KO* flies (Supplementary Fig. 8j). Given that the *UCH^KO* flies exhibited insulin resistance in neurons, it is reasonable to conclude that the decreased expression of *CUL1* in *UCH^KO* brains was attributed to impaired insulin signaling under ND. Also, it was expected that the expression of *CUL1* would not be increased in the brains of *UCH^KO* flies in response to HSD.

## GSK3B and snail govern T2D- and DSN-related phenotypes induced by HSD through CUL1

Considering our results that GSK3B and Snail controlled the transcription of *CUL1*, we tested if the genetic manipulations of *GSK3B* and *Snail* could regulate T2D- and DSN-like defects in *Drosophila*. Sensory neuron-specific expression of GSK3B^DN or Snail induced the axonal degeneration and death of leg sensory neurons (Fig. 6g and Supplementary Fig. 8k, l) and reduced the jumping responses to heat stimuli (Fig. 6h, i) in fruit flies. Furthermore, the whole-body expression of GSK3B^DN or Snail elevated hemolymph glucose levels (Fig. 6j). Interestingly, these defects were almost completely rescued by knocking down *CUL1* (Fig. 6g–j and Supplementary Fig. 8k, l). Similarly, *UCH* overexpression also improved the hyperglycemia and the DSN-related phenotypes induced by overexpression of GSK3B^DN or Snail (Supplementary Fig. 9a–d). Next, we tried to investigate if the HSD-induced diabetic phenotypes could be improved by expressing *GSK3B^CA* or knocking down *Snail*. The axonal degeneration and loss of leg sensory neurons, reduced escape responses to heat stimuli, and hyperglycemia induced by HSD were all alleviated by the expression of *GSK3B^CA* or

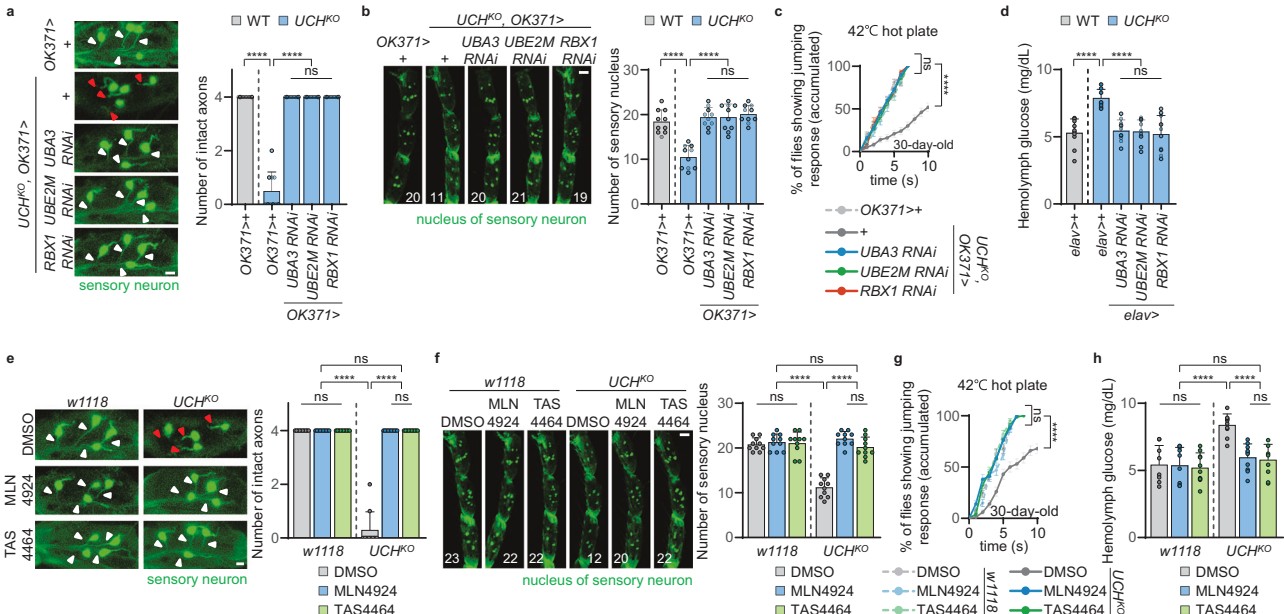

**Fig. 7 | T2D- and DSN-related phenotypes are improved by genetic and pharmacological suppression of neddylation. a** Left, confocal fluorescence images for axons and somas of sensory neurons of tarsal segments 4 at the front legs of 15-day-old flies expressing *OK371 > GFP*. Respective images were obtained from one of the left or right front legs. Green, sensory neuron. White or red triangles indicate intact or impaired axons, respectively. Scale bar, 10 μm. Right, the number of intact axons at tarsal segments 4 at the front legs of 15-day-old flies expressing *OK371 > GFP*. *n* = 10. **b** Left, confocal fluorescence images of tarsal segments 3, 4, and 5 at the front legs of 30-day-old flies expressing *OK371 > nlsGFP*. The number in panels indicates the number of green signals in each image. Respective images were obtained from one of the left or right front legs. Green, nucleus of sensory neuron. Scale bar, 20 μm. Right, the numbers of green signals at the tarsal segments 3, 4, and 5 of the front legs of 30-day-old flies expressing *OK371 > nlsGFP*. *n* = 10. **c** Cumulative percentage of 30-day-old flies showing escape responses on the 42 °C hot plates within 10 s. *n* = 100. **d** Glucose concentrations in the hemolymph of 3-day-old flies. *n* = 10. **e** Left, confocal fluorescence images for axons and somas of sensory neurons of tarsal segments 4 at the front legs of 15-day-old flies expressing *OK371 > GFP*. The flies were fed 0.05% DMSO, 20 μM MLN4924, or 20 μM TAS4464 continuously after eclosion. Respective images were obtained from one of the left or right front legs. Green, sensory neuron. White or red triangles indicate intact or impaired axons, respectively. Scale bar, 10 μm. **f** Left, confocal fluorescence images of tarsal segments 3, 4, and 5 at the front legs of 30-day-old flies expressing *OK371 > nlsGFP*. The flies were fed 0.05% DMSO, 20 μM MLN4924, or 20 μM TAS4464 continuously after eclosion. The number in panels indicates the number of green signals in each image. Respective images were obtained from one of the left or right front legs. Green, the nucleus of sensory neurons. Scale bar, 20 μm. Right, the numbers of green signals at the tarsal segments 3, 4, and 5 of the front legs of 30-day-old flies expressing *OK371 > nlsGFP*. *n* = 10. **g** Cumulative percentage of 30-day-old flies showing escape responses on the 42 °C hot plates within 10 s. The flies were fed 0.05% DMSO, 20 μM MLN4924, or 20 μM TAS4464 continuously after eclosion. *n* = 100. **h** Glucose concentrations in the hemolymph of 3-day-old flies. The flies were fed 0.05% DMSO, 20 μM MLN4924, or 20 μM TAS4464 continuously after eclosion. *n* = 10. Data are presented as mean ± SD. One-way ANOVA with Tukey's multiple comparison test was used (**a**, **b**, and **d**). Mantel–Cox test was used (**c** and **g**). Two-way ANOVA with Sidak's multiple comparison test was used (**e**, **f**, and **h**). ****$p < 0.0001$. ns no significant.

*Snail* RNAi (Supplementary Fig. 9e–h). Furthermore, the diabetes-related defects in *UCH^KO* flies were completely rescued by expressing *GSK3B^CA* or knocking down *Snail* (Supplementary Fig. 9i–l). Therefore, we concluded that the diabetic defects induced by HSD are developed by the upregulation of *CUL1* expression via GSK3B-Snail pathway, which is antagonized by UCHL1 (Fig. 6k).

## Inhibition of CUL1 neddylation improves T2D- or DSN-related defects in fruit flies

It has been well established that the SCF E3 ligase complexes are fully activated by NEDD8 conjugation to Cullin proteins, a process known as neddylation. Similar to ubiquitination, neddylation is also processed by E1, E2, and E3 ligases. NEDD8 is activated by the E1 ligase complex, comprised of ubiquitin-like modifier activating enzyme 3 (UBA3) and NEDD8 activating enzyme E1 subunit 1 (NAE1). Upon activation of NEDD8, one of the two E2 ligases, ubiquitin-conjugating enzyme E2 M (UBE2M) or ubiquitin-conjugating enzyme E2 F (UBE2F), prepares the NEDD8 transmission to Cullin proteins. Finally, E3 ligases, including RBX1 and RBX2, perform the neddylation of Cullin proteins[52]. In particular, UBE2M and RBX1 are responsible for CUL1 neddylation[53]. With this understanding, we hypothesized that inhibiting the neddylation process of CUL1 could rescue the T2D- and DSN-like symptoms of *UCH^KO*, *CUL1* overexpression, and HSD flies.

Interestingly, the axonal degeneration and loss of leg sensory neurons and decreased escape responses against heat stimuli in *UCH^KO* flies were completely mitigated by expressing RNAi against *UBA3*, *UBE2M*, or *RBX1* using sensory neuron-specific GAL4 (Fig. 7a–c). Also, hyperglycemia of *UCH^KO* was rescued by expressing these three RNAi using a pan-neuronal GAL4 (Fig. 7d). Similarly, we identified that the axonal defects in leg sensory neurons and loss of leg sensory neurons in flies subject to HSD (Supplementary Fig. 10e, f) or sensory neuron-specific expression of *CUL1* (Supplementary Fig. 10a, b) was ameliorated by additional *UBA3*, *UBE2M*, or *RBX1* knockdown. The decreased heat escape responses in *OK371 > CUL1* or HSD flies were also improved by genetic inhibition of neddylation-related enzymes (Supplementary Fig. 10c, g). Additionally, the hyperglycemia of flies expressing *CUL1* in the whole body and HSD flies were rescued by *UBA3*, *UBE2M*, or *RBX1* knockdown (Supplementary Fig. 10d, h). Therefore, we proposed that the genetic inhibition of CUL1 neddylation could mitigate the DSN- or T2D-related phenotypes of *UCH^KO*, *CUL1* overexpression, and HSD flies.

It has been reported that two other Cullin ubiquitin ligases, CUL3 and CUL5, are activated by neddylation and can degrade IRS1 proteins[12,54]. Hence, we tested if the protective effects of knocking down neddylation enzymes against T2D- or DSN-like defects were mediated by inhibiting CUL3 or CUL5 neddylation. We expressed *CUL3* or *CUL5* RNAi using *hs*-GAL4 driver in HSD flies. In contrast to the

results from *CUL1* knockdown, knockdown of *CUL3* or *CUL5* did not improve the hyperglycemia of HSD flies (Supplementary Fig. 10i). Our findings implicated that the T2D-mitigating effects of knocking down neddylation enzymes were specifically mediated by CUL1 in fruit flies.

As we have established that T2D- and DSN-like symptoms were improved by genetic inhibition of neddylation enzymes, we tested if these defects could be alleviated by pharmacological inhibition of neddylation enzymes. So far, MLN4924 and TAS4464 are known to inhibit neddylation by targeting particular E1 ligases, NAE1[55,56]. Therefore, we fed *UCH^{KO}* adult flies with food containing 20 μM of either neddylation inhibitor and subsequently observed DSN-related defects. Surprisingly, MLN4924 or TAS4464 administration improved the axonal atrophy and loss of leg sensory neurons in *UCH* mutant flies (Fig. 7e, f). Furthermore, the decreased escape responses to heat stimuli in *UCH^{KO}* flies were completely rescued by feeding MLN4924 or TAS4464 (Fig. 7g). Similarly, the increased hemolymph glucose levels were also almost completely improved by feeding either NAE1 inhibitor (Fig. 7h). We then investigated whether administration of neddylation inhibitors could ameliorate the diabetic defects from flies with *CUL1* overexpression or upon HSD. The sensory neuronal defects, including axonal degeneration and neuronal loss, in the sensory neuron-specific *CUL1* overexpression flies (Supplementary Fig. 11a, b) and HSD flies (Supplementary Fig. 11e, f) were mitigated by feeding 20 μM MLN4924 or TAS4464. Also, the impaired responses to heat stimuli were improved by the administration of NAE1 inhibitors (Supplementary Fig. 11c, g). The hyperglycemia in whole-body *CUL1* overexpression flies and HSD flies was dramatically rescued by feeding MLN4924 or TAS4464 (Supplementary Fig. 11d, h). Moreover, using mammalian cell lines, we found that CUL1 neddylation levels (Supplementary Fig. 11i) and IRS1 ubiquitination levels (Supplementary Fig. 11j, j') were dramatically decreased by 10 μM TAS4464 treatment. These results suggested that both genetic and pharmacological suppression of CUL1 neddylation are highly effective strategies to medicate T2D- or DSN-related symptoms.

## Discussion

In this study, we demonstrated that UCHL1 regulates insulin signaling by acting as a deubiquitinase of IRS1. Since *UCHL1* is highly expressed in neurons, especially sensory neurons, loss of *UCHL1* induces axonal degeneration of the leg sensory neurons and leads to DSN. Furthermore, we showed that GSK3B-Snail-CUL1 axis functions as a negative regulator of insulin signaling via IRS1 ubiquitination which is antagonized by UCHL1. Therefore, diabetic sensory neuropathic defects can be improved by administrating inhibitors of CUL1 neddylation. Our findings imply that UCHL1 and CUL1 are underlying key factors for T2D and DSN.

It has been reported that chronic hyperglycemic conditions could develop into DSN via activation of the polyol pathway, hexosamine pathway, or accumulation of advanced glycation end products (AGEs)[57]. Therefore, medications for improving hyperglycemia are believed to be effective for the treatment of DN. Several rodent studies have indicated that treating drugs that mitigate elevated glycemia, including metformin, glitazone families, and gliptin families, can ameliorate DN-like symptoms. However, clinical approaches demonstrated that these chemicals do not efficiently medicate DN, especially in T2D patients[6]. Additionally, treating drugs that can inhibit the polyol pathway (fidarestat and ranirestat)[58], hexosamine pathway (ruboxistaurin)[59], and AGEs formation (aminoguanidine)[60] are not effective in clinical trials for curing DN. Therefore, it becomes evident that the identification of specific genetic factors and signaling pathways responsible for the pathogenesis of DN is crucial. Similar to human DN patients, aged *UCHL1* mutant flies show axonal degeneration of sensory neurons and numbness to external stimuli, and the sensory neuronal defects induced by HSD are ameliorated by UCHL1 overexpression. Furthermore, we have shown that these DN-like

defects, to be more precise, DSN-like defects, are independent of hyperglycemia and are rather dependent on the increase of insulin resistance specifically in sensory neurons by using *UCHL1* transgenic flies and high-sucrose feeding. Similar to our results, recent studies have reported that insulin resistance is increased in sensory neurons of rodent models with T2D[61,62]. These results further support the fact that glycemic control alone is not sufficient to improve DN in T2D patients.

Despite the diverse symptoms associated with DSN, such as numbness, burning, pain, and tingling, the primary underlying cause of DN is ultimately the axonal degeneration of the peripheral nervous system (PNS), with a preference for affecting sensory neurons[6]. Furthermore, numerous studies examining both human DSN patients and rodent DSN models has not given due consideration to the loss (apoptosis) of sensory neurons as a contributing factor to DSN development[63–65]. In light of this, it is worth noting that in our findings, we observed axonal degeneration in sensory neurons in *UCH^{KO}* flies, which resembled DSN-like symptoms. However, it is important to emphasize that the phenotypes involving the apoptosis and the loss of sensory neurons in *UCH^{KO}* flies do not directly correspond to the DSN observed in human patients. Also, our discovery that *UCH^{KO}* flies with inhibition of apoptosis still exhibited numbness in response to heat stimulation, supporting the previous researches that axonal degeneration in sensory neurons is the primary cause for DSN, rather than apoptosis or neuronal loss. Therefore, we propose UCHL1 plays a crucial role in DSN and the maintenance of sensory neuronal functions.

Since the Cullin family of E3 ligases ubiquitinates and destabilizes cell cycle-related proteins, including cyclin D1 (CCND1), cyclin E (CCNE1), cyclin-dependent kinase inhibitor 1A (p21), cyclin-dependent kinase inhibitor 1B (p27), and Wee1 G2 checkpoint kinase (Wee1), increased activity of Cullin proteins via neddylation can be attributed tumorigenesis[53,66]. Hence, MLN4924 and TAS4464, which can inhibit neddylation process, were initially developed as anti-cancer drugs. However, MLN4924 failed to pass the phase III clinical trial in patients with rare leukemia. Similarly, the phase I clinical trial of TAS4464 administration was ceased for unannounced reasons. Although the efficacy of the two drugs in tumor suppression was not sufficient, the effects of these two chemicals in neddylation inhibition have been well observed in clinical trials[67,68]. Thus, administration of MLN4924 or TAS4464 can be a promising therapeutic for patients suffering from T2D or DSN as we have shown that T2D- and DSN-like phenotypes are rescued by feeding MLN4924 or TAS4464 in flies. Consistent with our results, it has been reported that T2D symptoms and decreased insulin signaling can be improved by the treatment of neddylation inhibitors. MLN4924 injection improves high-fat diet (HFD)-induced glucose intolerance via inhibiting the neddylation of peroxisome proliferator-activated receptor gamma (PPARγ) in mouse models[69]. Similarly, treating MLN4924 chronically to western diet (WD)-fed mice rescues hyperglycemia through inhibiting CUL3 neddylation. Additionally, cellular treatment of MLN4924 or TAS4464 elevates the phosphorylation of Akt[70]. Consistent with these results, we have demonstrated that MLN4924 or TAS4464 administration can alleviate not only hyperglycemia but also neuropathy-like defects. Our results, along with previous studies, underscore the necessity of conducting clinical trials to evaluate the efficacy of MLN4924 and TAS4464 as potential treatments for T2D and DSN.

*UCHL1* knockout mice, named as gracile axonal dystrophy (GAD) mice, show several phenotypes similar to our *Drosophila* results. GAD mice were initially discovered to have neuroaxonal dystrophy[15], and several other neurodegenerative phenotypes have been subsequently observed in the mice. In particular, the death of sensory neurons in the anterior gracilis following their axonal degeneration, so-called dying-back degeneration, leads to sensory ataxia in GAD mice[71]. Furthermore, we investigated whether *UCHL1* knockout mice exhibit phenotypes resembling DSN and T2D. Regardless of sex, we observed a significant increase in the time for

UCHL1 knockout (*UCHL1^KO*) to jump to avoid thermal pain compared to WT mice (Supplementary Fig. 12a). Also, we conducted a glucose tolerance test (GTT) on male and female mice of both WT and *UCHL1^KO*. At the 60- and 120-min intervals following glucose injection, we observed elevated blood glucose levels in male *UCHL1^KO* mice compared to male WT mice (Supplementary Fig. 12b). Moreover, at the 30-min mark after glucose injection, we observed higher blood glucose levels in female *UCHL1^KO* mice compared to female WT mice (Supplementary Fig. 12b). Additionally, we assessed the ability to clear blood glucose by calculating the area under curve (AUC) from the GTT graphs. In both male and female mice, the AUC was significantly greater in *UCHL1^KO* mice compared to WT mice, indicating higher overall blood glucose levels over the observed period (Supplementary Fig. 12b). Moreover, we performed antibody staining on sectioned dorsal root ganglion (DRG) samples, known for their abundance of sensory neurons, from WT mice against neurofilament-H (NF-H) to label sensory neurons[25] and UCHL1. We found that all cells expressing NF-H were also positively stained with anti-UCHL1 antibody. Moreover, we observed a significantly higher staining intensity of UCHL1 specifically in sensory neurons, identified as DAPI⁺NF-H⁺ cells, among the various cell types present within the DRG (Supplementary Fig. 12c). This observation was consistent with our results in *Drosophila*, where we observed a robust expression of *UCH* in sensory neurons. Additionally, we conducted experiments to investigate insulin signaling in sensory neurons within the DRGs. To assess this, we measured the staining intensity of pAkt in cells that were positively stained for DAPI and NF-H. While the staining intensity of Akt in the DRG sensory neurons was comparable between WT and *UCHL1^KO* mice (Supplementary Fig. 12d), our immunohistochemistry findings indicated a notable decrease in pAkt levels in the DRG sensory neurons of *UCHL1^KO* mice compared to control mice (Supplementary Fig. 12e). In addition to these preliminary data, we are currently in the process of investigating whether the other results obtained from our *Drosophila* study are also consistent in rodent models. Also, we aim to determine if defects in insulin signaling can induce neurodegenerative phenotypes similar to those observed in GAD mice.

Insulin signaling is mainly transmitted intracellularly via several kinases. Hence, identifying the regulatory mechanisms of insulin signaling and the pathogenesis of T2D had been focused on protein phosphorylation, especially a feedback phosphorylation of IRS1 by p70 ribosomal protein S6 kinase (S6K)[72]. On the other hand, recent research emphasizes the importance of transcriptional regulations of the genetic factors controlling insulin signaling and T2D. For instance, *F-box only protein 2* (*FBXO2*) is upregulated in HFD mice, degrading insulin receptors (IR). The ablation of *FBXO2* in mice, therefore, mitigates T2D-related phenotypes, including hyperglycemia and glucose intolerance[73]. Additionally, the downregulation of *aryl hydrocarbon receptor* (*AhR*) transcription has been observed in HFD-fed mice. The decrease of AhR leads to inhibition of insulin signaling by reducing *PI3K* and *Akt* transcription[74]. Similarly, we have established that *CUL1* transcription is upregulated in response to HSD, thereby controlling diabetic defects in *Drosophila*. Also, we have identified that *CUL1* transcription is controlled by GSK3B, one of the main components of insulin signaling, and Snail, a transcription factor. Therefore, we suggest the GSK3B-Snail pathway as a regulator of insulin signaling and T2D pathogenesis at the level of transcription.

In conclusion, we have discovered the neuron-specific roles of UCHL1 in insulin signaling and highlighted them as one of the potential causes for T2D and DSN. Mechanistically, UCHL1 counteracts CUL1-dependent IRS1 ubiquitination. Therefore, we propose the physiological functions of UCHL1 as an integrative factor connecting T2D and DSN in insulin signaling.

# Methods

## Ethical statement
This study fully complied with the ethical regulations set by Seoul National University and Pohang University of Science and Technology. The fly experiments in this study were approved by the Institutional Animal Care and Use Committee at Seoul National University. The mouse experiments in this study were approved by the Institutional Animal Care and Use Committee at Pohang University of Science and Technology.

## Fly stocks and their maintenance
The *Drosophila* lines used in the experiments are below.

Bloomington Drosophila Stock Center: *w1118* (3605), *cg*-GAL4 (7011), *mef2*-GAL4 (27390), *elav*-GAL4 (8765), *nSyb*-GAL4 (51635), tub-GAL4 (5138), *DILP2*-GAL4 (37516), *OK371*-GAL4 (26160), *hs*-GAL4 (2077), *OK6*-GAL4 (64199), *tub*-GAL80^TS (7017), UAS-*DILP2* (80936), UAS-*nlsGFP* (4775), UAS-*IR^CA* (8250), UAS-*PI3K^CAAX* (8294), UAS-*myrAkt* (80935), and UAS-*lamin-GFP* (7378), *UAS-GFP* (1521).

Vienna Drosophila Resource Center: UAS-*UCH RNAi* (103614), UAS-*TRAF4 RNAi* (21214), UAS-*Snail RNAi* (6232), UAS-*UBA3 RNAi* (17139), UAS-*UBE2M RNAi* (35219), UAS-*RBX1 RNAi* (32399), UAS-*CUL3 RNAi* (25875), and UAS-*CUL5 RNAi* (52176).

FlyORF: UAS-*CUL1* (F001068) and UAS-*Snail* (F000066).

Kyoto Drosophila Stock Center: UAS-*GSK3B^CA* (108125) and UAS-*GSK3B^DN* (108184).

Japanese National Institute of Genetics: UAS-*Cbl RNAi* (7037R-1) and UAS-*CUL1 RNAi* (1877R-1), UAS-*Drice* RNAi (7788R-1), UAS-*IRS1 RNAi* (HMS01553), UAS-*CCND RNAi* (HMS00059), UAS-*CCNE RNAi* (3938R-2), UAS-*CDK4 RNAi* (5072R-3), UAS-*Wee1 RNAi* (GL00039), UAS-*DILP2 RNAi* (HMS00476), UAS-*Akt RNAi* (4006R-1).

UAS-*dIRS1* was provided by Dr. S. Naganos[36]. UAS-*DILP2-Flag* was provided by Dr. K. Yu[20]. *UCH^KO* and *UCH^C93S*, UAS-*PKM* were generated by Dr. D. Lee in the previous paper[19]. cDNA sequence of *UCH* (GH02396 from Drosophila Genomics Resource Center) was cloned into the pUAST vector and the pUAST-*UCH* plasmid was microinjected into *w1118* embryos for the generation of UAS-*UCH*. Normal diet (ND)-fed flies were grown on food containing 35 g cornmeal, 70 g dextrose, 5 g agar, 50 g dry active yeast, 4.6 ml propionic acid, and 7.3 ml Tegosept (100 g/l in ethanol) per liter at 25 °C. High-sucrose diet (HSD)-fed flies were grown on food containing 35 g cornmeal, 70 g dextrose, 5 g agar, 50 g dry active yeast, 4.6 ml propionic acid, 7.3 ml Tegosept (100 g/l in ethanol), 250 g sucrose per liter at 25 °C after eclosion. MLN4924 (HY-70062, MedChemExpress) or TAS4464 (HY-128586, MedChemExpress) dissolved in DMSO was added into the normal or high-sucrose food. *w1118* was used as a wild-type control and the flies were fed normal food unless otherwise indicated. We crossed *UCH^KO* and *UCH^C93S* lines with *w1118* flies for 10 generations to ensure that they have the same genetic background as the *w1118* control group. We identified that all the rescue or suppression experiments conducted using more than one UAS transgene did not affect transgene expression efficacy.

## Antibodies
Anti-pAkt (S505) for *Drosophila* (4054S, Cell Signaling), anti-DILP2 (provided by Dr. K. Yu)[20], anti-pAkt (S473) (4060S, Cell Signaling), anti-Akt (9272S, Cell Signaling), anti-Flag (2368S, Cell Signaling), anti-Flag (M185-3L, MBL), anti-UCHL1 (13179S, Cell Signaling), anti-Tubulin (DSHB), anti-HA (3724S, Cell Signaling), anti-IRS1 (2390S, Cell Signaling), anti-Myc (M192-3, MBL), anti-CUL1 (4995S, Cell Signaling), anti-NEDD8 (2745, Cell Signaling), anti-mouse HRP (115-035-146, Jackson), anti-rabbit HRP (111-035-144, Jackson), anti-rabbit TRITC (111-296-144, Jackson), and anti-Neurofilament-H (NF-H) (AB5539, Sigma) antibodies were used in this study.

## Hemolymph glucose measurements

15 male and 15 female flies were punctured at the thorax with a microneedle. After that, the flies were transferred to a 600 µl tube with a tiny hole at the bottom to obtain 1 µl of hemolymph and centrifuged at 3000 rpm for 5 min at 4 °C. The collected hemolymph was diluted 1:10. 10 µl of diluted hemolymph was added into 200 µl of glucose assay reagent (G3293, Sigma) and incubated for 10 min at 37 °C. Then, the absorbance at 340 nm was measured using Tecan Plate Reader Infinite 200.

## TAG measurements

5 male and 5 female flies were homogenized in 500 µl of 0.1% PBST (PBS + 0.1% Tween 20) using plastic pestles and 10 µl of lysate was used to measure the total protein levels by Pierce BCA protein assay kit (23225, Thermo Scientific). 490 µl of lysate was heated at 70 °C for 5 min and chilled on ice for 10 min. 1 µl of lipoprotein lipase from *Chromobacterium viscosum* (437707, Calbiochem) was mixed into the lysate and incubated at 37 °C overnight. After overnight incubation, the lysate was centrifuged at 14,000 rpm for 10 min. 20 µl of the supernatant was added into 180 µl of free glycerol reagent (F6428, Sigma) and incubated at 37 °C for about 10 min. Then, the absorbance at 540 nm was measured using Tecan Plate Reader Infinite 200 and these data were divided by each of the total protein levels[75].

## Glycogen measurements

5 male and 5 female flies were homogenized in 500 µl of 0.1% PBST (PBS + 0.1% Tween 20) using plastic pestles and 10 µl of lysate was used to measure the total protein levels by Pierce BCA protein assay kit (23225, Thermo Scientific). 490 µl of lysate was heated at 70 °C for 5 min and chilled on ice for 10 min. 1 µl of lipoprotein lipase from *Chromobacterium viscosum* (437707, Calbiochem) was mixed into the lysate and incubated at 37 °C overnight. After overnight incubation, the lysate was centrifuged at 14,000 rpm for 10 min. 30 µl of the supernatant was incubated with 14 units of amyloglucosidase (10115, Sigma) at 50 °C for 1 h. 15 µl of the mixture was added into 200 µl of glucose assay reagent (G3293, Sigma) at 37 °C for 30 min. Then, the absorbance at 340 nm was measured using Tecan Plate Reader Infinite 200 and these data were divided by each of the total protein levels[75].

## Trehalose measurements

3 male and 3 female flies were homogenized in 500 µl of 70% ethanol. The lysate was centrifuged at 14,000 rpm for 10 min. After the centrifugation, the lysate was vacuum-centrifuged at 50 °C for about 1 h until the samples were completely dried. The dried sample was mixed with 200 µl of 2% sodium hydroxide and incubated at 100 °C for 10 min. 40 µl of the mixture was added into 1 ml of Anthrone reagent (0.2% anthrone (10740, Fluka) + 72% sulfuric acid (320501, Sigma)) and incubated at 90 °C for 20 min. The mixture was cooled down to room temperature (RT) and the absorbance at 620 nm was measured using Tecan Plate Reader Infinite 200[75].

## Food intake assay

For the CAFE assay, two capillaries filled with 5 µl of 10% sucrose solution were provided to 4 flies in a vial for 24 h and then measured the ingestion volumes[76].

For the visualization and spectrophotometric analysis of food intake (Fig. 1e), female flies were starved for 24 h only with water supply. The fasted flies were fed on food consisting of 10% sucrose solution and 1% brilliant blue FCF (80717, Merck) for 1 h. These flies were then homogenized in 200 µl of distilled water. A further 800 µl of water was added and the homogenates were filtered through 0.22 µm Millex filter (SLMP025SS, Millipore). The absorbance of the filtered homogenates was measured at 629 nm using Tecan Plate Reader Infinite 200[77,78].

## RNA extraction and real-time PCR

For extracting RNA from *Drosophila*, 5 male and 5 female flies were collected for RNA extraction. The flies were homogenized in 500 µl of TRIzol solution (15596018, Invitrogen). The lysate was centrifuged at 14,000 rpm 4 °C for 15 min. 400 µl of supernatant was mixed with 100 µl of chloroform and the mixture was centrifuged at 14,000 rpm 4 °C for 15 min. After centrifugation, 200 µl of the transparent solution was mixed with 200 µl of isopropanol and the mixture was centrifuged at 14,000 rpm 4 °C for 15 min. Then, the supernatant was removed and the pellet was washed using 500 µl of 75% ethanol (centrifuge at 14,000 rpm 4 °C for 5 min). The ethanol supernatant was discarded and the pellet was dried for about 10 min. 10 µl of water was mixed with the pellet. Afterward, 10 µl of RNA solution was mixed with 1 µl of random primer (C1181, Promega) and incubated at 70 °C for 10 min. The solution was cooled down in the ice and the cDNA was synthesized using M-MLV reverse transcriptase (M1701, Promega). Real-time PCR was performed using TOPreal multi-probe qPCR premix (RT620S, Enzynomics). All the gene expression data were normalized by *rp49* expressions.

For extracting RNA from HEK293E cells, cells were cultured in 6-well plates for RNA extraction. The cells were harvested and homogenized in 500 µl of TRIzol solution (15596018, Invitrogen). Afterwards, the above procedures were used and all the gene expression data were normalized by *rp30* expressions.

## Glucose tolerance test and insulin tolerance test in *Drosophila*

For the glucose tolerance test (GTT), 3-day-old flies were collected and fasted for 1 day. After 1-day fasting, the flies were fed on food containing 10% sucrose soaked on Whatman papers for 1 h. Then, the flies starved again continuously and the hemolymph glucose was measured at the time point of 0.5 h, 1 h, 1.5 h, 2 h, 2.5 h, or 3 h.

For the insulin tolerance test (ITT), the flies carrying *tub*-GAL80[TS], *DILP2*-GAL4, and UAS-*DILP2* were generated. The flies were grown at 19 °C from the embryo stage to adult stage. When the flies got 3-day-old, they fasted for 6 h only with water supply. After the 6-h fasting, the flies were placed at 29 °C for 30 min to express *DILP2* while keeping the flies fast. The hemolymph glucose was measured every 15 min still keeping the flies fast after *DILP2* induction.

## Immunostaining of pAkt and immunostaining/staining of DILP2 in *Drosophila*

For immunostaining of the adult brains or muscles, each tissue was dissected in Schneider's *Drosophila* medium (21720-024, Gibco). After dissection, the tissues were incubated in Schneider's *Drosophila* medium with or without 1 µM human insulin (11376497001, Roche) treatment for 20 min. The tissues were then fixed with 4% paraformaldehyde for 30 min at RT and washed twice with 0.1% PBST. The washed tissues were permeabilized with 0.5% PBST for 20 min. The permeabilized tissues were washed once again with 0.1% PBST and blocked with 0.1% PBST containing 3% BSA. Anti-pAkt antibody (S505, 4054S, Cell Signaling)[79] was treated at 4 °C for overnight. The samples were washed three times with 0.1% PBST and incubated at RT for 1 h with a secondary antibody, anti-rabbit TRITC (111-296-144, Jackson). The samples were mounted in 80% PBG (20% PBS + 80% glycerol) and observed by LSM710 confocal microscope (Carl Zeiss) via Z-stack analysis.

For immunostaining of DILP2 neurons, male flies were fixed with 4% paraformaldehyde for 3 h. After fixation, the brains were dissected and washed twice with 0.1% PBST. The brains were permeabilized with 0.5% PBST for 20 min and washed once again with 0.1% PBST. The brains were blocked with 0.1% PBST containing 3% BSA for 1 h and then treated with anti-DILP2 antibody[20] at a dilution of 1:200. The samples were washed three times with 0.1% PBST and incubated at RT for 1 h with a secondary antibody, anti-rabbit TRITC (111-296-144, Jackson).

The samples were mounted in 80% PBG and observed by LSM710 confocal microscope (Carl Zeiss) via Z-stack analysis.

For staining of DILP2 neurons, flies expressing UAS-*GFP* under *DILP2*-GAL4 were generated. Male flies were fixed with 4% paraformaldehyde for 3 h. After fixation, the brains were dissected. washed three times with 0.1% PBST, and mounted in 80% PBG and observed by LSM710 confocal microscope (Carl Zeiss) via Z-stack analysis.

Fluorescence intensity of these tissues were quantified using ImageJ[80].

### ELISA for DILP2

DILP2 ELISA was performed with the flies expressing DILP2-Flag under *DILP2*-GAL4 driver[20]. 0.5 µl of hemolymph was collected from 15 males and 15 females. The samples were then diluted in PBS at a ratio of 1:200. The 96-well plate (M4561, Merck) were coated overnight at 4 °C with the diluted hemolymph. After that, the wells were cleared and the DILP2 antibody was added to the wells at a ratio of 1:500 and incubated at RT for 2 h. The secondary antibody, anti-rabbit HRP, was treated at RT for 1 h at a ratio of 1:1000. TMB solution (34028, Thermo Scientific) was used for color development, and the absorbance was measured at 450 nm. The standard curve for quantification was analyzed with diluted 3× Flag peptide (F4799, Sigma).

### Staining and counting the sensory neurons of fly front legs

For staining the nuclei of leg sensory neurons, the male flies carrying *OK371*-GAL4 and UAS-*nlsGFP* were generated. For staining the somas and axons of sensory neurons, the male flies carrying *OK371*-GAL4 and UAS-*GFP* were generated[27]. The front legs of the flies were dissected in PBS and mounted in 80% PBG (sex combs at the upper side). The legs were observed by LSM710 confocal microscope via Z-stack analysis. After the Z-stack imaging, the green dots were counted for data quantification.

For the TUNEL assay, the male flies carrying *OK371*-GAL4 and UAS-*nlsGFP* were also generated. When these flies became 20-day-old, the front legs of the flies were dissected in PBS and fixed with 4% paraformaldehyde for 20 min at RT. After fixation, the front legs were incubated with 0.1% sodium citrate in 0.1% PBST for 30 min at 60 °C. In situ cell death detection kit (12156792910, Roche) was used to visualize apoptosis. The tissues were mounted in 80% PBG and observed by LSM710 confocal microscope via Z-stack analysis.

### Escape response assay

For measuring escape responses to heat stimuli, the heat block (BF-20HB, Biofree) was set to 42 °C. Then, the flies were put onto the 42 °C plate with an oral aspirator without $CO_2$ anesthetization. The times when the flies showed the first jumping (escape) responses were measured.

For measuring escape responses to acid stimuli, Whatman paper soaked with 12% sulfuric acid (320501, Sigma) was placed on the cell culture dish with 90 mm diameter (20100, SPL). Then, the flies were put into the acid dish with an oral aspirator without $CO_2$ anesthetization. The times when the flies showed the first jumping (escape) responses were measured.

### Nuclei extraction from the fly whole body or front leg

Flies expressing UAS-*lamin-GFP* under appropriate GAL4 lines were collected in the 1.5 ml tube. The flies were submerged in 100% ethanol for 5 s and washed twice for 30 s with Schneider's *Drosophila* medium. For nuclei extraction from the front legs, 100 pairs of the front legs were dissected in Schneider's *Drosophila* medium and were ground in a homogenization buffer. For nuclei extraction from the whole body, 30 flies were ground in a homogenization buffer. The lysate was filtered with a 10 µm cell strainer (43-50010-03, PluriSelect) and centrifuged for 10 min at $1000 \times g$ at 4 °C. After centrifugation, the supernatant was discarded and the pellet was resuspended gently

using 500 µl of resuspension buffer. The nuclei-containing solution was transferred to a FACS tube and was stained with 1 µl of Hoechst 33342 (62249, Thermo) for 5 min[28].

For setting up FACS (FACS AriaIII, BD biosciences) gates, we first ran the unstained control. Next, we loaded the samples stained with Hoechst, but not expressing lamin-GFP. After gating setup, we gated Hoechst⁺ and then gated GFP⁺. The average numbers of Hoechst⁺GFP⁺ nuclei extracted from the whole body were about 26,000 (*hs > lamin-GFP*), 16,300 (*elav > lamin-GFP*), 3,00 (*OK6 > lamin-GFP*), and 2800 (*OK371 > lamin-GFP*). The average numbers of Hoechst⁺GFP⁺ nuclei extracted from the front legs were about 5200 (*hs > lamin-GFP*), 870 (*elav > lamin-GFP*), 60 (*OK6 > lamin-GFP*), and 440 (*OK371 > lamin-GFP*). RNAs were extracted from the extracted nuclei right after the nuclei sorting and the gene expressions were measured (see the "RNA extraction and real-time PCR" section).

### Tracking and analyzing fly movements

Male flies were collected under $CO_2$ anesthesia 3 h before the video tracking. A single fly was transferred into a petridish (10060, SPL) filled with silicon (Sylgard 182 Silicone Elastomer kit, Dow Corning). For the high quality of movement detection, we filled the petridish space <2 mm below the top of the dish with silicon. An insulated chamer (675 × 440 × 410 mm) for video recording was built with a white LED light and a camera (HD pro-webcam C910, Logitech). The flies were acclimated in the chamber for 10 min and the total trajectories were recorded for 5 min (30.06 frames/s, 0.033 s/frame). These videos with about 9,018 frames were analyzed using Ctrax software[81]. The distances flies walked were measured using the behavioral microarray MATLAB toolbox from Ctrax website[82].

### Measuring *Drosophila* sleep patterns

Locomotor activity and sleep was recorded using the *Drosophila* Activity Monitoring System (Trikinetics). Flies were placed into individual 65-mm tubes containing food consisting of 2% agar and 5% sucrose. Flies were acclimated overnight, after which locomotor activity was monitored. Locomotor activity was measured in 1-min bins and sleep was defined as immobility lasting at least 5 min. Data was analyzed using SCAMP software[83].

### Protein binding prediction using ColabFold

The amino acid sequences of the PH-PTB domain of human IRS1 (1-300; P35568) and human UCHL1 (1-223; P09936) were predicted simultaneously from ColabFold website[35]. The prediction model with the highest pTMscore was chosen and visualized via MolStar (Mol*) from Protein Data Bank (PDB).

### Transcription factor prediction using *JASPAR*

The genomic sequences of *CUL1* in human and *Drosophila* around the translation start site (−350 to +150 base pair) were analyzed through *JASPAR* website (human: ATGATTTTGA-GTATCCTCCT, *Drosophila*: TTTTATTGCT-TCACGTCTAT). These sequences were scanned by 80% of the relative profile score threshold. Among the predicted transcription factors, the TFs with a relative score above 0.97 were chosen[42].

### Cell culture, transfection, and chemical treatment

HEK293E and MEF cells, kindly gifted from Dr. John Blenis at Weill Cornell Medicine, were cultured in DMEM (LM 001-05, Welgene) supplemented with 10% fetal bovine serum (FBS; 16000044, Gibco) at 37 °C in a humidified atmosphere of 5% $CO_2$. SH-SY5Y cells, obtained from Korean Cell Line Bank, were cultured in DMEM (LM 001-05, Welgene) supplemented with 20% FBS at 37 °C in a humidified atmosphere of 5% $CO_2$. SNU398 cells, obtained from Korean Cell Line Bank, were cultured in RPMI 1640 (LM 011-03, Welgene) supplemented with 10% FBS at 37 °C in a humidified atmosphere of 5% $CO_2$. HEK293E cells

were transfected using polyethylenimine, branched (PEI) (408727, Sigma) according to the instructions by the manufacturer. For siRNA transfection, we used RNAi max (13778150, Invitrogen). Cells were treated with insulin (11376497001, Roche), MG132 (474790, Millipore), or TAS4464 (HY-128586, MedChemExpress). siRNA sequences used in this paper are shown in Supplementary Table 2.

## Plasmid constructs

pcDNA3.1 zeo(+) UCHL1 (NM_004181) Myc/His-tagged and pcDNA3.1 zeo(+) UCHL1 were used[19]. UCHL1 C90A mutant was generated using standard site-directed point mutagenesis method. Tag2B Flag IRS1 (NM_005544.3) was gifted from Dr. Ko[34]. PRK5 HA ubiquitin was obtained from Addgene (#17608). pcDNA3 Myc3 CUL1 (NM_003592.3) was also obtained from Addgene (#19896). CUL1 DN mutant (1-452) was cloned into pcDNA Myc3 vector. pcDNA3 Myc3 RBX1 (NM_014248.4) was obtained from Addgene (#20717). NEDD8 (NM_006156.3) was cloned into the pCMV10 vector. pcDNA3 HA GSK3B (NM_002093.4) mutants (S9A and K85A) were generated using standard site-directed point mutagenesis method.

## Generation of UCHL1 KO cells

The CRISPR-Cas9 system was used for the generation of *UCHL1* KO cells[19]. To generate *UCHL1* KO HEK293E cells, the guide RNA sequence (GTGGCGCTTCGTGGACGTGC) was inserted into the PX459 plasmid (62988, Addgene). The vector was transfected into HEK293E cells. 2 days after transfection, the transfected cells were selected with puromycin (5 µg/ml) for 3 days. Then, single colonies were transferred onto 96-well plates with one colony in each well. Immunoblot analysis with anti-UCHL1 antibody (13179S, Cell signaling) was performed to screen *UCHL1* KO clones. *UCHL1* KO SH-SY5Y cells were generated in the previous paper[19].

## Immunoprecipitation and immunoblot analysis

Cells were lysed using a lysis buffer [20 mM tris (pH 7.5), 100 mM NaCl, 1 mM EDTA, 2 mM EGTA, 50 mM β-glycerophosphate, 50 mM NaF, 1 mM sodium vanadate, 1 mM phenylmethylsulfonyl fluoride (PMSF), leupeptin (10 µg/ml), pepstatin A (1 µg/ml), and 1% Triton X-100] to perform immunoprecipitation. The cell lysates were centrifuged at 13,000 rpm 4 °C for 20 min. Then, the supernatant incubated at 4 °C overnight after the addition of primary antibodies. The lysates were incubated with protein A/G agarose beads for 2 h at 4 °C, washed four times in detergent-free lysis buffer, and eluted with 2× Laemmli buffer at 95 °C. Cells were lysed using radioimmunoprecipitation assay (RIPA) buffer [50 mM tris (pH 8.0), 150 mM NaCl, 0.5% sodium deoxycholate, 1% NP-40, 0.1% SDS, 50 mM NaF, 1 mM sodium vanadate, 1 mM PMSF, leupeptin (10 µg/ml), and pepstatin A (1 µg/ml)] to perform immunoblot analysis. Total protein was quantified using Pierce BCA protein assay kit (23225, Thermo Scientific). Lysates were subjected to SDS−PAGE analysis. LAS-4000 (Fujifilm) was used to develop and observe the immunoblots. Immunoblot band intensity was quantified using ImageJ.

## Mice

C57BL/6J (wild-type, Stock #000664) and *Uchl1*$^{gad-J}$/J (UCHL1+/-) mice (Stock #024355)[84] were obtained from the Jackson Laboratory. *UCHL1* heterozygous mice (*UCHL1*$^{+/-}$) had normal fertility and were bred to produce *UCHL1*$^{-/-}$ mutant mice (*UCHL1*$^{KO}$). All the mice were maintained in a specific pathogen-free facility, following a 12-h light – 12-h dark cycle. The room temperature was maintained at 22 °C ± 1 °C with a humidity level of 50% ± 10%. Mice were provided with ad libitum access to rodent diet (Purina Laboratory Rodent Diet 38057) and water. All procedures were approved by the Institutional Animal Care and Use Committee at Pohang University of Science and Technology and performed in accordance with its guidelines (POSTECH-2022-0080).

## Glucose tolerance tests in mice

After fasting for 16 h overnight, the mice received an intraperitoneal injection of a 20% D-Glucose solution (Sigma, G8769) at a dosage of 2 mg of glucose per g of body weight. Blood glucose levels were assessed by obtaining tail vein blood samples before the injection and at 30, 60, 90, and 120 min after the injection, using a Contour Plus glucometer (Bayer).

## Hot plate test in mice

The mice were positioned on a hot plate (C&A Scientific Co., Inc., XH-2001, US) set to a temperature of 52 °C. The mice were immediately removed from the hot plate after measuring how long it took for the initial indication of nociception, such as paw licking or a jumping response to appear. To ensure the safety of their paws, a maximum time limit of 25 s was implemented[85].

## Isolation and immunostaining of mouse DRG

Mice were subjected to anesthesia using isoflurane and thereafter perfused with 10 ml of PBS followed by 4% paraformaldehyde (PFA) (158127, Sigma) at RT. The DRGs were dissected and afterward fixed in a 4% PFA solution at 4 °C overnight. After fixation, the DRGs were washed in PBS, immersed overnight at 4 °C in a 30% sucrose solution (S0389, Sigma), and then embedded in the Tissue-Tek OCT compound (4583, Sakura Finetek). Frozen DRG tissues were sectioned into 20 µm slices. Each section was incubated in a blocking buffer consisting of 5% donkey serum, 2% cold-water fish gelatin, and 1% bovine serum albumin in 0.3% Triton X-100 in PBS for 1 h at RT, followed by incubating with primary antibodies at 4 °C for 12 h. Following the incubation of primary antibodies, all sections were rinsed with PBS three times and subsequently stained with secondary antibodies for 2–4 h at RT. The following antibodies and dilutions were used: UCHL1 (13179S, Cell Signaling, 1:200), pAkt (4060S, Cell Signaling, 1:400), Akt (4685S, Cell Signaling, 1:400), NF-H (AB5539, MilliporeSigma, 1:1000), Cy3-conjugated anti-rabbit antibody (711-165-152, Jackson, 1:400) and Alexa Fluor 488-conjugated anti-chicken antibody (703-545-155, Jackson, 1:400). Subsequently, all sections were mounted with DAPI Fluoromount-G (0100-20, SouthernBiotech,). The images were obtained using a Yokogawa CSU-W1 SoRa Spinning Confocal Microscope (Nikon).

## Statistical analysis

A blind manner was used in all experiments and analyses. For p value measurements, two-tailed paired Student's *t*-test, one-way ANOVA with Tukey's multiple comparison test or Holm−Sidak's multiple comparison test, RM one-way ANOVA with Holm−Sidak's multiple comparison test, two-way ANOVA with Sidak's multiple comparison test, and Mantel−Cox test were used. All tests were performed via GraphPad Prism v.9 for statistical analysis.

## Reporting summary

Further information on research design is available in the Nature Portfolio Reporting Summary linked to this article.

## Data availability

JASPAR databases are available at https://jaspar.genereg.net. ColabFold databases are available at https://colabfold.mmseqs.com. Mol* 3D Viewer is available at https://www.rcsb.org/3d-view. Clustal Omega for protein sequence alignment is available at https://www.ebi.ac.uk/Tools/msa/clustalo/. ARCHS4 databases are available at https://maayanlab.cloud/archs4/. All data needed to evaluate the conclusions in the paper are present in the paper and the supplementary information. The raw data and *p* values for all Figures and Supplementary Figures are available in Source Data files, accompanying this paper. All additional information is available upon request to the corresponding author (J.C.: jkc@snu.ac.kr). Source data are provided with this paper.

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

## Acknowledgements

We would like to thank FlyBase, Bloomington Drosophila Stock Center, Vienna Drosophila Resource Center, Japanese National Institute of Genetics, FlyORF, Kyoto Drosophila Stock Center, and Drosophila Genomics Resource Center for fly stocks and related resources. We would like to acknowledge Dr. S. Naganos and M. Saitoe for generously sharing their dIRS1-related fly lines. We are also grateful to Dr. K. Yu for DILP2-related fly lines and reagents. Illustrations in this manuscript were created using Biorender.com. This study was supported by grants from National Research Foundation of Korea (NRF) funded by the Korean government (MSIT): NRF-2020R1A5A1018081 and NRF-2021R1A2C1010577 to J.C. NRF-2022R1A6A3A01085862 to D.L. RS-2023-00211029 to S.J.H. NRF-2022R1C1C1011895 to S.C. D.L., E.Y., D.H.L, S.K., and J.C. were supported by BK21 Plus Program from the Ministry of Education, Republic of Korea.

## Author contributions

D.L., E.Y., S.J.H., and J.C. designed this research. D.L. and S.J.H. initiated this research. D.L., E.Y., S.J.H., D.H.L., K.L., H.J., S.C., and J.C. performed the experiments and analyzed the data. D.L., E.Y., S.J.H., D.W., D.H.L., S.K., K.L., H.J., S.C., and J.C. wrote the manuscript and discussed the results and commented on the manuscript.

## Competing interests

The authors declare no competing interests.
