## [Peer Review File · Nature Communications]

Diabetic sensory neuropathy and insulin resistance are induced by loss of UCHL1 in *Drosophila*REVIEWER COMMENTS

Reviewer #1 (Remarks to the Author):

This study describes the use of a fly model to characterize the potential role of UCHL proteins in the pathogenesis of diabetes. It is an interesting study that sheds light on a mechanism that may be important for diabetes. The manuscript includes an extensive and impressive set of experiments that mostly support the conclusions. I support publication, but some concerns need to be addressed first. Most experiments are done with mutants, indicating that the effects could be developmental defects. The authors should induce knockdown or knockout in the adult stage to exclude this possibility. Since the fly *Unc* is predominantly expressed in the nervous system, its loss likely affects the behavior of the animal. This could indirectly change metabolism and other parameters. If the animal is overeating then that could explain the increased blood sugar levels, and if animals are inactive or sleep disturbed then that could also lead to many other effects. The authors should look more carefully at behavioral changes for example activity and sleep pattern. I was also wondering about the genetic background control. They authors use *w1118*, which is widely used, but it is not clear that it is the right background control for their mutants. Were the mutants generated in this specific background? They should also use some combination of trans-heterozygous and they could use heterozygous as control as well, at least for a few key phenotypes.

Other points:

1. *Drosophila Unc* is more homologous to human UCHL3 than UCHL1. The human genes are obviously paralogs, but it is better to say the fly gene is homologous.
 2. They should explain the exact nature of the UCHKO and UCHC93S aberrations and alleles. How was it confirmed that the catalytic domain was disrupted?
 3. When comparing hemolymph glucose levels or other parameters it is important that animals are the same development stage. The authors should show whether these mutants are delay in their development. The easiest would be to show timing of pupariation. If there is no delay in pupariation then hemolymph glucose and other parameters can be compared. However, if mutant larvae are delay it is more complicated. A way to address this would be to measure for example 24 h before pupariation in all genotypes. This is also a good time for metabolic measurements because it is before the wandering stage. If there is no developmental delay, this would also indicate the effects observed in the adult are not caused by developmental defects.
 4. It is surprising that both hemolymph glucose and glycogen goes up. If tissues are insulin resistant or insulin signaling is reduced in some way, then increased hemolymph (circulating) glucose is expected. However, tissues should have impaired glucose uptake and therefore decrease glycogen. How do the authors explain this?
 5. Fig. 1d. The authors measure DILP expression, but expression does not reflect insulin secretion. They need to stain for DILP2, -3 and -5 and somehow measure systemic insulin signaling. The best way would be to do a DILP2 ELISA to measure DILP2 in the hemolymph. It would also be good to stain for whole-body phosphorylated AKT.
 6. Fig. 1e. When measuring food intake, it is not sufficient to draw conclusions based on images of flies with food dye color in the abdomens. The author must measure actual food consumption spectrophotometrically.
- Fig. 1f. The authors write that it takes significantly longer for the *Unc* mutants flies to decrease their blood sugar levels, but it seems the slope is the same as the control. This means that they clear blood sugar at the same rate, but they just begin with higher levels. This would not indicate insulin resistance.

Fig. 1g. Based on the lower levels of hemolymph glucose in control animals overexpressing DILP2 than mutants, the authors conclude the mutant animals are glucose intolerant and insulin resistant. To draw this conclusion they need to stimulate tissue with insulin and measure response for example by measuring pAKT, which they do later. But here they cannot yet conclude it.

Reviewer #2 (Remarks to the Author):

In this manuscript, Lee et al report that UCHL1, a deubiquitinase, antagonizes the function of E3 ligase Cullin 1 (CUL1) by deubiquitinating insulin receptor substrate 1 (IRS1). Consequently, overexpression of UCH, the homolog of UCHL1 in *Drosophila*, improves insulin sensitivity, while dysfunction of UCH contributes to diabetic neuropathy (DN) and T2D in *Drosophila*. On the other hand, genetic and pharmacological suppression of CUL1 activity fully rescues the T2D- and DN-associated phenotypes.

This is an interesting study of molecular mechanism underlying DN. The main conclusion is supported by extensive studies using *Drosophila* models. Nevertheless, there are some key issues need to be addressed.

1. Studies have suggested that UCHL1 is an integrative factor related to neuropathy such as Parkinson's disease by suppressing glycolysis. Does this function of UCHL1 affect the insulin sensitivity?
2. If UCH mainly expressed in the neurons/brain, why did the dysfunction of UCH induce systemic insulin resistance and hyperglycemia in the KO/KI flies? Is the brain the major organ responsible for the majority of glucose uptake in *Drosophila*?
3. Figure 1j, was the level of Akt phosphorylation lower in the UCH-KO relative to the WT (w1118) at basal?
4. If UCH highly expressed in the sensory neurons, why deletion of UCH in sensory neuron had no effect on glucose level? In fact, glucose level was slightly increased with UCH knocked-down in sensory neurons. In contrast, global knockout of UCH, which was also mainly in the neurons, markedly increased hemolymph glucose. How to reconcile these results?
5. IRS1 is also the downstream of IGF signaling. Would UCH deficiency have any effect on neuronal cell growth/development or brain size?
6. UCH deficiency induces sensory neuron apoptosis. Would this affect the behavior of UCH KO flies, and thus, changes their food intake and indirectly contribute to hyperglycemia?
7. If UCH is mainly expressed in neurons and enhances insulin signaling, would the UCH knockout flies have higher levels of CUL1 in the brain region after HSD?
8. How does HSD elevated the expression of CUL1 is still not clearly elucidated. How did GSK3b become dysregulated after HSD, especially in non-neuronal cells?
9. CUL1 is a component of SCF E3 ligase which is responsible for the turnover of many regulatory proteins. Is the effect of inhibiting CUL1 completely mediated by the upregulation of IRS1? Is there any other factors involved?
10. HEK293E cell is not suitable for assessing insulin signaling. The same test in Figure 3d should be performed in other cell types to support the conclusions regarding changes in the insulin signaling pathway.
11. To translate the observations/conclusions obtained from *Drosophila* to mammalian or human, a mammalian model such as UCHL1 KO mice should be included.

Minor points:

1. For data presented in bar graphs, please utilize dot blots, so the reader can 'see' individual data points.
2. The statistical results of Figures 3d and 4c should be re-evaluated, as $n = 2$?

Reviewer #3 (Remarks to the Author):

This is an interesting and elegant study using the power of *Drosophila* genetics to dissect the role of UCHL1 in regulating insulin signaling in sensory neurons. The work has clinical significance in relation to development of type 2 diabetes. The authors reveal that UCHL1 acts as a deubiquitinase of IRS1, is highly expressed in sensory neurons and under a high fat diet there is suppression of expression in sensory neurons leading to neuronal loss (via an apoptotic mechanism). Strong mechanistic investigations are performed, including rescue studies, to carefully dissect the role of UCHL1 in sensory neuron signaling (with a focus on the insulin pathway). The GSK3B-Snail-CUL1 signaling axis is shown to play a key negative role in regulating UCHL1 expression via IRS1 ubiquitination. Inhibitors of CUL1 neddylation reversed the UCHL1 loss of expression and corrected neuropathy.

I will only make some general comments since I think the data is well presented, quite clear and appropriate statistical tests have been performed. The methods are helpful and accurate. The work shows an interesting and potentially important role for UCHL1 in regulating insulin signaling and could be relevant in the context of development of insulin insensitivity in type 2 diabetes.

However, a major problem with the manuscript is that this particular *Drosophila* model does not accurately reflect development of diabetic neuropathy. Thus, interpretation of the potential role of UCHL1 in sensory neuron signaling under diabetic conditions is fraught with problems. At the core of the problem is the loss of UCHL1 causes apoptotic-induced loss of sensory neuron soma in the forelimb of the flies. In this fly model this leads to the behavioral impairments related to sensory neuropathy. Unfortunately, there is no evidence in humans or animal models of type 1 or type 2 diabetes that apoptosis of sensory neurons occurs.

Early studies using cultures of embryonic sensory neurons did show that high glucose concentrations induced apoptosis of neurons. However, subsequent work in cultured adult sensory neurons revealed that high glucose concentration did not cause sensory neuron cell death. In follow-up work in rat models of type 1 diabetes (from the Zochodne and Sima labs) it was clearly shown that distal fiber loss of sensory neurons occurred early and preceded any neuronal loss in the dorsal root ganglia. Furthermore, any loss of neuronal somata was not via an apoptotic mechanism; in fact, a necrotic-like process was at work (see Kamiya et al 2006 *Diabetologia*).

Therefore, the focus on the analysis of neuronal cell loss via apoptosis is a major flaw. The authors should have assessed axonal atrophy and/or loss as their primary endpoint since diabetic neuropathy is a dying-back neurodegenerative disease. The cited study by Miura et al (1993) would have been a superior approach. However, at the crux of the problem is the evidence that UCHL1 loss leads to apoptosis of sensory neurons – this is simply not reproducing the degenerative process in rodent models of the disease. It should also be added that evidence for neuronal soma loss in the human disease is lacking. Studies by Schmidt et al in 1997 in *JNEN* found no clear pathological evidence of major cell body loss in dorsal root ganglia from post-mortem samples derived from persons with type 1 or 2 diabetes.

In summary, this is a well performed study using the *Drosophila* system to dissect role of UCHL1 in sensory neuron survival and function. A central flaw is the failure to accurately model the nature of development of sensory neuron degeneration in diabetes. The information is certainly relevant to mechanisms of survival of sensory neurons and UCHL1 appears to have an important role in mediating insulin signaling. However, this does not seem to be relevant to the process of neurodegeneration driven by hyperglycemia and/or insulin loss in the context of type 1 or type 2 diabetes.

Point-by-point responses to the reviewers

Our responses to the reviewers' questions are shown in **BLUE** (see below).

Reviewer #1 (Remarks to the Author):

This study describes the use of a fly model to characterize the potential role of UCHL proteins in the pathogenesis of diabetes. It is an interesting study that sheds light on a mechanism that may be important for diabetes. The manuscript includes an extensive and impressive set of experiments that mostly support the conclusions. I support publication, but some concerns need to be addressed first. Most experiments are done with mutants, indicating that the effects could be developmental defects.

The authors should induce knockdown or knockout in the adult stage to exclude this possibility.

> As suggested by the reviewer, we generated flies with inducible *UCH* knockdown using the *UCH* RNAi line, *tub-GAL4*, and *tub-GAL80^{TS}*. The flies were raised at 18°C during larval stages to prevent the expression of *UCH* RNAi. After eclosion, they were grown at 29°C for 3 days to induce *UCH* knockdown, and we measured the levels of hemolymph glucose. Similar to *UCH* mutant flies, flies expressing *UCH* RNAi only during adult stages had higher levels of hemolymph glucose compared to control flies (a). Additionally, we used sensory neuron-specific GAL4, *OK371-GAL4*, *nlsGFP*, and *tub-GAL80^{TS}* to express *UCH* RNAi in adult stages for 30 days and counted the number of sensory nuclei at tarsal segments 3, 4, and 5 of the front legs. We found that the number of sensory nuclei was significantly reduced by the 30-day-induction of *UCH* RNAi (b). (Since the expression of *nlsGFP* was not induced at 18°C, the data for 18°C control flies were not shown). Moreover, we observed delayed escape responses to heat stimuli upon conditional knockdown of *UCH* (c). Based on the fact that the flies with inducible *UCH* knockdown in the adult stages phenocopied the *UCH* knockout flies, we concluded that developmental issues are unlikely to be associated with the diabetic phenotypes observed in *UCH* mutants. **These new data are added in Supplementary Fig. 4a-c.**

a, Relative levels of hemolymph glucose from 3-day-old flies. Normalized to the glucose level of *tub-GAL80^{TS}, tub>+* flies raised at 18°C. Flies of indicated genotypes were raised at 18°C until eclosion. Following eclosion, they were either kept at 18°C or transferred to 29°C for a period of 3 days before hemolymph was extracted. n = 10. **b**, Left, confocal fluorescence images of tarsal segments 3, 4, and 5 at the front legs of 30-day-old flies expressing *OK371>nlsGFP*. The number in panels indicates the number of green signals in each image. Respective images were obtained from one of the left or right front legs. Green, the nucleus of sensory neuron. Scale bar, 20 μm. Right, the numbers of green signals at the tarsal segments 3,

4, and 5 of the front legs of 30-day-old flies expressing *OK371>nlsGFP*. Flies of indicated genotypes were raised at 18°C until eclosion. Following eclosion, they were transferred to 29°C for a period of 30 days before the leg images were obtained. n = 10. **c**, Cumulative percentage of 30-day-old flies showing escape responses on the 42°C hot plate within 10 seconds. Flies of indicated genotypes were raised at 18°C until eclosion. Following eclosion, they were either kept at 18°C or transferred to 29°C for a period of 30 days before the experiments were conducted. n = 100. Data are presented as mean \pm SD. Two-way ANOVA with Sidak's multiple comparison test was used (**a**). Student's t-test was used (**b**). Mantel-Cox test was used (**c**). ****p < 0.0001. ns, no significant.

Since the fly *Unc* is predominantly expressed in the nervous system, its loss likely affects the behavior of the animal. This could indirectly change metabolism and other parameters. If the animal is overeating then that could explain the increased blood sugar levels, and if animals are inactive or sleep disturbed then that could also lead to many other effects. The authors should look more carefully at behavioral changes for example activity and sleep pattern.

> Following the reviewer's suggestion, we investigated behavioral changes, including activity and sleep. We observed that *w1118*, *UCH^{KO}* and *UCH^{C93S}* flies showed a typical bimodal pattern of locomotor activity during the light-dark (LD) cycle, indicating an anticipation of the light-on and light-off periods. Furthermore, during the constant darkness (DD) cycle, both morning and evening peak in the *UCH^{KO}* and *UCH^{C93S}* were slightly delayed, similar to *w1118* control flies (a). Also, the activity pattern of *UCH^{KO}* and *UCH^{C93S}* was comparable to that of control flies (b). Additionally, we measured the sleep time of flies with *UCH* mutations and found that the pattern and duration of sleep in *UCH^{KO}* or *UCH^{C93S}* flies were similar to those in *w1118* flies (c). Therefore, we concluded that the diabetes-like defects observed in *UCH* mutants were not caused by changes in activity or sleep. **These new data are newly added in Supplementary Fig. 4f,g.**

a, Actogram of flies recorded under a 12:12 light-dark cycle (LD) and under constant dark cycle (DD). n = 15. **b**, Daily activity profiles of adult flies. n = 15. **c**, Sleep traces of adult flies.

n = 15. ZT, Zeitgeber time. Data are presented as mean \pm SD. One-way ANOVA with Tukey's multiple comparison test was used (c). ns, no significant.

I was also wondering about the genetic background control. The authors use w1118, which is widely used, but it is not clear that it is the right background control for their mutants. Were the mutants generated in this specific background?

> To generate *UCH^{KO}* mutant flies, we injected plasmids into the embryos of *w1118/nos-Cas9*, where Cas9 is expressed under the *nos* promoter. After injection, we obtained a male line (to exclude female recombination issues) which carries *UCH^{KO}* allele on the *w1118* chromosome. For the generation of *UCH^{C93S}* mutant flies, we injected plasmids into the embryos of *w1118* flies, which we used as one of the control flies in this study. Importantly, we crossed each of these mutant lines with *w1118* flies for 10 generations to ensure that they have the same genetic background as the *w1118* control group. Additional information regarding the generation of mutants can be found in the response to “Other point 2”. **These comments are newly added in the Material and Method section.**

They should also use some combination of trans-heterozygous and they could use heterozygous as control as well, at least for a few key phenotypes.

> As recommended by the reviewer, we generated trans-heterozygous flies which carrying one copy of UCH^{KO} and one copy of UCH^{C93S} , and observed phenotypes resembling those of type 2 diabetes and diabetic neuropathy. Our findings showed that the UCH^{KO}/UCH^{C93S} flies had higher levels of hemolymph glucose compared to those of $+/+$, $UCH^{KO}/+$, and $UCH^{C93S}/+$ flies (a). Also, the trans-heterozygous mutation of UCH led to a reduction in the number of sensory nuclei at the tarsal segments 3, 4, and 5 of the front legs (b). Moreover, we observed decreased escape responses to heat stimuli in flies carrying one copy of UCH^{KO} and one copy of UCH^{C93S} (c). We added these new data in Supplementary Fig. 4h-j.

a, Relative levels of hemolymph glucose from 3-day-old flies. Normalized to the glucose level of $+/+$ flies. $n = 10$. **b**, Left, confocal fluorescence images of tarsal segments 3, 4, and 5 at the front legs of 30-day-old flies expressing $OK371>nlsGFP$. The number in panels indicates the number of green signals in each image. Respective images were obtained from one of the left or right front legs. Green, the nucleus of sensory neuron. Scale bar, 20 μ m. Right, the numbers of green signals at the tarsal segments 3, 4, and 5 of the front legs of 30-day-old flies expressing $OK371>nlsGFP$. $n = 10$. **c**, Cumulative percentage of 30-day-old flies showing escape responses on the 42°C hot plate within 10 seconds. $n = 100$. Data are presented as mean \pm SD. One-way ANOVA with Tukey's multiple comparison test was used (a and b). Mantel-Cox test was used (c). **** $p < 0.0001$. ns, no significant.

Other points:

1. *Drosophila* Unc is more homologous to human UCHL3 than UCHL1. The human genes are obviously paralogs, but it is better to say the fly gene is homologous.

> As the reviewer suggested, we have revised the sentence “*Drosophila UCH* gene (*cg4265*) is the orthologous gene for human *UCHL1*” to “*Drosophila UCH* gene (*cg4265*) is the homologous gene for human *UCHL1*”.

Moreover, we investigated to determine whether human UCHL3 has the ability to deubiquitinate IRS1. In contrast to human UCHL1 and *Drosophila* UCH, we observed that co-expression of *UCHL3* did not reduce the elevated ubiquitination of IRS1 induced by the expression of *IRS1* and *ubiquitin* (a). Therefore, despite suggestions from certain databases, such as FlyBase, proposing a closer homology between *Drosophila UCH* and human *UCHL3*, our findings indicated that the activity of *Drosophila UCH*, at least on IRS1, more closely resembled that of human *UCHL1*.

a, Top, immunoblot analysis of IRS1 ubiquitination in HEK293E cells co-expressing *IRS1*, *ubiquitin* and *UCHL1* or *UCHL3*. The cells were co-transfected with the empty plasmids or the plasmids carrying Flag-tagged *IRS1*, HA-tagged *ubiquitin* (Ub), and *UCHL1* or Myc/His-tagged *UCHL3* upon 40 μ M MG132 treatment for 4 hours to all samples. Bottom, relative quantification of anti-HA immunoblot band intensity from anti-Flag immunoprecipitation normalized to anti-Flag immunoblot band intensity from anti-Flag immunoprecipitation. n = 4.

IP, immunoprecipitation. WCL, whole cell lysate. Data are presented as mean \pm SD. One-way ANOVA with Tukey's multiple comparison test was used (**a**). **** $p < 0.0001$. ns, no significant.

2. They should explain the exact nature of the UCHKO and UCHC93S aberrations and alleles. How was it confirmed that the catalytic domain was disrupted?

> We employed CRISPR-Cas9 system to generate *UCH* knockout and *UCH* C93S knockin mutations.

To generate *UCH* knockout flies, we utilized the *CAS-0001* fly strain obtained from the Japanese National Institute of Genetics, which expresses Cas9 proteins under the control of *nos* promoter. We selected a Cas9 target site located as far forward as possible in the *UCH* first exon and produced single-guide RNA (sgRNA) using in vitro transcription (GCCACTTGAATCTAATCCCG). Since both the *UCH* gene and *nos-Cas9* are located on the second chromosome, we crossed *CAS-0001* with *w1118* and collected the resulting embryos. After injecting sgRNA into the embryos, we crossed the resulting adult male flies with *bc/cyo*. To confirm that DNA breakage had not occurred on the *nos-Cas9* chromosome, we conducted polymerase chain reaction (PCR) to identify that flies did not carry *nos-Cas9*, but instead carried the *w1118* chromosome. We then amplified the target loci by PCR and analyzed the resulting sequences. Through this approach, we generated a line carrying an early-stop frame shift deletion mutation in *UCH* gene (control: MLTWTPLESNPEVLTKYIHK..... -> *UCH*^{KO}: MLTWTPLESKWRGFDQVHT^{stop}) (a) and we crossed the line carrying the deletion mutation with *w1118* for 10 generations.

Also, for generation of *UCH*^{C93S} flies, we injected three different types of plasmids into *w1118* embryos: a Cas9 expression vector (pHsp70-Cas9; #45945, Addgene), sgRNA expression vectors (pU6-Bbs1-chiRNA vector (#45946, Addgene)) and donor plasmids for homologous recombination. The sgRNA (GTTGGCCACGCTGTGGATCA) was designed to target the *UCH* gene and was used to guide Cas9 near cysteine 93 where the C93S mutation should occur. For the donor plasmids, we amplified the homology arms from fly genomic DNA, which were about 2 kb in length and contained a protospacer adjacent motif (PAM) sequence at the center, by PCR. These homology arms were then cloned into the pBS vector, and a point mutation was created using site-directed mutagenesis to change cysteine 93 to serine 93. In addition, as the homology arms contained sequences that the sgRNA could recognize, we modified the sgRNA sequences in the homology arms using site-directed mutagenesis without altering the *UCH* protein sequence. Subsequently, all three plasmids were injected into *w1118* embryos to generate *UCH* C93S knockin flies. We identified the mutations by conducting PCR and subsequent DNA sequencing (b). After that, we crossed the line carrying *UCH* C93S mutation with *w1118* for 10 generations. Using this approach, we were able to generate *UCH*^{C93S} flies.

We confirmed that *UCH*^{KO} flies did not express intact *UCH* mRNA and UCH proteins, and *UCH*^{C93S} flies expressed *UCH* mRNA and UCH proteins at levels similar to those of *w1118* (c, d) (*Drosophila* UCH antibody was kindly gifted from Dr. Thao¹).

In order to confirm the impact of the C93S mutation on the catalytic activity of UCH proteins, we produced UCH WT or C93S recombinant proteins from *Escherichia coli* BL21(DE3) and measured their deubiquitinase (DUB) activity using a DUB activity assay kit (Biovision, K485). As expected, UCH proteins carrying the C93S mutation displayed significantly lower activity compared to UCH WT proteins. Furthermore, treatment with LDN-57444, an inhibitor of UCHL1, reduced the activity of UCH WT to the level of UCH C93S (e). We also used human recombinant UCHL1 proteins and found that UCHL1 with C90S mutation, equivalent to the

C93S mutation in fly UCH, exhibited markedly reduced DUB activity compared to UCHL1 WT proteins. Treatment with LDN-57444 reduced the activity of UCHL1 WT to the level of UCHL1 C90S (f). Based on these experiments, we concluded that the C93S mutation in *Drosophila* UCH and the C90S mutation in human UCHL1 disrupted their respective DUB activities.

As these results have already been fully described in our previous paper on *UCHL1*², we assigned this reference accordingly.

a, Generation of UCH^{KO} mutant flies using CRISPR-Cas9 system. The red arrowhead indicates the location of sgRNA. **b**, Sequencing data of UCH^{C93S} flies. DNA sequence at top panel was from *w1118* and DNA sequence at low panel was from UCH^{C93S} flies. **c**, Immunoblot analysis of UCH proteins in *w1118*, UCH^{KO}, and UCH^{C93S} flies. **d**, Relative expression of UCH normalized to *rp49* expression from *w1118*. n = 10. **e**, Measurement of deubiquitinase (DUB) activities of *Drosophila* UCH WT and UCH C93S proteins. n = 3. **f**, Measurement of DUB activities of human UCHL1 WT and UCHL1 C90S proteins. n = 3. Data are presented as mean ± SD. One-way ANOVA with Tukey's multiple comparison test was used (**d**). ****p < 0.0001. ns, no significant.

3. When comparing hemolymph glucose levels or other parameters it is important that animals are the same development stage. The authors should show whether these mutants are delay in their development. The easiest would be to show timing of pupariation. If there is no delay in pupariation then hemolymph glucose and other parameters can be compared. However, if mutant larvae are delay it is more complicated. A way to address this would be to measure for example 24 h before pupariation in all genotypes. This is also a good time for metabolic measurements because it is before the wandering stage. If there is no developmental delay, this would also indicate the effects observed in the adult are not caused by developmental defects.

> Following the reviewer's feedback, we conducted observations on the morphology of *UCH^{KO}* and *UCH^{C93S}* larvae at various time points including 24, 48, 96, 132, and 180 hours after egg laying (AEL). Our findings revealed that the larvae with *UCH* knockout or *UCH^{C93S}* knockin did not show any developmental defects at these time points (a). Furthermore, we measured the percentage of pupariation and found no differences in pupariation between control flies and *UCH^{KO}* or *UCH^{C93S}* (b). Based on these results, we concluded that the diabetes-like phenotypes of *UCH* mutants were not due to developmental impairments. These results are now added in Supplementary Fig. 4d,e.

a, Images of larvae with indicated genotypes at 24, 48, 96, 132, and 180 hours after egg laying (AEL). Scale bar, 0.5 mm. **b**, Percentage of pupariation at each time point. n = 100. Data are presented as mean \pm SD. One-way ANOVA with Tukey's multiple comparison test was used (b). ns, no significant.

4. It is surprising that both hemolymph glucose and glycogen goes up. If tissues are insulin resistant or insulin signaling is reduced in some way, then increased hemolymph (circulating) glucose is expected. However, tissues should have impaired glucose uptake and therefore decrease glycogen. How do the authors explain this?

> We acknowledge the reviewer's comment that impaired glucose uptake caused by insulin resistance should have led to decreased glycogen levels in cells since glycogen is synthesized from cellular glucose. However, multiple studies have reported elevated glycogen levels in type 2 diabetes animal models or patients. For instance, flies with *glyoxalase 1 (Glo1)* knockout exhibit insulin resistance, which results in increased glycogen levels³. Additionally, glycogen levels are elevated by a high-sucrose diet (HSD), which we utilized to generate T2D model flies in this study^{4,5}. Furthermore, several tissues, including hearts^{6,7}, pancreatic beta cells^{6,7}, and adipose tissues⁸, show higher glycogen levels in patients with T2D. To support our and these results, we stained the brains of *UCH^{KO}* and *UCH^{C93S}* flies using glycogen antibody⁹ and found that glycogen levels were increased in the brains of *UCH^{KO}* and *UCH^{C93S}* flies (a). While these genetic and clinical findings indicated that insulin resistance leads to elevated glycogen levels, the exact mechanisms underlying these phenomena remain unclear, and further research is required to elucidate them.

a, Left, confocal immunofluorescence images of the fly brain, stained with anti-glycogen antibody. Red, glycogen. Scale bar, 20 μ m. Right, the relative fluorescence intensity of glycogen staining normalized to the intensity of *w1118*. n = 10. Data are presented as mean \pm SD. One-way ANOVA with Tukey's multiple comparison test was used. ****p < 0.0001. ns, no significant.

5. Fig. 1d. The authors measure DILP expression, but expression does not reflect insulin secretion. They need to stain for DILP2, -3 and -5 and somehow measure systemic insulin signaling. The best way would be to do a DILP2 ELISA to measure DILP2 in the hemolymph. It would also be good to stain for whole-body phosphorylated AKT.

> As suggested by the reviewer, we stained DILP2 neurons using anti-DILP2 antibody¹⁰. We observed a slight increase of about 15 % in DILP2 staining intensity in *UCH^{KO}* and *UCH^{C93S}* flies compared to control flies (a). To further confirm these findings, we used *DILP2>GFP* and measured the GFP intensity. The GFP intensity was also moderately elevated by the *UCH* mutations (b). Furthermore, we performed ELISA against Flag antigen using the extracted hemolymph of *DILP2>DILP2-Flag*¹⁰. Our results showed that the levels of Flag-tagged DILP2 were increased by about 10% in *UCH^{KO}* and *UCH^{C93S}* flies (c). Therefore, we concluded that *UCH* mutations led to an increase in the expression of DILP2 and circulating DILP2 levels in the hemolymph, which are two of the T2D phenotypes. **These new data are added in Supplementary Fig. 1e-g.**

a, Left, confocal immunofluorescence images of the brain stained with anti-DILP2 antibody. Right, the relative fluorescence intensity of DILP2 staining normalized to the intensity of *w1118*. $n = 20$. Scale bar, 10 μm . **b**, Left, confocal fluorescence images of the brain expressing *DILP2>GFP*. Right, the relative fluorescence intensity of GFP normalized to the intensity of control flies. $n = 20$. Scale bar, 10 μm . **c**, Measuring DILP2-Flag concentration using DILP2-Flag ELISA. $n = 10$. Data are presented as mean \pm SD. One-way ANOVA with Tukey's multiple comparison test was used (**a**, **b**, and **c**). * $p < 0.05$. ** $p < 0.01$. ns, no significant.

6. Fig. 1e. When measuring food intake, it is not sufficient to draw conclusions based on images of flies with food dye color in the abdomens. The author must measure actual food consumption spectrophotometrically.

> According to the reviewer's suggestion, we spectrophotometrically measured the actual food consumption using brilliant blue dye-containing food¹¹. We found that the food uptake of 3-day-old *UCH^{KO}* and *UCH^{C93S}* flies was comparable to that of control flies (**a**). However, at the age of 30 days, *UCH^{KO}* and *UCH^{C93S}* flies exhibited a higher food uptake (**b**). These findings were consistent with those shown in Fig. 1e and Supplementary Fig. 1h. We added these new data in Fig. 1f.

a, Measuring food uptake of 3-day-old flies using spectrophotometric assay. n = 10. **b**, Measuring food uptake of 30-day-old flies using spectrophotometric assay. n = 10. Data are presented as mean \pm SD. One-way ANOVA with Tukey's multiple comparison test was used (**a** and **b**). ****p < 0.0001. ns, no significant.

Fig. 1f. The authors write that it takes significantly longer for the Unc mutant flies to decrease their blood sugar levels, but it seems the slope is the same as the control. This means that they clear blood sugar at the same rate, but they just begin with higher levels. This would not indicate insulin resistance.

> As the reviewer suggested, it appears that the slopes after glucose feeding of *UCH^{KO}* and *UCH^{C93S}* in Fig. 1f (now in Fig. 1g) are similar to that of *w¹¹¹⁸*. However, we observed that the absolute values of the slope from right after the sucrose feeding to 3 hours after the sucrose feeding in *UCH* mutant flies were significantly reduced compared to those in control flies ($|(hemolymph\ glucose\ at\ 3\ hours\ after\ sucrose\ feeding - hemolymph\ glucose\ at\ 0\ hours\ after\ sucrose\ feeding) / 3|$). Based on these results, we concluded that *UCH^{KO}* or *UCH^{C93S}* flies clear hemolymph glucose much more slowly than control flies. Also, we included the slope values in fly GTT data, which can be found in Supplementary Fig. 1m and Supplementary Fig. 5e. **These new data are now added in Supplementary Fig. 1i,m and Supplementary Fig. 5e.**

a. Absolute value of slope of Fig. 1g from right after the sucrose feeding to 3 hours after the sucrose feeding ($|(hemolymph\ glucose\ at\ 0\ hours\ after\ sucrose\ feeding - hemolymph\ glucose\ at\ 3\ hours\ after\ sucrose\ feeding) / 3|$). $n = 3$. Data are presented as mean \pm SD. One-way ANOVA with Tukey's multiple comparison test was used (**a**). ****** $p < 0.01$. ns, no significant.

Fig. 1g. Based on the lower levels of hemolymph glucose in control animals overexpressing DILP2 than mutants, the authors conclude the mutant animals are glucose intolerant and insulin resistant. To draw this conclusion, they need to stimulate tissue with insulin and measure response for example by measuring pAKT, which they do later. But here they cannot yet conclude it.

> We agree with the reviewer's concern that it is not appropriate to draw the conclusion that insulin resistance or glucose intolerance was increased by *UCH* mutations based solely on the data from Fig. 1g,h. Therefore, we have revised the paragraph explaining Fig. 1g,h and mentioned the issues of glucose intolerance or insulin resistance issues after the sentences describing Fig. 1k.

Reviewer #2 (Remarks to the Author):

In this manuscript, Lee et al report that UCHL1, a deubiquitinase, antagonizes the function of E3 ligase Cullin 1 (CUL1) by deubiquitinating insulin receptor substrate 1 (IRS1). Consequently, overexpression of UCH, the homolog of UCHL1 in drosophila, improves insulin sensitivity, while dysfunction of UCH contributes to diabetic neuropathy (DN) and T2D in Drosophila. On the other hand, genetic and pharmacological suppression of CUL1 activity fully rescues the T2D- and DN-associated phenotypes. This is an interesting study of molecular mechanism underlying DN. The main conclusion is supported by extensive studies using Drosophila models. Nevertheless, there are some key issues need to be addressed.

1. Studies have suggested that UCHL1 is an integrative factor related to neuropathy such as Parkinson's disease by suppressing glycolysis. Does this function of UCHL1 affect the insulin sensitivity?

> Our previous paper that the reviewer mentioned² elucidated the glycolytic function of *UCHL1* by suggesting pyruvate kinase (*PKM*) as a direct target of *UCHL1*. According to the reviewer's suggestion, we have examined whether *PKM* can regulate the T2D-related phenotypes of *UCH^{KO}* flies. We found that overexpressing *PKM* in neurons using *elav*-GAL4 did not reduce the elevated hemolymph glucose levels in *UCH* mutant flies (**a**). Similarly, overexpression of *PKM* in sensory neurons using *OK371*-GAL4 did not rescue the loss of leg sensory nuclei (**b**) or the decreased responses to heat stimulation (**c**) in *UCH^{KO}* flies. Furthermore, knockdown of *PKM* in the whole body (*hs*-GAL4) or sensory neurons (*OK371*-GAL4) did not result in any T2D-related pathologies observed in *UCH^{KO}* flies (**d**, **e**, and **f**). Therefore, we concluded that the glycolytic role of *UCHL1* is not related to the role of *UCHL1* in maintaining insulin sensitivity. **We added these new data in Supplementary Fig. 5a-c.**

a, Relative levels of hemolymph glucose from 3-day-old flies. Normalized to the glucose level of WT *elav*>+ flies. n = 10. **b**, Left, confocal fluorescence images of tarsal segments 3, 4, and 5 at the front legs of 30-day-old flies expressing *OK371*>*nlsgFP*. The number in panels indicates the number of green signals in each image. Respective images were obtained from one of the left or right front legs. Green, the nucleus of sensory neuron. Scale bar, 20 μ m. Right, the numbers of green signals in tarsal segments 3, 4, and 5 of the front legs of 30-day-old flies

expressing *OK371>nlsGFP*. n = 10. **c**, Cumulative percentage of 30-day-old flies showing escape responses on the 42°C hot plate within 10 seconds. n = 100. **d**, Relative levels of hemolymph glucose from 3-day-old flies. Normalized to the glucose level of *hs>+* flies. n = 10. **e**, Left, confocal fluorescence images of tarsal segments 3, 4, and 5 at the front legs of 30-day-old flies expressing *OK371>nlsGFP*. The number in panels indicates the number of green signals in each image. Respective images were obtained from one of the left or right front legs. Green, the nucleus of sensory neuron. Scale bar, 20 μm. Right, the numbers of green signals in tarsal segments 3, 4, and 5 of the front legs of 30-day-old flies expressing *OK371>nlsGFP*. n = 10. **f**, Cumulative percentage of 30-day-old flies showing escape responses on the 42°C hot plate within 10 seconds. n = 100. Data are presented as mean ± SD. Two-way ANOVA with Sidak's multiple comparison test was used (**a** and **b**). Mantel-Cox test was used (**c** and **f**). Student's t-test was used (**d** and **e**). ****p < 0.0001. ns, no significant.

2. If UCH mainly expressed in the neurons/brain, why did the dysfunction of UCH induce systemic insulin resistance and hyperglycemia in the KO/KI flies? Is the brain the major organ responsible for the majority of glucose uptake in drosophila?

> As the reviewer pointed out, it is difficult to accept the results that inducing insulin resistance solely in neurons can lead to an increase in hemolymph glucose levels, as it is well known that the main organs responsible for glucose uptake are muscle, fat, and liver in mammals. To validate our findings, we generated flies with tissue-specific knockdown of *IRS1* to induce insulin resistance in neurons, muscles, and fat bodies using *elav*- or *nSyb*-GAL4, *mef2*-GAL4, and *cg*-GAL4, respectively. As expected, we observed increased hemolymph glucose levels in *IRS1* knockdown flies in muscles and fat bodies. Surprisingly, we also found elevated glucose levels in the hemolymph of flies with neuron-specific knockdown of *IRS1* using *elav*-GAL4 or *nSyb*-GAL4. These results led us to conclude that neuronal insulin resistance can contribute to hyperglycemia, at least in *DROSOPHILA*. Furthermore, a previous study has shown that mice with neuron-specific disruption of insulin receptor display diabetic phenotypes, such as obesity and hypertriglyceridemia¹². However, additional investigations are required to verify whether neuronal insulin resistance can induce hyperglycemia in vertebrates and clinical settings. **We added these new data in Supplementary Fig. 6n.**

a, Relative levels of hemolymph glucose from 3-day-old flies. Normalized to the glucose level of +>*IRS1* RNAi flies. n = 10. One-way ANOVA with Tukey's multiple comparison test was used (a). ****p < 0.0001. ns, no significant.

3. Figure 1j, was the level of Akt phosphorylation lower in the UCH-KO relative to the WT (w1118) at basal?

> We do apologize for any confusion caused by the unclear description of the data presented in Fig. 1j (now in Fig. 1k). To clarify, the immunohistochemistry (IHC) data obtained from the brain demonstrated that the level of Akt phosphorylation was decreased in *UCH^{KO}* flies compared to control flies without insulin treatment (at basal). On the other hand, the muscle data indicated that Akt phosphorylation in *UCH^{KO}* flies was similar to that in control flies without insulin treatment. We have revised the sentences explaining Fig. 1j (now Fig. 1k) to provide a clearer understanding of the results.

4. If UCH highly expressed in the sensory neurons, why deletion of UCH in sensory neuron had no effect on glucose level? In fact, glucose level was slightly increased with UCH knocked-down in sensory neurons. In contrast, global knockout of UCH, which was also mainly in the neurons, markedly increased hemolymph glucose. How to reconcile these results?

> As the reviewer noted, Supplementary Fig. 2f showed a slight increase in hemolymph glucose levels with *UCH* knockdown in sensory neurons. However, despite increasing the number of samples for the experiments to $n = 50$ (a, shown below), we did not observe a statistically significant difference between control flies and sensory neuron-specific *UCH* knockdown flies. This could be because sensory neurons make up only a small proportion of all neurons. Therefore, decreased glucose uptake in sensory neurons alone may not be sufficient to produce a statistically significant increase in hemolymph glucose levels. However, in flies with whole-body knockout of *UCH* or pan-neuronal knockdown of *UCH*, glucose uptake could be reduced in all types of neurons, including sensory neurons, motor neurons, and interneurons. This could explain why statistically significant differences were observed in these flies. Our conclusion is that as the number of neurons with reduced glucose uptake increases, the hemolymph glucose levels become more elevated in fruit flies. We have revised the sentences explaining Supplementary Fig. 2f to provide a clearer explanation.

a, Relative levels of hemolymph glucose from 3-day-old flies. Normalized to the glucose level of +>*UCH RNAi* flies. $n = 50$. One-way ANOVA with Tukey's multiple comparison test was used (a). **** $p < 0.0001$. ns, no significant.

5. IRS1 is also the downstream of IGF signaling. Would UCH deficiency have any effect on neuronal cell growth/development or brain size?

> Following the reviewer's suggestion, we aimed to measure the brain size of flies with *UCH* mutations. We found that the brain size of *UCH^{KO}* and *UCH^{C93S}* flies was similar to that of *w¹¹¹⁸* (a and b). To further support this result, we generated the flies with *DIRS1* (*chico*) knockdown using *elav*- and *tub*-GAL4, respectively. Consistent with previous studies¹³, we found that flies with *DIRS1* knockdown throughout the body had smaller brain than control flies. However, to the surprise, our findings indicated that *DIRS1* knockdown using *elav*-GAL4 did not affect the brain size of the flies (c and d). Taken together, these results suggested that the brain size of fruit flies is not solely determined by IGF signaling in neurons, but rather regulated by the systemic effects of various DILPs and IGF signaling. Therefore, *UCH* deficiency, similar to *DIRS1* knockdown in neurons, might have no effect on the regulation of brain size in *Drosophila*. We added these data in Supplementary Fig. 6a-b.

a, Images of the fly brain with indicated genotypes. **b**, Relative width of brain. Normalized to the brain width of *w¹¹¹⁸*. n = 10. Scale bar, 0.1 mm. **c**, Images of the fly brain with indicated genotypes. **c**, Relative width of brain. Normalized to the brain width of *+>* *DIRS1* RNAi. n = 10. Scale bar, 0.1 mm. One-way ANOVA with Tukey's multiple comparison test was used (**b** and **d**). **p < 0.01. ns, no significant.

6. UCH deficiency induces sensory neuron apoptosis. Would this affect the behavior of UCH KO flies, and thus, changes their food intake and indirectly contribute to hyperglycemia?

> We conducted a study to investigate the potential impact of apoptosis in sensory neurons on feeding behavior and hemolymph glucose level. Using *OK371*-GAL4, we knocked down *UCH* specifically in sensory neurons, which resulted in apoptotic signals in the sensory neurons of tarsal segments 3, 4, and 5 in the front legs (a). However, we observed no changes in the feeding behavior of the flies with sensory neuron-specific *UCH* knockdown (b). Additionally, we explored whether blocking sensory neuronal apoptosis could alter feeding behavior. Our findings revealed that sensory neuron-specific knockdown of *death related ICE-like caspase* (*Drice*; *caspase 3* homolog of *Drosophila*¹⁴), one of the primary triggers of apoptosis, completely prevented sensory neuronal apoptosis in *UCH* knockout flies (c). However, the increased food intake observed in the *UCH*^{KO} flies was not affected by the sensory neuron-specific knockdown of *Drice* (d). Similarly, *Drice* knockdown in sensory neurons had no effect on the increased hemolymph glucose levels observed in *UCH*^{KO} flies (e). Based on these findings, it appears that apoptosis in sensory neurons had no significant effect on the feeding behavior or hemolymph glucose levels in *Drosophila*. These new data have been added in Fig. 3a and Supplementary Fig. 3a-d.

a, Confocal fluorescence images for TUNEL assays of tarsal segments 3, 4, and 5 at the front legs of flies expressing *OK371>nlsGFP*. Respective images were obtained from one of the left or right front legs. Red, TUNEL signal. Green, the nucleus of sensory neuron. Scale bar, 20 µm. **b**, Measuring food uptake of 30-day-old flies using CAFE assay. n = 10. **c**, Confocal fluorescence images for TUNEL assays of tarsal segments 3, 4, and 5 at the front legs of flies expressing *OK371>nlsGFP*. Respective images were obtained from one of the left or right front

legs. Red, TUNEL signal. Green, the nucleus of sensory neuron. Scale bar, 20 μm . **d**, Measuring food uptake of 30-day-old flies using CAFE assay. $n = 10$. **e**, Relative levels of hemolymph glucose from 30-day-old flies. Normalized to the glucose level of WT *OK371*>+ flies. $n = 10$. Data are presented as mean \pm SD. Student's t-test was used (**b**). Two-way ANOVA with Sidak's multiple comparison test was used (**d** and **e**). **** $p < 0.0001$. ns, no significant.

7. If UCH is mainly expressed in neurons and enhances insulin signaling, would the UCH knockout flies have higher levels of CUL1 in the brain region after HSD?

> As the reviewer commented, we investigated the expression levels of *CUL1* in the brains of *UCH^{KO}* flies under ND and HSD conditions. We observed a decrease in *CUL1* expression in the brains of *UCH^{KO}* flies compared to control flies under ND. Additionally, while the expression of *CUL1* increased in the brains of *w¹¹¹⁸* flies under HSD, it did not have the same effect on *CUL1* expression in the brains of *UCH^{KO}* flies. Given that the *UCH^{KO}* flies exhibited insulin resistance in neurons, it is reasonable to conclude that the decreased expression of *CUL1* in *UCH^{KO}* brains is attributed to impaired insulin signaling. Also, it is expected that the expression of *CUL1* would not be increased in the brains of *UCH^{KO}* flies in response to HSD. These new data have been added in Supplementary Fig. 8j.

a, Relative expressions of *CUL1* normalized to *rp49* expression from the head of the flies upon ND or HSD. n = 10. ND, normal diet. HSD, high-sucrose diet. Data are presented as mean \pm SD. Two-way ANOVA with Sidak's multiple comparison test was used (**a**). ****p < 0.0001. ns, no significant.

8. How does HSD elevated the expression of CUL1 is still not clearly elucidated. How did GSK3b become dysregulated after HSD, especially in non-neuronal cells?

> When fruit flies are exposed to excessive amounts of nutrients through HSD, it stimulates the secretion of DILPs. As a result, insulin signaling is activated in both neuronal and non-neuronal cells, which leads to the activation of Akt and the inhibition of GSK3B through sequential phosphorylation. This, in turn, stabilizes Snail, which upregulates the expression of *CUL1*. Subsequently, increased CUL1 proteins ubiquitinate and destabilize IRS1, which we propose as a negative feedback mechanism for insulin signaling. However, when overnutrition persists owing to continuous exposure to HSD, CUL1 proteins accumulate in cells and eliminate IRS1 proteins, leading to insulin resistance. Since insulin signaling occurs ubiquitously in all cell types, the dysregulation of the GSK3B-CUL1 axis induces insulin resistance in both neuronal and non-neuronal cells.

To test our hypothesis, we attempted to observe the genetic interaction between one of the DILPs, *DILP2*, and Akt in terms of *CUL1* expression under HSD. However, as genetic intervention of *DILP2* or *Akt* was lethal to fruit flies, we regulated these genes using *tub-GAL80^{TS}*. Firstly, flies with *DILP2* knockdown for 24 hours under normal diet (ND) exhibited reduced levels of *CUL1* expression compared to control flies under ND. Interestingly, the expression of *CUL1* in flies with 24-hour *DILP2* knockdown combined with simultaneous 24-hour HSD was similar to that of flies with 24-hour *DILP2* knockdown under ND (a). On the contrary, flies with *myrAkt* overexpression for 24 hours under ND had elevated *CUL1* expression compared to control flies under ND. Furthermore, 24-hour *myrAkt* expressing flies under simultaneous 24-hour HSD exhibited similar levels of *CUL1* expression compared to 24-hour *myrAkt* expressing flies under ND (a). To our surprise, when both *DILP2* RNAi and *myrAkt* were expressed for 24 hours under ND, the expression of *CUL1* remained increased, similar to the levels observed in flies with *myrAkt* expression for 24 hours under ND (a). Also, *CUL1* expression in flies with *DILP2* knockdown and *myrAkt* expression for 24 hours under simultaneous 24-hour HSD was comparable to the levels observed in flies with *myrAkt* expression for 24 hours under ND (a). These findings led us to conclude that HSD elevated *CUL1* expression via DILP2-Akt axis.

Similarly, we tested the genetic interaction between Akt and GSK3B with respect to *CUL1* expression under HSD. We observed that *CUL1* expression decreased when *Akt* was knocked down for 24 hours under ND. Also, flies with 24-hour *Akt* knockdown under simultaneous 24-hour HSD displayed similar levels of *CUL1* expression compared to flies with 24-hour *Akt* knockdown under ND. Conversely, *CUL1* expression increased when *GSK3B^{DN}* was overexpressed for 24 hours even under ND compared to control flies under ND. Furthermore, *CUL1* expression in flies with 24-hour *GSK3B^{DN}* overexpression under simultaneous 24-hour HSD was similar to that in flies with 24-hour *GSK3B^{DN}* overexpression under ND (b). When *Akt* RNAi and *GSK3B^{DN}* were expressed for 24 hours under ND, *CUL1* expression remained increased, similar to the levels observed in flies with *GSK3B^{DN}* expressing for 24 hours under ND (b). Also, *CUL1* expression in flies with *Akt* knockdown and *GSK3B^{DN}* expression for 24 hours under simultaneous 24-hour HSD was comparable to the levels observed in flies with *GSK3B^{DN}* for 24 hours under ND (b). These results suggested that elevated expression of *CUL1* in response to HSD was a result of the Akt-GSK3B axis.

Based on these findings, we concluded that HSD elevates *CUL1* expression through a sequential mechanism involving DILP2, Akt, and GSK3B.

We added these new results in Supplementary Fig. 8h,i.

a, Relative expressions of *CUL1* normalized to *rp49* expressions from the flies upon ND or HSD. $n = 10$. Flies of indicated genotypes were raised at 18°C. Afterward, the flies were placed at 29°C to induce exogenous expression of *DILP2 RNAi*, *myrAkt*, or both constructs for 24 hours. During this time, HSD groups of flies were fed a high sucrose-containing diet for 24 hours. RNA extractions were then performed right after the 24-hour induction and HSD. $n = 10$. **b**, Relative expressions of *CUL1* normalized to *rp49* expressions from the flies upon ND or HSD. $n = 10$. Flies of indicated genotypes were raised at 18°C. Afterward, the flies were placed at 29°C to induce exogenous expression of *Akt RNAi*, *GSK3B^{DN}*, or both constructs for 24 hours. During this time, HSD groups of flies were fed a high sucrose-containing diet for 24 hours. RNA extractions were then performed right after the 24-hour induction and HSD. $n = 10$. ND, normal diet. HSD, high-sucrose diet. Data are presented as mean \pm SD. Two-way ANOVA with Sidak's multiple comparison test was used (**a** and **b**). **** $p < 0.0001$. ns, no significant.

9. *CUL1* is a component of SCF E3 ligase which is responsible for the turnover of many regulatory proteins. Is the effect of inhibiting *CUL1* completely mediated by the upregulation of *IRS1*? Is there any other factors involved?

> In order to demonstrate that the effects of *CUL1* observed in our manuscript were indeed mediated by *IRS1*, we investigated the genetic interactions between *CUL1* and *IRS1*. Our findings indicated that the increased hemolymph glucose level in flies overexpressing *CUL1* using a whole-body driver, *hs-GAL4*, was rescued when *IRS1* was co-overexpressed (a). Moreover, the loss of sensory nuclei at tarsal segments 3, 4, and 5 of the front legs (b) as well as the decrease in escape responses to heat stimulation (c) observed in flies with sensory neuron-specific *CUL1* overexpression, were both alleviated by the expression of *IRS1*. For further investigation, we tested whether knocking down other known substrates of *CUL1*, such as *cyclin D (CCND)*, *cyclin E (CCNE)*, *cyclin-dependent kinase 4 (CDK4)*, and *wee1 G2 checkpoint kinase (WEE1)* could increase the hemolymph glucose level using *hs-GAL4*. However, knockdown of *CCND*, *CCNE*, *CDK4*, and *WEE1* did not induce increased levels of hemolymph glucose. Only flies with *IRS1* knockdown exhibited higher levels of hemolymph glucose (d). Based on these findings, we concluded that the diabetic effects of *CUL1* are primarily mediated by *IRS1*. We added these new data in Supplementary Fig. 6j-m.

a, Relative levels of hemolymph glucose from 3-day-old flies. Normalized to the glucose level of control *hs>+* flies. *n* = 10. **b**, Left, confocal fluorescence images of tarsal segments 3, 4, and 5 at the front legs of 30-day-old flies expressing *OK371>nlsGFP*. The number in panels indicates the number of green signals in each image. Respective images were obtained from

one of the left or right front legs. Green, the nucleus of sensory neuron. Scale bar, 20 μm . Right, the numbers of green signals at the tarsal segments 3, 4, and 5 of the front legs of 30-day-old flies expressing *OK371>nlsGFP*. n = 10. **c**, Cumulative percentage of 30-day-old flies showing escape responses on the 42°C hot plate within 10 seconds. n = 100. **d**, Relative levels of hemolymph glucose from 3-day-old flies. Normalized to the glucose level of control *hs>+* flies. n = 10. Data are presented as mean \pm SD. Two-way ANOVA with Sidak's multiple comparison test was used (**a** and **b**). Mantel-Cox test was used (**c**). One-way ANOVA with Tukey's multiple comparison test was used (**d**). ****p < 0.0001. ns, no significant.

10. HEK293E cell is not suitable for assessing insulin signaling. The same test in Figure 3d should be performed in other cell types to support the conclusions regarding changes in the insulin signaling pathway.

> Following the reviewer's suggestion, we conducted further experiments to measure the phosphorylation of Akt on serine 473 by manipulating *UCHL1* in various cell lines, including neuroblastoma (SH-SY5Y), mouse embryonic fibroblasts (MEF), and hepatocellular carcinoma (SNU398). Similar to the results obtained in the HEK293E cell line, we observed a decrease in Akt phosphorylation in *UCHL1* KO SH-SY5Y cells compared to *UCHL1* WT SH-SY5Y cells, regardless of insulin treatment (a) (*UCHL1* KO SH-SY5Y cells were generated in previous paper²). Moreover, when we knocked down *UCHL1* in MEF cell line and examined Akt phosphorylation, we found that MEF cells expressing *UCHL1* siRNA showed reduced levels of Akt phosphorylation compared to MEF cells expressing control siRNA with or without insulin treatment (b). Both control siRNA and *UCHL1* siRNA expressing SNU398 cells showed negligible levels of Akt phosphorylation in the absence of insulin treatment. However, when treated with insulin, SNU398 cells expressing control siRNA displayed higher levels of Akt phosphorylation compared to SNU398 cells expressing *UCHL1* siRNA (c). We added these new data in Supplementary Fig. 5f-h.

a, Top, immunoblot analysis of Akt phosphorylation (S473) in SH-SY5Y *UCHL1* WT and KO cells upon insulin treatment. The cells were treated with 50 nM insulin for 2 hours. Bottom, relative quantification of immunoblot band intensity of anti-pAkt (S473) normalized to that of anti-Akt. n = 4. **b**, Top, immunoblot analysis of Akt phosphorylation (S473) in MEF cells transfected with control siRNA or *UCHL1* siRNA (SiUCHL1) upon insulin treatment. The cells were treated with 50 nM insulin for 2 hours. Bottom, relative quantification of immunoblot

band intensity of anti-pAkt (S473) normalized to that of anti-Akt. n = 3. **c**, Top, immunoblot analysis of Akt phosphorylation (S473) in SNU398 cells transfected with control siRNA or *UCHL1* siRNA (siUCHL1) upon insulin treatment. The cells were treated with 50 nM insulin for 2 hours. Bottom, relative quantification of immunoblot band intensity of anti-pAkt (S473) normalized to that of anti-Akt. n = 3. Data are presented as mean \pm SD. One-way ANOVA with Holm-Sidak's multiple comparison test was used (**a**). One-way ANOVA with Tukey's multiple comparison test was used (**b** and **c**) *p < 0.05, **p < 0.01.

11. To translate the observations/conclusions obtained from *Drosophila* to mammalian or human, a mammalian model such as *UCHL1* KO mice should be included.

> We agree with the reviewer's comment that including phenotypes of mammalian models would enhance the robustness of our manuscript. Firstly, we have found previous studies using rodent models that support our findings. In mice models, it has been demonstrated that heterozygous mutation of *UCHL1* accelerates the onset of type 2 diabetes (T2D) of *human islet amyloid polypeptide (hIAPP)*-induced T2D mice¹⁵. Additionally, *UCHL1* knockout mice, known as gracile axonal dystrophy (GAD) mice, exhibit increased glucose intolerance and insulin resistance when subjected to high-fat diet (HFD)¹⁶. Furthermore, administration of MLN4924, an inhibitor of CUL1 neddylation, has been shown to ameliorate glucose intolerance and pyruvate intolerance in mice fed either a chow diet or western diet¹⁷. Based on these findings, we are confident that our observations, wherein *UCHL1* knockout and *CUL1* inhibition induced T2D phenotypes and rescued them, respectively, are consistent with the results observed in mice models.

Next, in line with the reviewer's suggestion, we further examined phenotypes related to diabetic neuropathy and T2D in mice. Regardless of sex, we observed a significant increase in the time it took for *UCHL1* knockout (*UCHL1*^{KO}) mice to jump to avoid thermal pain compared to wild-type (WT) mice (a). Also, we conducted a glucose tolerance test (GTT) on male and female mice of both WT and *UCHL1*^{KO}. At the 60- and 120-minute intervals following glucose injection, we observed elevated blood glucose levels in male *UCHL1*^{KO} mice compared to male WT mice (b, left). Moreover, at the 30-minute mark after glucose injection, we observed higher blood glucose levels in female *UCHL1*^{KO} mice compared to female WT mice (b, right). Additionally, we assessed the ability to clear blood glucose by calculating the area under curve (AUC) from the GTT graphs. In both male and female mice, the AUC was significantly greater in *UCHL1*^{KO} mice compared to WT mice, indicating higher overall blood glucose levels over the observed period (b). These findings led us to conclude that *UCHL1* knockout mice exhibited diabetic neuropathy-like and type 2 diabetes-like phenotypes, similar to our results observed in *Drosophila*. **These results are newly added in Supplementary Fig. 11i,j.**

a, Measuring the latency of first paw pain response in 5-week-old male mice (left) or female mice (right) of WT and *UCHLI*^{KO}. n = 5 (male), 6 (female). **b**, Left, glucose tolerance test (GTT) of 5-week-old male mice of WT and *UCHLI*^{KO}. Glucose was injected at a dosage of 2 mg per gram of body weight, and blood glucose levels were measured before the injection and at 30, 60, 90, and 120 minutes after glucose injection. The bar graph at the upper side represents area under curve (AUC) of the line graph in the same panel. n = 4. Right, GTT of 5-week-old female mice of WT and *UCHLI*^{KO}. Glucose was injected at a dosage of 2 mg per gram of body weight, and blood glucose levels were measured before the injection and at 30, 60, 90, and 120 minutes after glucose injection. The bar graph at the upper side represents AUC of the line graph in the same panel. n = 4. Student's t-test was used (**a** and **b**). *p < 0.05. **p < 0.01.

Minor points:

1. For data presented in bar graphs, please utilize dot blots, so the reader can 'see' individual data points.

> Following the reviewer's suggestion, we have revised all the bar graphs in order to incorporate individual data points (Fig. 4c and Supplementary Fig. 5c (now Supplementary Fig. 5i)).

2. The statistical results of Figures 3d and 4c should be re-evaluated, as $n = 2$?

> As the reviewer suggested, we have conducted an additional experiment to increase the sample size for Fig. 3d and 4c and Supplementary Fig. 5c (now Fig. 4d and 5c and Supplementary Fig. 7c). As a result, the sample size for these figures has been increased from 2 to 3 ($n = 3$). Also, we have reevaluated the statistical analysis using the new data. **These new data have been updated in Fig. 4d and 5c and Supplementary Fig. 7c.**

Reviewer #3 (Remarks to the Author):

This is an interesting and elegant study using the power of *Drosophila* genetics to dissect the role of UCHL1 in regulating insulin signaling in sensory neurons. The work has clinical significance in relation to development of type 2 diabetes. The authors reveal that UCHL1 acts as a deubiquitinase of IRS1, is highly expressed in sensory neurons and under a high fat diet there is suppression of expression in sensory neurons leading to neuronal loss (via an apoptotic mechanism). Strong mechanistic investigations are performed, including rescue studies, to carefully dissect the role of UCHL1 in sensory neuron signaling (with a focus on the insulin pathway). The GSK3B-Snail-CUL1 signaling axis is shown to play a key negative role in regulating UCHL1 expression via IRS1 ubiquitination. Inhibitors of CUL1 neddylation reversed the UCHL1 loss of expression and corrected neuropathy.

I will only make some general comments since I think the data is well presented, quite clear and appropriate statistical tests have been performed. The methods are helpful and accurate. The work shows an interesting and potentially important role for UCHL1 in regulating insulin signaling and could be relevant in the context of development of insulin insensitivity in type 2 diabetes.

However, a major problem with the manuscript is that this particular *Drosophila* model does not accurately reflect development of diabetic neuropathy. Thus, interpretation of the potential role of UCHL1 in sensory neuron signaling under diabetic conditions is fraught with problems. At the core of the problem is the loss of UCHL1 causes apoptotic-induced loss of sensory neuron soma in the forelimb of the flies. In this fly model this leads to the behavioral impairments related to sensory neuropathy. Unfortunately, there is no evidence in humans or animal models of type 1 or type 2 diabetes that apoptosis of sensory neurons occurs.

Early studies using cultures of embryonic sensory neurons did show that high glucose concentrations induced apoptosis of neurons. However, subsequent work in cultured adult sensory neurons revealed that high glucose concentration did not cause sensory neuron cell death. In follow-up work in rat models of type 1 diabetes (from the Zochodne and Sima labs) it was clearly shown that distal fiber loss of sensory neurons occurred early and preceded any neuronal loss in the dorsal root ganglia. Furthermore, any loss of neuronal somata was not via an apoptotic mechanism; in fact, a necrotic-like process was at work (see Kamiya et al 2006 *Diabetologia*).

Therefore, the focus on the analysis of neuronal cell loss via apoptosis is a major flaw. The authors should have assessed axonal atrophy and/or loss as their primary endpoint since diabetic neuropathy is a dying-back neurodegenerative disease. The cited study by Miura et al (1993) would have been a superior approach. However, at the crux of the problem is the evidence that UCHL1 loss leads to apoptosis of sensory neurons – this is simply not reproducing the degenerative process in rodent models of the disease. It should also be added that evidence for neuronal soma loss in the human disease is lacking. Studies by Schmidt et al in 1997 in JNEN found no clear pathological evidence of major cell body loss in dorsal root ganglia from post-mortem samples derived from persons with type 1 or 2 diabetes.

In summary, this is a well performed study using the *Drosophila* system to dissect role of

UCHL1 in sensory neuron survival and function. A central flaw is the failure to accurately model the nature of development of sensory neuron degeneration in diabetes. The information is certainly relevant to mechanisms of survival of sensory neurons and UCHL1 appears to have an important role in mediating insulin signaling. However, this does not seem to be relevant to the process of neurodegeneration driven by hyperglycemia and/or insulin loss in the context of type 1 or type 2 diabetes.

> We highly appreciate the pathological insights on diabetic neuropathy (DN) that were provided by the reviewer. We also agree with the reviewer's assessment that the DN-like phenotypes observed in our manuscript using *Drosophila* might not fully reflect the development of DN in human patients. However, we now have addressed these concerns by performing additional experiments that provide further insight and clarification.

Several studies have presented evidence that apoptosis in sensory neurons may occur in diabetic rodent models as well as human diabetic patients. In dorsal root ganglions (DRGs) dissociated from streptozotocin (STZ)-induced diabetic rats, cytochrome C is released from mitochondria into the cytoplasm, triggering apoptosis¹⁸. Additionally, increased apoptotic signals are observed in DRGs and cultured DRGs from STZ-treated diabetic rats¹⁹. In a recent clinical study utilizing transcriptome analysis, increased expression of *cyclin-dependent kinase 2 (CDK2)*, which can induce neuronal apoptosis²⁰, is observed in the L4 and L5 DRGs of human DN patients. Moreover, moderate to severe loss of L4 DRGs is observed in DN patients²¹. Therefore, considering both these studies and the papers that the reviewer recommended, the loss of sensory neurons can be developed in DN patients, which might be caused by either apoptosis or necrosis.

Subsequently, we investigated whether the loss of sensory neurons and the decreased escape responses to noxious stimulation observed in *UCH^{KO}* flies occurred owing to apoptosis or necrosis. For this purpose, we generated flies that can block apoptosis in sensory neurons by knocking down *death related ICE-like caspase (Drice; caspase 3 homolog of Drosophila¹⁴)* using *OK371-GAL4*. The apoptotic signals in sensory neurons at tarsal segments at 3, 4, and 5 of the *UCH^{KO}* front legs disappeared and the loss of leg sensory neurons in *UCH* knockout flies was alleviated with *Drice* knockdown (**a** and **b**). However, the decreased escape responses to heat stimulation were only partially rescued by blocking apoptosis (**c**). These results suggested that the loss/apoptosis of sensory neurons alone may not fully account for the decreased escape responses, which we proposed as a DN-like phenotype in our manuscript.

Therefore, as suggested by the reviewer, we tried to assess axonal atrophy of leg sensory neurons in *UCH^{KO}* flies. To visualize the axons and somas of sensory neurons located just beneath the surface layer of tarsal segment 4 in the front legs, we employed the sensory neuron-specific *OK371-GAL4* driver to express GFP [Due to the hard cuticle layers of flies, it was challenging to visualize the axons of sensory neurons located deep within the legs of fruit flies. As a result, in the investigation using *OK371>nlsGFP*, eight sensory somas were observed (Fig. 2a,b), while in the observation with *OK371>GFP*, only four sensory neurons were visible (**d**, **e**).] We observed four sensory neurons, with each axon bundled at the center of the leg and extending to the ventral nerve cord (VNC), which is analogous to the vertebrate spinal cord (**d**). Control flies (*OK371/+*) had four sensory neurons, with well-connected axons that remained intact from 5 to 25 days of age (highlighted by a white triangle). In contrast, *UCH^{KO}* flies had

four sensory neurons with initially well-connected axons at 5 days of age, but the axons began to be severed and disconnected from the bundle at the center by 10 days of age (highlighted by a red triangle). By 15 days of age, all observed axons in tarsal segment 4 had degenerated, and the loss of sensory neurons began at 20 days of age. Surprisingly, even in *UCH^{KO}* flies with *Drice* knockdown, sensory axons showed degeneration after 10 days of age. However, sensory neurons were not lost in these flies at 20 and 25 days of age, despite all axons being disconnected (**e**, **f**, and **g**).

Additionally, we measured escape responses to heat stimulation in flies of these genotypes at 5, 10, 15, 20, and 25 days of age. We found that the escape responses to heat stimulation in *UCH^{KO}* and *UCH^{KO}* with *Drice* knockdown were significantly decreased after 10 days of age, when sensory axons began to degenerate. Surprisingly, at 20 days of age, when only *UCH^{KO}* flies showed loss of sensory neurons, *UCH^{KO}* flies exhibited a steeper decline in heat escape responses compared to *UCH^{KO}* flies with *Drice* knockdown. Thus, our findings suggested that the decreased escape responses in *UCH^{KO}* flies resulted from the combined effects of axonal degeneration/atrophy and apoptotic cell death in sensory neurons (**h** and **i**).

To further investigate, we aimed to determine whether axonal degeneration of sensory neurons led to apoptosis/loss of sensory neurons. As genetic strategies to induce or prevent only axonal atrophy in fruit flies were not well-established, we utilized conditional *UCH* expression using *tub-GAL80^{TS}* and overexpressed *UCH* using *OK371-GAL4* at different time points in both control and *UCH^{KO}* flies. When we began to express continuously *UCH* in *UCH^{KO}* at 10 days old, approximately half of the sensory axons were disconnected at 15 days old. Interestingly, although the sensory neurons with degenerated axons disappeared with age, those with the other half of axons, intact axons, did not disappear until they reached 35 days old (**j**). Additionally, when we generated *UCH^{KO}* flies that expressed continuously *UCH* starting at 15 days old, all axons were impaired at 16 days old, and all the sensory neurons with these axons died progressively with age, despite the expression of *UCH* (**k**). While our results did not provide direct evidence that sensory neurons with impaired axons undergo apoptosis, our findings indicated that the death of sensory neurons only occurred in those with axonal atrophy in *UCH* knockout.

In addition to our results, numerous studies have provided evidence that axonal degeneration can trigger neuronal apoptosis. For instance, a study involving retinal ganglion cells (RGCs) extracted from mice with severed optic nerves demonstrated the activation of caspase 3, indicating apoptosis²². Similarly, DRG explants with degenerated axons exhibited positive staining for Annexin V, a marker associated with apoptosis²³. Another study utilizing cultured sensory neurons with genetically induced axonal degeneration revealed elevated levels of cleaved caspase 3 in both the cell bodies and axons²⁴. These findings further support our conclusion that axonal degeneration can induce apoptosis in sensory neurons.

Through these approaches, our findings indicated that the *Drosophila* models of DN with *UCH* mutations exhibited a decrease in leg sensory neurons after the degeneration of sensory axons. Furthermore, these neuronal losses were caused by the apoptotic mechanisms. We acknowledge that further investigations in mammalian or clinical settings are necessary to fully comprehend the precise mechanisms underlying the pathogenesis of human DN. Nevertheless, we firmly believe that our discovery regarding the role of *UCHLI* can contribute to a better

understanding of DN development and aid in the discovery of remedies for patients suffering from DN. We added these new data in Fig. 3 and Supplementary Fig. 3e,f.

a, Confocal fluorescence images for TUNEL assays of tarsal segments 3, 4, and 5 at the front legs of flies expressing *OK371>nlsGFP*. Respective images were obtained from one of the left or right front legs. Red, TUNEL signal. Green, the nucleus of sensory neuron. Scale bar, 20 μ m. **b**, Left, confocal fluorescence images of tarsal segments 3, 4, and 5 at the front legs of 30-day-old flies expressing *OK371>nlsGFP*. The number in panels indicates the number of green signals in each image. Respective images were obtained from one of the left or right front legs. Green, the nucleus of sensory neuron. Scale bar, 5 μ m. Right, the numbers of green signals at the tarsal segments 3, 4, and 5 of the front legs of 30-day-old flies expressing *OK371>nlsGFP*. $n = 10$. **c**, Left, cumulative percentage of 30-day-old flies showing escape responses on the 42°C hot plate within 10 seconds. $n = 100$. Right, the area under curve (AUC) of the left graph in the same panel. $n = 10$. Data are presented as mean \pm SD. Two-way ANOVA with Sidak's multiple comparison test was used (**b** and AUC graph of **c**). **** $p < 0.0001$. ns, no significant.

d, A schematic diagram of *Drosophila* tarsal segment 4 at the front leg showing sensory somas and axons. **e**, Confocal fluorescence images for axons and somas of sensory neurons of tarsal segments 4 at the front legs of flies expressing *OK371>GFP* at 5, 10, 15, 20, and 25 days of age. Respective images were obtained from one of the left or right front legs. Green, sensory neuron. White or red triangles indicate intact or impaired axons, respectively. Scale bar, 10 μ m. **f**, The number of sensory neurons at tarsal segments 4 at the front legs of flies expressing *OK371>GFP* at 5, 10, 15, 20, and 25 days of age. $n = 10$. **g**, The number of intact axons at tarsal segments 4 at the front legs of flies expressing *OK371>GFP* at 5, 10, 15, 20, and 25 days of age. $n = 10$. **h**, Cumulative percentage of 5-, 10-, 15-, 20-, and 25-day-old flies showing escape responses on the 42°C hot plate within 10 seconds. $n = 100$. **i**, The area under curve (AUC) of the graph in “h”. $n = 10$. Data are presented as mean \pm SD. Two-way ANOVA with Sidak’s multiple comparison test was used (**f**, **g**, and **i**). **** $p < 0.0001$. ns, no significant.

j, Confocal fluorescence images for axons and somas of sensory neurons of tarsal segments 4 at the front legs of flies expressing *OK371>GFP* at 15, 20, 25, 30, and 35 days of age. Starting from 10 days of age, the flies were placed at 29°C to induce continuous exogenous expression of *UCH*. Green, sensory neuron. White or red triangles indicate intact or impaired axons, respectively. Scale bar, 5 μm. **k**, Confocal fluorescence images for axons and somas of sensory neurons of tarsal segments 4 at the front legs of flies expressing *OK371>GFP* at 16, 20, 25, 30, and 35 days of age. Starting from 15 days of age, the flies were placed at 29°C to induce continuous exogenous expression of *UCH*. Sensory neurons were observed at 16 days of age owing to the GFP induction for 1 day. Green, sensory neuron. White or red triangles indicate intact or impaired axons, respectively. Scale bar, 5 μm.

References

- 1 Tram, N. T., Trang, N. T., Thao, D. T. & Thuoc, T. L. Production of polyclonal anti-dUCH (Drosophila ubiquitin carboxyl-terminal hydrolase) antibodies. *Monoclon Antib Immunodiagn Immunother* **32**, 105-112, doi:10.1089/mab.2012.0109 (2013).
- 2 Ham, S. J. *et al.* Loss of UCHL1 rescues the defects related to Parkinson's disease by suppressing glycolysis. *Science advances* **7**, eabg4574, doi:10.1126/sciadv.abg4574.
- 3 Moraru, A. *et al.* Elevated Levels of the Reactive Metabolite Methylglyoxal Recapitulate Progression of Type 2 Diabetes. *Cell metabolism* **27**, 926-934.e928, doi:10.1016/j.cmet.2018.02.003 (2018).
- 4 Maiturare, H. M. *et al.* 5,6-dehydrokawain improves glycaemic control by modulating AMPK target genes in Drosophila with a high-sucrose diet-induced hyperglycaemia. *Phytomedicine Plus* **2**, 100261, doi:<https://doi.org/10.1016/j.phyplu.2022.100261> (2022).
- 5 Morris, S. N. S. *et al.* Development of diet-induced insulin resistance in adult Drosophila melanogaster. *Biochimica et Biophysica Acta (BBA) - Molecular Basis of Disease* **1822**, 1230-1237, doi:<https://doi.org/10.1016/j.bbadis.2012.04.012> (2012).
- 6 Ashcroft, F. M., Rohm, M., Clark, A. & Brereton, M. F. Is Type 2 Diabetes a Glycogen Storage Disease of Pancreatic β Cells? *Cell metabolism* **26**, 17-23, doi:<https://doi.org/10.1016/j.cmet.2017.05.014> (2017).
- 7 Sullivan, M. A. & Forbes, J. M. Glucose and glycogen in the diabetic kidney: Heroes or villains? *EBioMedicine* **47**, 590-597, doi:<https://doi.org/10.1016/j.ebiom.2019.07.067> (2019).
- 8 Ceperuelo-Mallafré, V. *et al.* Adipose tissue glycogen accumulation is associated with obesity-linked inflammation in humans. *Molecular Metabolism* **5**, 5-18, doi:<https://doi.org/10.1016/j.molmet.2015.10.001> (2016).
- 9 Nakamura-Tsuruta, S. *et al.* Comparative analysis of carbohydrate-binding specificities of two anti-glycogen monoclonal antibodies using ELISA and surface plasmon resonance. *Carbohydr Res* **350**, 49-54, doi:10.1016/j.carres.2011.12.029 (2012).
- 10 Kwak, S. J. *et al.* Drosophila adiponectin receptor in insulin producing cells regulates glucose and lipid metabolism by controlling insulin secretion. *PLoS One* **8**, e68641, doi:10.1371/journal.pone.0068641 (2013).
- 11 Wong, R., Piper, M. D. W., Wertheim, B. & Partridge, L. Quantification of Food Intake in Drosophila. *PLOS ONE* **4**, e6063, doi:10.1371/journal.pone.0006063 (2009).
- 12 Brüning, J. C. *et al.* Role of Brain Insulin Receptor in Control of Body Weight and Reproduction. *Science (New York, N.Y.)* **289**, 2122-2125, doi:10.1126/science.289.5487.2122 (2000).
- 13 Böhni, R. *et al.* Autonomous Control of Cell and Organ Size by CHICO, a Drosophila Homolog of Vertebrate IRS1-4. *Cell* **97**, 865-875, doi:[https://doi.org/10.1016/S0092-8674\(00\)80799-0](https://doi.org/10.1016/S0092-8674(00)80799-0) (1999).
- 14 McSharry, S. S. & Beitel, G. J. The Caspase-3 homolog DrICE regulates endocytic trafficking during Drosophila tracheal morphogenesis. *Nat Commun* **10**, 1031, doi:10.1038/s41467-

- 019-09009-z (2019).
- 15 Costes, S., Gurlo, T., Rivera, J. F. & Butler, P. C. UCHL1 deficiency exacerbates human islet amyloid polypeptide toxicity in β -cells: evidence of interplay between the ubiquitin/proteasome system and autophagy. *Autophagy* **10**, 1004-1014, doi:10.4161/auto.28478 (2014).
- 16 Chu, K. Y., Li, H., Wada, K. & Johnson, J. D. Ubiquitin C-terminal hydrolase L1 is required for pancreatic beta cell survival and function in lipotoxic conditions. *Diabetologia* **55**, 128-140, doi:10.1007/s00125-011-2323-1 (2012).
- 17 Chen, C. *et al.* Cullin neddylation inhibitor attenuates hyperglycemia by enhancing hepatic insulin signaling through insulin receptor substrate stabilization. *Proceedings of the National Academy of Sciences* **119**, e2111737119, doi:doi:10.1073/pnas.2111737119 (2022).
- 18 Srinivasan, S., Stevens, M. & Wiley, J. W. Diabetic peripheral neuropathy: evidence for apoptosis and associated mitochondrial dysfunction. *Diabetes* **49**, 1932-1938, doi:10.2337/diabetes.49.11.1932 (2000).
- 19 Russell, J. W., Sullivan, K. A., Windebank, A. J., Herrmann, D. N. & Feldman, E. L. Neurons Undergo Apoptosis in Animal and Cell Culture Models of Diabetes. *Neurobiology of Disease* **6**, 347-363, doi:<https://doi.org/10.1006/nbdi.1999.0254> (1999).
- 20 COPANI, A. *et al.* Mitotic signaling by β -amyloid causes neuronal death. *The FASEB Journal* **13**, 2225-2234, doi:<https://doi.org/10.1096/fasebj.13.15.2225> (1999).
- 21 Hall, B. E. *et al.* Transcriptomic analysis of human sensory neurons in painful diabetic neuropathy reveals inflammation and neuronal loss. *Scientific Reports* **12**, 4729, doi:10.1038/s41598-022-08100-8 (2022).
- 22 Finn, J. T. *et al.* Evidence That Wallerian Degeneration and Localized Axon Degeneration Induced by Local Neurotrophin Deprivation Do Not Involve Caspases. *The Journal of Neuroscience* **20**, 1333-1341, doi:10.1523/jneurosci.20-04-01333.2000 (2000).
- 23 Sievers, C., Platt, N., Perry, V. H., Coleman, M. P. & Conforti, L. Neurites undergoing Wallerian degeneration show an apoptotic-like process with Annexin V positive staining and loss of mitochondrial membrane potential. *Neurosci Res* **46**, 161-169, doi:10.1016/s0168-0102(03)00039-7 (2003).
- 24 Simon, D. J. *et al.* Axon Degeneration Gated by Retrograde Activation of Somatic Pro-apoptotic Signaling. *Cell* **164**, 1031-1045, doi:10.1016/j.cell.2016.01.032 (2016).

REVIEWER COMMENTS

Reviewer #1 (Remarks to the Author):

The authors have significantly revised the manuscript and addressed the concerns. I believe these revisions have significantly improved the work and strengthened the conclusion. Consequently, I recommend publication. However, during the submission process, I suggest they modify the graph in Fig. 1d, which currently depicts Dilp expression as a heatmap, to a format that clearly presents statistics, enabling readers to see significant changes.

Reviewer #2 (Remarks to the Author):

The author performed additional experiments and provided new data to address my questions. I am generally satisfied with their responses and have no further comments.

Reviewer #3 (Remarks to the Author):

In this resubmission the authors have included some persuasive new data showing UCHL1 knockout-induced loss of distal sensory axons in the fly does occur and appears to precede apoptosis of sensory neurons in the leg. The expression of the fly version of caspase 3 is manipulated to show that distal axon loss is quite separate from any apoptotic process that may be occurring. This is interesting data and does fit more with the distal dying-back process of diabetic neuropathy in vertebrates as the field understands. Considerable new data is added regarding other aspects of the work (mostly in response to the other reviewers). I have the remaining concerns:

1. The title is unclear – UCHL1 is not controlling diabetic neuropathy – its loss or down-regulation is leading to diabetic sensory neuropathy (there was no effect on motor neuropathy and autonomic neurons were not studied)
2. In the introduction if sensory neurons are indeed insensitive to insulin then insulin therapy in T2D should be ineffective in correcting neuropathy – is this true? In addition, what evidence is there that insulin insensitivity even occurs in sensory neurons in rodent models of T1D or T2D?
3. Throughout the paper the actual concentration of glucose (and other metabolites) is never included – we just see relative values. Why is this?
4. In the results section looking at the DILP2 effects the text should state clearly that statistically significant effects were seen (Supp Fig.1e-g)
5. Fig.1h did not show any significant effect of UCH manipulation under DILP2 elevated expression
6. The failure of UCH knockout/mutant flies to show any effect on motor neuron function does weaken the relevance of this fly model – diabetic neuropathy is a polyneuropathy where sensory, motor and autonomic neurons are all impaired
7. Supp Fig.5 presents new data in various cell lines focused on Akt signalling. Given some of the disconnects between this fly model of diabetic neuropathy (more accurately diabetic sensory neuropathy) and the vertebrate models of diabetic neuropathy it would have been much stronger to include some work on cultured adult sensory neurons in this section. In addition, confirming high expression of UCHL1 in adult DRG sensory neurons would have added major support for the central rationale of the work
8. A major issue remains the almost total focus on UCH knockout/mutant-induced loss of sensory neurons as the key endpoint. In figures 5-7 where the signalling axis is being dissected there is no

attempt to measure axonal loss as nicely described in Fig.3. This is key since the authors show that correcting apoptosis only partially corrected the neuropathic phenotype
9. I find the discussion weak in its failure to address the focus on apoptosis in the fly model – even though in vertebrates with diabetic neuropathy the role of apoptosis of sensory neurons has been shown to be irrelevant

Specific comments around the authors rebuttal

The authors cite papers by Russell et al (1999) and Srinivasan et al (1999) to support presence of apoptosis in sensory neurons in T1D. These papers did not accurately measure neuronal loss and these studies were short term – 6 weeks to 3 months of diabetes. At these time points no axonal loss is observed at mid-sciatic nerve in the STZ-rat model and so no cell loss can be occurring – this has been confirmed multiple times (ad nauseam). Also, follow-up studies in long term STZ-rats out to 12 months of diabetes using proper cell counting techniques found no evidence of any significant loss of neurons (Zochodne et al 2001). The field now accepts (apart from 1 or 2 stubborn outliers) that apoptosis is not relevant in the context of diabetic neuropathy. Markers of apoptosis do appear as a sign of stress but this does not proceed to cell death in adult sensory neurons.

Therefore, the remaining central focus on apoptosis and sensory cell loss remains an issue in terms of relevance to disease in vertebrates. The data presented in Figs. 5-7 would be much stronger if distal axonal loss is the primary endpoint given its central role under diabetic conditions where a distal dying-back neuropathy is the primary neurodegenerative process.

Point-by-point responses to the reviewer's comments [Answers are shown in **BLUE**]

Reviewer #1 (Remarks to the Author):

Q) The authors have significantly revised the manuscript and addressed the concerns. I believe these revisions have significantly improved the work and strengthened the conclusion. Consequently, I recommend publication. However, during the submission process, I suggest they modify the graph in Fig. 1d, which currently depicts Dilp expression as a heatmap, to a format that clearly presents statistics, enabling readers to see significant changes.

A) In response to the comment from the Reviewer #1, we have altered the graphical representation of Fig. 1d to bar graphs with statistics.

Reviewer #3 (Remarks to the Author):

In this resubmission the authors have included some persuasive new data showing UCHL1 knockout-induced loss of distal sensory axons in the fly does occur and appears to precede apoptosis of sensory neurons in the leg. The expression of the fly version of caspase 3 is manipulated to show that distal axon loss is quite separate from any apoptotic process that may be occurring. This is interesting data and does fit more with the distal dying-back process of diabetic neuropathy in vertebrates as the field understands. Considerable new data is added regarding other aspects of the work (mostly in response to the other reviewers). I have the remaining concerns:

Q1) The title is unclear – UCHL1 is not controlling diabetic neuropathy – its loss or down-regulation is leading to diabetic sensory neuropathy (there was no effect on motor neuropathy and autonomic neurons were not studied)

A1) In accordance with the Reviewer #3's suggestion, we have revised the title of our manuscript to "Diabetic sensory neuropathy and insulin resistance are induced by loss of *UCHLI*".

Q2) In the introduction if sensory neurons are indeed insensitive to insulin then insulin therapy in T2D should be ineffective in correcting neuropathy – is this true? In addition, what evidence is there that insulin insensitivity even occurs in sensory neurons in rodent models of T1D or T2D?

A2) Multiple studies have reported the ineffectiveness of insulin therapy for improving diabetic neuropathy in patients with type 2 diabetes (T2D). One clinical study focused on elderly T2D patients, aged 67 and older, who received 0.52 units of insulin per kilogram of body weight daily for one year. This study found that diabetic neuropathic symptoms did not improve with the insulin therapy¹. Furthermore, despite several limitations, including the duration of diabetes and economic status, the prevalence of diabetic neuropathy was higher in insulin-treated T2D patients compared to non-insulin-treated patients in Bangladesh and Iran^{2,3}. Consistent with these clinical findings, several studies have demonstrated that glycemic control, the primary aim of insulin therapy in T2D patients, is insufficient to alleviate diabetic neuropathy in these patients. T2D patients in the UK who managed their glycemia with sulfonylurea, insulin, or dietary control did not show significant improvement in the biothesiometer reading, a diagnostic method for diabetic neuropathy, compared to T2D patients without glycemic control⁴. Moreover, the prevalence of diabetic neuropathy significantly increased after a two-year follow-up in T2D patients receiving standard glycemic control with one morning insulin injection per day, as well as in various intensive glycemic control groups⁵. There were no significant differences in neuropathy symptoms observed between the intensive therapy group and the standard therapy group during a 5.6-year follow-up period⁶. Based on these references, we described that insulin therapy or glycemic control are not efficient to correct diabetic neuropathy in T2D patients.

Also, recent findings suggest that insulin resistance is increased in sensory neurons of rodent models with T2D. In a study involving T2D *ob/ob* mice model, it was observed that intrathecal insulin injection activated Akt in the dorsal root ganglia (DRG) of control mice, while the DRG of *ob/ob* mice did not show an increase in pAkt levels following insulin injection⁷. Furthermore, primary cultures of lumbar DRG from *ob/ob* mice exhibited reduced levels of phosphorylated Akt upon insulin treatment when compared to cultures from nondiabetic control mice⁸. Additionally, DRG

cultures obtained from 16-week-old and 24-week-old *db/db* mice demonstrated reduced pAkt levels following one-hour insulin treatment in comparison to DRG cultures from age-matched control mice⁹.

In the Introduction and Discussion section of the newly revised manuscript, we have included several new sentences to further explain the aforementioned previous results.

Q3) Throughout the paper the actual concentration of glucose (and other metabolites) is never included – we just see relative values. Why is this?

A3) Many *Drosophila* researchers have traditionally presented hemolymph metabolite levels, such as glucose, TAG, glycogen, and trehalose, using relative values. Nevertheless, following further considerations including the reviewer's comment, we have now converted the relative concentrations back into the corresponding actual metabolite concentrations.

Q4) In the results section looking at the DILP2 effects the text should state clearly that statistically significant effects were seen (Supp Fig. 1e-g)

A4) In response to the reviewer's feedback, we have revised the sentences describing Supplementary Fig. 1e-g, which pertains to the experiments involving DILP2, in a clearer manner. We have now included explicit mentions of the statistical significance in the sentences.

Q5) Fig. 1h did not show any significant effect of UCH manipulation under DILP2 elevated expression

A5) We appreciate the reviewer's point that the graph in Fig. 1h might confuse the readers. As also raised in Q3, we have replaced the previous graph in Fig. 1h with a new one that accurately represents the actual concentrations of hemolymph glucose. The previous graph has been moved to Supplemental Fig. 1j. In addition, to enhance the precision of our results, we have incorporated asterisks to indicate statistical significance at each time interval. In this new graph, following a 0.5-hour DILP2 expression, the glucose levels in control flies (*w¹¹¹⁸*) exhibited a noteworthy decrease with statistical significance when compared to both *UCH^{KO}* and *UCH^{C93S}* with p-values less than 0.0001 indicated with four asterisks. Furthermore, we provided a detailed explanation of the relative hemolymph glucose levels "from baseline" in Supplemental Fig. 1j to ensure clarity for the readers.

Q6) The failure of UCH knockout/mutant flies to show any effect on motor neuron function does weaken the relevance of this fly model – diabetic neuropathy is a polyneuropathy where sensory, motor and autonomic neurons are all impaired

A6) We would like to acknowledge the reviewer's comment regarding the limitation of our manuscript in demonstrating solely sensory neuronal defects. Due to the comparatively lower expression of *UCH* in motor neurons, as indicated in Supplementary Fig. 2b, any potential defective phenotypes related to motor neurons or autonomic neurons remained elusive. In light of this, we have opted to revise the manuscript, changing "diabetic neuropathy" to "diabetic sensory

neuropathy", thereby properly describing the observed results in this study.

Q7) Supp Fig.5 presents new data in various cell lines focused on Akt signalling. Given some of the disconnects between this fly model of diabetic neuropathy (more accurately diabetic sensory neuropathy) and the vertebrate models of diabetic neuropathy it would have been much stronger to include some work on cultured adult sensory neurons in this section. In addition, confirming high expression of UCHL1 in adult DRG sensory neurons would have added major support for the central rationale of the work

A7) Following the reviewer's suggestion, we attempted to carry out experiments aimed at observing insulin signaling and *UCHL1* expression in cultured sensory neurons from DRGs. However, we were not well-versed in the process of culturing sensory neurons from DRGs, so we instead conducted these experiments using antibody staining against pAkt and UCHL1 on sectioned DRGs.

Firstly, we performed antibody staining on sectioned DRG samples from WT (C57BL/6J) mice using anti-neurofilament-H (NF-H) antibody to label sensory neurons¹⁰ and anti-UCHL1 antibody. Our findings revealed that all cells expressing NF-H were also positively stained with anti-UCHL1 antibody. Moreover, we observed a significantly higher staining intensity of UCHL1 specifically in sensory neurons, which were identified as DAPI⁺NF-H⁺ cells, among the various cell types present within the DRG samples (**a**). This observation aligns with our results in *Drosophila*, where we observed a robust expression of *UCH* in sensory neurons. Additionally, we conducted experiments to investigate insulin signaling in sensory neurons within the DRGs. To assess this, we quantified the staining intensity of pAkt in cells that were positively stained for DAPI and NF-H. While the staining intensity of Akt in the DRG sensory neurons was comparable between WT and UCHL1KO mice (**b**), our immunohistochemistry findings indicated a notable decrease in pAkt levels in the DRG sensory neurons of UCHL1KO mice compared to control mice (**c**). These results were consistent with our cell data from Supplementary Fig. 5g-i.

These new results were now added in Supplementary Fig. 12c-e.

a, Left, confocal immunofluorescence images of sectioned DRG from WT (C57BL/6J) mice. Red, NF-H. Green, UCHL1. Blue, DAPI. Scale bar, 50 μ m. Right, the relative fluorescence intensity of UCHL1 (green) staining between DAPI⁺NF-H⁺ and DAPI⁺NF-H⁻ cells from DRG samples. n = 495 DAPI⁺NF-H⁺ cells, 395 DAPI⁺NF-H⁻ cells from 5 mice. **b**, Left, confocal immunofluorescence images of sectioned DRG from WT or *UCHL1*^{KO} mice. Red, Akt. Green, NF-H. Blue, DAPI. Scale bar, 50 μ m. Right, the relative fluorescence intensity of Akt (red) staining in DAPI⁺NF-H⁺ cell between WT and *UCHL1*^{KO} mice. n = 528 DAPI⁺NF-H⁺ cells from 4 WT mice, 702 DAPI⁺NF-H⁺ cells from 5 *UCHL1*^{KO} mice. **c**, Left, confocal immunofluorescence images of sectioned DRG from WT or *UCHL1*^{KO} mice. Red, pAkt. Green, NF-H. Blue, DAPI. Scale bar, 50 μ m. Right, the relative fluorescence intensity of pAkt (red) staining in DAPI⁺NF-H⁺ cell between WT and *UCHL1*^{KO} mice. n = 851 DAPI⁺NF-H⁺ cells from 7 WT mice, 1164 DAPI⁺NF-H⁺ cells from 8 *UCHL1*^{KO} mice. Student's t-test was used (**a**, **b**, and **c**). ****p < 0.0001. ns, no significant.

Q8) A major issue remains the almost total focus on UCH knockout/mutant-induced loss of sensory neurons as the key endpoint. In figures 5-7 where the signalling axis is being dissected there is no

attempt to measure axonal loss as nicely described in Fig.3. This is key since the authors show that correcting apoptosis only partially corrected the neuropathic phenotype.

A8) In response to the reviewer's suggestion, we have incorporated new axonal images of sensory neurons and conducted a comprehensive assessment of axonal loss throughout the manuscript. These new additions include Figures 4, 5, 6, and 7 and Supplementary Figures 4, 5, 6, 9, 10, and 11 that illustrate the axonal morphologies. Notably, we have focused on presenting axonal images from the 15-day-old flies, a critical time point when the distinct onset of axonal degeneration becomes evident (Fig. 3e,g). Given that the hallmark of diabetic sensory neuropathy lies in the axonal atrophy of sensory neurons, our manuscript places particular emphasis on showcasing these axonal images as the primary phenotype of interest as suggested by the reviewer.

Q9) I find the discussion weak in its failure to address the focus on apoptosis in the fly model – even though in vertebrates with diabetic neuropathy the role of apoptosis of sensory neurons has been shown to be irrelevant.

A9) In response to the reviewer's recommendation, we have included a new paragraph within the discussion section. This paragraph underscores that the predominant factor contributing to diabetic sensory neuropathy in human patients as well as our *Drosophila* mutant model is the axonal loss of sensory neurons, not neuronal loss or apoptosis.

Specific comments around the authors rebuttal:

Q) The authors cite papers by Russell et al (1999) and Srinivasan et al (1999) to support presence of apoptosis in sensory neurons in T1D. These papers did not accurately measure neuronal loss and these studies were short term – 6 weeks to 3 months of diabetes. At these time points no axonal loss is observed at mid-sciatic nerve in the STZ-rat model and so no cell loss can be occurring – this has been confirmed multiple times (ad nauseam). Also, follow-up studies in long term STZ-rats out to 12 months of diabetes using proper cell counting techniques found no evidence of any significant loss of neurons (Zochodne et al 2001). The field now accepts (apart from 1 or 2 stubborn outliers) that apoptosis is not relevant in the context of diabetic neuropathy. Markers of apoptosis do appear as a sign of stress but this does not proceed to cell death in adult sensory neurons.

Therefore, the remaining central focus on apoptosis and sensory cell loss remains an issue in terms of relevance to disease in vertebrates. The data presented in Figs. 5-7 would be much stronger if distal axonal loss is the primary endpoint given its central role under diabetic conditions where a distal dying-back neuropathy is the primary neurodegenerative process.

A) We appreciate the reviewer's feedback concerning the disparity between diabetic neuropathy and sensory neuronal cell death. As the reviewer pointed out, this research did not establish a direct correlation between **diabetic neuropathy** and apoptosis in sensory neurons. As a result, we have now included a discussion that emphasizes the central role of axonal degeneration in diabetic neuropathy, rather than focusing on apoptosis or neuronal loss. Additionally, we have provided a comprehensive explanation of the lack of relevance between diabetic sensory neuropathy and the observed loss (apoptosis) of sensory neurons in this manuscript. Moving forward, our future studies will be

dedicated to investigating the precise mechanisms through which UCHL1 regulates axonal degeneration independently of apoptosis in sensory neurons. We intend to explore the underlying impacts of insulin signaling on the axonal degeneration of sensory neurons in diabetic neuropathy by identifying related molecular factors using *Drosophila* genetics. We now added these comments in the Discussion section.

References

- 1 Tovi, J., Svanborg, E., Nilsson, B. Y. & Engfeldt, P. Diabetic neuropathy in elderly Type 2 diabetic patients: effects of insulin treatment. *Acta Neurol Scand* **98**, 346-353, doi:10.1111/j.1600-0404.1998.tb01746.x (1998).
- 2 Mørkrid, K., Ali, L. & Hussain, A. Risk factors and prevalence of diabetic peripheral neuropathy: A study of type 2 diabetic outpatients in Bangladesh. *Int J Diabetes Dev Ctries* **30**, 11-17, doi:10.4103/0973-3930.60004 (2010).
- 3 Janghorbani, M. *et al.* Peripheral neuropathy in type 2 diabetes mellitus in Isfahan, Iran: Prevalence and risk factors. *International Journal of Diabetes and Metabolism* **14**, 126-133, doi:10.1159/000497604 (2019).
- 4 Intensive blood-glucose control with sulphonylureas or insulin compared with conventional treatment and risk of complications in patients with type 2 diabetes (UKPDS 33). *The Lancet* **352**, 837-853, doi:10.1016/S0140-6736(98)07019-6 (1998).
- 5 Azad, N. *et al.* The Effects of Intensive Glycemic Control on Neuropathy in the VA Cooperative Study on Type II Diabetes Mellitus (VA CSDM). *Journal of Diabetes and its Complications* **13**, 307-313, doi:https://doi.org/10.1016/S1056-8727(99)00062-8 (1999).
- 6 Duckworth, W. *et al.* Glucose Control and Vascular Complications in Veterans with Type 2 Diabetes. *New England Journal of Medicine* **360**, 129-139, doi:10.1056/NEJMoa0808431 (2009).
- 7 Grote, C. W. *et al.* Peripheral nervous system insulin resistance in ob/ob mice. *Acta Neuropathologica Communications* **1**, 15, doi:10.1186/2051-5960-1-15 (2013).
- 8 Grote, C. W., Morris, J. K., Ryals, J. M., Geiger, P. C. & Wright, D. E. Insulin Receptor Substrate 2 Expression and Involvement in Neuronal Insulin Resistance in Diabetic Neuropathy. *Experimental Diabetes Research* **2011**, 212571, doi:10.1155/2011/212571 (2011).
- 9 Kim, B., McLean, L. L., Philip, S. S. & Feldman, E. L. Hyperinsulinemia induces insulin resistance in dorsal root ganglion neurons. *Endocrinology* **152**, 3638-3647, doi:10.1210/en.2011-0029 (2011).
- 10 Genç, B. *et al.* Visualization of Sensory Neurons and Their Projections in an Upper Motor Neuron Reporter Line. *PLOS ONE* **10**, e0132815, doi:10.1371/journal.pone.0132815 (2015).

REVIEWERS' COMMENTS

Reviewer #3 (Remarks to the Author):

The authors have rigorously addressed all my concerns. I think the inclusion of the additional axonal degeneration work now balances optimally with the cell body loss findings. This is an interesting paper which introduces a novel target for drug development in diabetic neuropathy. The work is almost entirely focused on Drosophila work and so I feel the title should indicate this fact.

Point-by-point responses to the reviewer's comments [Answers are shown in **BLUE**]

Reviewer #3 (Remarks to the Author):

The authors have rigorously addressed all my concerns. I think the inclusion of the additional axonal degeneration work now balances optimally with the cell body loss findings. This is an interesting paper which introduces a novel target for drug development in diabetic neuropathy. The work is almost entirely focused on *Drosophila* work and so I feel the title should indicate this fact.

A) Following the reviewer's suggestion, we have changed the manuscript title to highlight our *Drosophila* work: Diabetic sensory neuropathy and insulin resistance are induced by loss of *UCHL1* in *Drosophila*.